# Initialization Matters: Privacy-Utility Analysis of Overparameterized Neural Networks

Jiayuan Ye[†], Zhenyu Zhu[‡], Fanghui Liu[♮],[*] Reza Shokri[†], Volkan Cevher[‡]

[†] National University of Singapore    [‡] EPFL, Switzerland    [♮] University of Warwick
[†]{jiayuan,reza}@comp.nus.edu.sg   [‡]{zhenyu.zhu, volkan.cevher}@epfl.ch
[♮]fanghui.liu@warwick.ac.uk

## Abstract

We analytically investigate how over-parameterization of models in randomized machine learning algorithms impacts the information leakage about their training data. Specifically, we prove a privacy bound for the KL divergence between model distributions on worst-case neighboring datasets, and explore its dependence on the initialization, width, and depth of fully connected neural networks. We find that this KL privacy bound is largely determined by the expected squared gradient norm relative to model parameters during training. Notably, for the special setting of linearized network, our analysis indicates that the squared gradient norm (and therefore the escalation of privacy loss) is tied directly to the per-layer variance of the initialization distribution. By using this analysis, we demonstrate that privacy bound improves with increasing depth under certain initializations (LeCun and Xavier), while degrades with increasing depth under other initializations (He and NTK). Our work reveals a complex interplay between privacy and depth that depends on the chosen initialization distribution. We further prove excess empirical risk bounds under a fixed KL privacy budget, and show that the interplay between privacy utility trade-off and depth is similarly affected by the initialization.

## 1 Introduction

Deep neural networks (DNNs) in the over-parameterized regime (i.e., more parameters than data) perform well in practice but the model predictions can easily leak private information about the training data under inference attacks such as membership inference attacks [44] and reconstruction attacks [17, 7, 29]. This leakage can be mathematically measured by the extent to which the algorithm's output distribution changes if DNNs are trained on a neighboring dataset (differing only in a one record), following the differential privacy (DP) framework [23].

To train differential private model, a typical way is to randomly perturb each gradient update in the training process, such as stochastic gradient descent (SGD), which leads to the most widely applied DP training algorithm in the literature: DP-SGD [2]. To be specific, in each step, DP-SGD employs gradient clipping, adds calibrated Gaussian noise, and yields differential privacy guarantee that scales with the noise multiplier (i.e., per-dimensional Gaussian noise standard deviation divided by the clipping threshold) and number of training epochs. However, this privacy bound [2] is overly general as its gradient clipping artificially neglects the network properties (e.g., width and depth) and training schemes (e.g., initializations). Accordingly, a natural question arises in the community:

*How does the over-parameterization of neural networks (under different initializations) affect the privacy bound of the training algorithm over* worst-case *datasets?*

---

[*]Work was done while Fanghui was at LIONS, EPFL.

37th Conference on Neural Information Processing Systems (NeurIPS 2023).

Table 1: Our privacy utility trade-off bounds for training linearized network (3) via Langevin diffusion, under different hidden-layer width $m$, depth $L$ and initializations. The per-layer widths $m_0 = d$, $m_1, \cdots, m_{L-1} = m$ and $m_L = o$ where $d$ is the data dimension and $o$ is number of classes. For KL privacy bounds, we assume Assumption 2.2 holds and $L \geq 2$ for simplicity. For the excess risk bounds, we assume $o = 1$, $d, m = \tilde{\Omega}(n)$ are large, and Assumption 2.2. Under LeCun and Xavier, we prove privacy utility trade-offs that improve with over-parameterization (increasing depth).

| Initialization | Variance $\beta_l$ for layer $l$ | Gradient norm constant $B$ (7) | Approximate lazy training distance $R$ (9) | Excess Empirical risk under $\varepsilon$-KL privacy (Corollary 6.4) |
|---|---|---|---|---|
| LeCun [34] | $1/m_{l-1}$ | $\frac{om(L-1+\frac{d}{m})}{2^{L-1}}$ | $\tilde{\mathcal{O}}\left(\frac{n}{m}\cdot\frac{1}{L-1+\frac{d}{m}}\right)$ | $\tilde{\mathcal{O}}\left(\frac{1}{n^2}+\sqrt{\frac{1}{2^L\varepsilon}}\right)$ |
| He [30] | $2/m_{l-1}$ | $om(L-1+\frac{d}{m})$ | $\tilde{\mathcal{O}}\left(\frac{n}{m}\cdot\frac{1}{L-1+\frac{d}{m}}\right)$ | $\tilde{\mathcal{O}}\left(\frac{1}{n^2}+\sqrt{\frac{1}{\varepsilon}}\right)$ |
| NTK [3] | $2/m_l, l<L$ $1/o, l=L$ | $dm\left(\frac{L-1}{2}+\frac{o}{m}\right)$ | $\tilde{\mathcal{O}}\left(\frac{n}{m}\cdot\frac{1}{\frac{L-1}{2}+\frac{1}{m}}\right)$ | $\tilde{\mathcal{O}}\left(\frac{1}{n^2}+\sqrt{\frac{d}{\varepsilon}}\right)$ |
| Xavier [27] | $\frac{2}{m_{l-1}+m_l}$ | $\frac{od(L-1+\frac{d+o}{2m})}{2^{L-3}(1+\frac{d}{m})(1+\frac{o}{m})}$ | $\tilde{\mathcal{O}}\left(\frac{n}{d}\cdot\frac{(1+\frac{d}{m})(1+\frac{1}{m})}{L-1+\frac{d+o}{2m}}\right)$ | $\tilde{\mathcal{O}}\left(\frac{1}{n^2}+\sqrt{\frac{1}{2^L\varepsilon}}\right)$ |

To answer this question, we circumvent the difficulties of analyzing gradient clipping, and instead *algorithmically* focus on analyzing privacy for the Langevin diffusion algorithm *without* gradient clipping nor Lipschitz assumption on loss function. [2] It avoids an artificial setting in DP-SGD [2] where a constant sensitivity constraint is enforced for each gradient update and thus makes the privacy bound insensitive to the network over-parameterization. *Theoretically*, we prove that the KL privacy loss for Langevin diffusion scales with the expected gradient difference between the training on any two worst-case neighboring datasets (Theorem 3.1). [3] By proving precise upper bounds on the expected $\ell_2$-norm of this gradient difference, we thus obtain KL privacy bounds for fully connected neural network (Lemma 3.2) and its linearized variant (Corollary 4.2) that changes with the network width, depth and per-layer variance for the initialization distribution. We summarized the details of our KL privacy bounds in Table 1, and highlight our key observations below.

- Width always worsen privacy, under all the considered initialization schemes. Meanwhile, the interplay between network depth and privacy is much more complex and crucially depends on which initialization scheme is used and how long the training time is.

- Regarding the specific initialization schemes, under small per-layer variance in initialization (e.g. in LeCun and Xavier), if the depth is large enough, our KL privacy bound for training fully connected network (with a small amount of time) as well as linearized network (with finite time) decays exponentially with increasing depth. To the best of our knowledge, this is the first time that an improvement of privacy bound under over-parameterization is observed.

We further perform numerical experiments (Section 5) on deep neural network trained via noisy gradient descent to validate our privacy analyses. Finally, we analyze the privacy utility trade-off for training linearized network, and prove that the excess empirical risk bound (given any fixed KL privacy budget) scales with a lazy training distance bound $R$ (i.e., how close is the initialization to a minimizer of the empirical risk) and a gradient norm constant $B$ throughout training (Corollary 6.4). By analyzing these two terms precisely, we prove that under certain initialization distributions (such as LeCun and Xavier), the privacy utility trade-off strictly improves with increasing depth for linearized network (Table 1). To our best knowledge, this is the first time that such a gain in privacy-utility trade-off due to over-parameterization (increasing depth) is shown. Meanwhile, prior results only prove (nearly) dimension-independent privacy utility trade-off for such linear models in the literature [45, 32, 37]. Our improvement demonstrates the unique benefits of our algorithmic framework and privacy-utility analysis in understanding the effect of over-parameterization.

---

[2] A key difference between this paper and existing privacy utility analysis of Langevin diffusion [26] is that we analyze in the absence of gradient clipping or Lipschitz assumption on loss function. Our results also readily extend to discretized noisy GD with constant step-size (as discussed in Appendix E).

[3] We focus on KL privacy loss because it is a more relaxed distinguishability notion than standard $(\varepsilon, \delta)$-DP, and therefore could be upper bounded even without gradient clipping. Moreover, KL divergence enables upper bound for the advantage (relative success) of various inference attacks, as studied in recent works [39, 28].

## 1.1 Related Works

***over-parameterization in DNNs and NTK***. Theoretical demonstration on the benefit of over-parameterization in DNNs occurs in global convergence [3, 21] and generalization [4, 16]. Under proper initialization, the training dynamics of over-parameterized DNNs can be described by a kernel function, termed as neural tangent kernel (NTK) [31], which stimulates a series of analysis in DNNs. Accordingly, over-parameterization has been demonstrated to be beneficial/harmful to several topics in deep learning, e.g., robustness [15, 54], covariate shift [50]. However, the relationship between over-parameterization and privacy (based on the differential privacy framework) remains largely an unsolved problem, as the training dynamics typically change [14] after adding new components in the privacy-preserving learning algorithm (such as DP-SGD [2]) to enforce privacy constraints.

***Membership inference privacy risk under over-parameterization***. A recent line of works [47, 48] investigates how over-parameterization affects the theoretical and empirical privacy in terms of membership inference advantage, and proves novel trade-off between privacy and generalization error. These literature are closet to our objective of investigating the interplay between privacy and over-parameterization. However, Tan et al. [47, 48] focus on proving upper bounds for an average-case privacy risk defined by the advantage (relative success) of membership inference attack on models trained from randomly sampled training dataset from a population distribution. By contrast, our KL privacy bound is heavily based on the strongest adversary model in the differential privacy definition, and holds under an arbitrary *worst-case* pair of neighboring datasets, differing only in one record. Our model setting (e.g., fully connected neural networks) is also quite different from that of Tan et al. [47, 48]. The employed analysis tools are accordingly different.

***Differentially private learning in high dimension***. Standard results for private empirical risk minimization [9, 46] and private stochastic convex optimization [11, 12, 5] prove that there is an unavoidable factor $d$ in the empirical risk and population risk that depends on the model dimension. However, for unconstrained optimization, it is possible to seek for the dimension-dependency in proving risk bounds for certain class of problems (such as generalized linear model [45]). Recently, there is a growing line of works that proves dimension-independent excess risk bounds for differentially private learning, by utilizing the low-rank structure of data features [45] or gradient matrices [32, 37] during training. Several follow-up works [33, 13] further explore techniques to enforce the low-rank property (via random projection) and boost privacy utility trade-off. However, all the works focus on investigating a general high-dimensional problem for private learning, rather than separating the study for different network choices such as width, depth and initialization. Instead, our study focuses on the fully connected neural network and its linearized variant, which enables us to prove more precise privacy utility trade-off bounds for these particular networks under over-parameterization.

## 2  Problem and Methodology

We consider the following standard multi-class supervised learning setting. Let $\mathcal{D} = (\boldsymbol{z}_1, \cdots, \boldsymbol{z}_n)$ be an input dataset of size $n$, where each data record $\boldsymbol{z}_i = (\boldsymbol{x}_i, \boldsymbol{y}_i)$ contains a $d$-dimensional feature vector $\boldsymbol{x}_i \in \mathbb{R}^d$ and a label vector $\boldsymbol{y}_i \in \mathcal{Y} = \{-1, 1\}^o$ on $o$ classes. We aim to learn a neural network output function $\boldsymbol{f_W}(\cdot) : \mathcal{X} \to \mathcal{Y}$ parameterized by $\boldsymbol{W}$ via empirical risk minimization (ERM)

$$\min_{\boldsymbol{W}} \mathcal{L}(\boldsymbol{W}; \mathcal{D}) := \frac{1}{n} \sum_{i=1}^{n} \ell(\boldsymbol{f_W}(\boldsymbol{x}_i); \boldsymbol{y}_i), \qquad (1)$$

where $\ell(\boldsymbol{f_W}(\boldsymbol{x}_i); \boldsymbol{y}_i)$ is a loss function that reflects the approximation quality of model prediction $\boldsymbol{f_W}(\boldsymbol{x}_i)$ compared to the ground truth label $\boldsymbol{y}_i$. For simplicity, throughout our analysis, we employ the cross-entropy loss $\ell(\boldsymbol{f_W}(\boldsymbol{x}); \boldsymbol{y}) = -\langle \boldsymbol{y}, \log \mathrm{softmax}(\boldsymbol{f_W}(\boldsymbol{x})) \rangle$ for multi-class network with $o \geq 2$, and $\ell(\boldsymbol{f_W}(\boldsymbol{x}); \boldsymbol{y}) = \log(1 + \exp(-\boldsymbol{y}\boldsymbol{f_W}(\boldsymbol{x}))$ for single-output network with $o = 1$.

***Fully Connected Neural Networks***. We consider the $L$-layer, multi-output, fully connected, deep neural network (DNN) with ReLU activation. Denote the width of hidden layer $l$ as $m_l$ for $l = 1, \cdots, L - 1$. For consistency, we also denote $m_0 = d$ and $m_L = o$. The network output $\boldsymbol{f_W}(\boldsymbol{x}) := \boldsymbol{h}_L(\boldsymbol{x})$ is defined recursively as follows.

$$\boldsymbol{h}_0(\boldsymbol{x}) = \boldsymbol{x}; \quad \boldsymbol{h}_l(\boldsymbol{x}) = \phi(\boldsymbol{W}_l \boldsymbol{x}) \text{ for } l = 1, \cdots, L-1; \quad \boldsymbol{h}_L(\boldsymbol{x}) = \boldsymbol{W}_L \boldsymbol{h}_{L-1}(\boldsymbol{x}), \qquad (2)$$

where $h_l(\boldsymbol{x})$ denotes the post-activation output at $l$-th layer, and $\{\boldsymbol{W}_l \in \mathbb{R}^{m_l \times m_{l-1}} : l = 1, \ldots, L\}$ denotes the set of per-layer weight matrices of DNN. For brevity, we denote the vector $\boldsymbol{W} := (\text{Vec}(\boldsymbol{W}_1), \ldots, \text{Vec}(\boldsymbol{W}_L)) \in \mathbb{R}^{m_1 \cdot d + m_2 \cdot m_1 + \cdots + o \cdot m_{L-1}}$, i.e., the the concatenation of vectorizations for weight matrices of all layers, as the model parameter.

***Linearized Network***. We also analyze the following ***linearized network***, which is used in prior works [35, 3, 41] as an important tool to (approximately and qualitatively) analyze the training dynamics of DNNs. Formally, the linearized network $\boldsymbol{f}_{\boldsymbol{W}}^{lin,0}(\boldsymbol{x})$ is a first-order Taylor expansion of the fully connected ReLU network at initialization parameter $\boldsymbol{W}_0^{lin}$, as follows.

$$\boldsymbol{f}_{\boldsymbol{W}}^{lin,0}(\boldsymbol{x}) \equiv \boldsymbol{f}_{\boldsymbol{W}_0^{lin}}(\boldsymbol{x}) + \frac{\partial \boldsymbol{f}_{\boldsymbol{W}}(\boldsymbol{x})}{\partial \boldsymbol{W}}\Big|_{\boldsymbol{W}=\boldsymbol{W}_0^{lin}} \left(\boldsymbol{W} - \boldsymbol{W}_0^{lin}\right), \tag{3}$$

where $\boldsymbol{f}_{\boldsymbol{W}_0^{lin}}(\boldsymbol{x})$ is the output function of the fully connected ReLU network (2) at initialization $\boldsymbol{W}_0^{lin}$. We denote $\mathcal{L}_0^{lin}(\boldsymbol{W}; \mathcal{D}) = \frac{1}{n}\sum_{i=1}^{n} \ell\left(\boldsymbol{f}_{\boldsymbol{W}_0^{lin}}(\boldsymbol{x}_i) + \frac{\partial \boldsymbol{f}_{\boldsymbol{W}}(\boldsymbol{x})}{\partial \boldsymbol{W}}|_{\boldsymbol{W}=\boldsymbol{W}_0^{lin}}(\boldsymbol{W} - \boldsymbol{W}_0^{lin}); \boldsymbol{y}_i\right)$ as the empirical loss function for training linearized network, by plugging (3) into (1).

***Langevin Diffusion***. Regarding the optimization algorithm, we focus on the *Langevin diffusion* algorithm [36] with per-dimensional noise variance $\sigma^2$. Note that we aim to *avoid gradient clipping* while still proving KL privacy bounds. After initializing the model parameters $\boldsymbol{W}_0$ at time zero, the model parameters $\boldsymbol{W}_t$ at subsequent time $t$ evolves as the following stochastic differential equation.

$$\mathrm{d}\boldsymbol{W}_t = -\nabla\mathcal{L}(\boldsymbol{W}_t; \mathcal{D})\mathrm{d}t + \sqrt{2\sigma^2}\mathrm{d}\boldsymbol{B}_t. \tag{4}$$

***Initialization Distribution***. The initialization of parameters $\boldsymbol{W}_0$ crucially affects the convergence of Langevin diffusion, as observed in prior literatures [52, 25, 24]. In this work, we investigate the following general class of Gaussian initialization distributions with different (possibly depth-dependent) variances for the parameters in each layer. For any layer $l = 1, \cdots, L$, we have

$$[\boldsymbol{W}^l]_{ij} \sim \mathcal{N}(0, \beta_l), \text{ for } (i,j) \in [m_l] \times [m_{l-1}], \tag{5}$$

where $\beta_1, \cdots, \beta_L > 0$ are the per-layer variance for Gaussian initialization. By choosing different variances, we recover many common initialization schemes in the literature, as summarized in Table 1.

## 2.1 Our objective and methodology

We aim to understand the relation between privacy, utility and over-parameterization (depth and width) for the Langevin diffusion algorithm (under different initialization distributions). For privacy analysis, we prove a KL privacy bound for running Langevin diffusion on any two *worst-case* neighboring datasets. Below we first give the definition for neighboring datasets.

**Definition 2.1.** We denote $\mathcal{D}$ and $\mathcal{D}'$ as neighboring datasets if they are of same size and only differ in one record. For brevity, we also denote the differing records as $(\boldsymbol{x}, \boldsymbol{y}) \in \mathcal{D}$ and $(\boldsymbol{x}', \boldsymbol{y}') \in \mathcal{D}'$.

**Assumption 2.2** (Bounded Data). For simplicity, we assume bounded data, i.e., $\|\boldsymbol{x}\|_2 \leq \sqrt{d}$.

We now give the definition for KL privacy, which is a more relaxed, yet closely connected privacy notion to the standard $(\varepsilon, \delta)$ differential privacy [22], see Appendix A.2 for more discussions. KL privacy and its relaxed variants are commonly used in previous literature [8, 10, 53].

**Definition 2.3** (KL privacy). A randomized algorithm $\mathcal{A}$ satisfies $\varepsilon$-KL privacy if for any neighboring datasets $\mathcal{D}$ and $\mathcal{D}'$, we have that the KL divergence $\text{KL}(\mathcal{A}(\mathcal{D})\|\mathcal{A}(\mathcal{D}')) \leq \varepsilon$, where $\mathcal{A}(\mathcal{D})$ denotes the algorithm's output distribution on dataset $\mathcal{D}$.

In this paper, we prove KL privacy upper bound for $\max_{\mathcal{D},\mathcal{D}'} \text{KL}(\boldsymbol{W}_{[0:T]}\|\boldsymbol{W}'_{[0:T]})$ when running Langevin diffusion on any *worst-case* neighboring datasets. For brevity, here (and in the remaining paper), we abuse the notations and denote $\boldsymbol{W}_{[0:T]}$ and $\boldsymbol{W}'_{[0:T]}$ as the distributions of model parameters trajectory during Langevin diffusion processes Eq. (4) with time $T$ on $\mathcal{D}$ and $\mathcal{D}'$ respectively.

For utility analysis, we prove the upper bound for the excess empirical risk given any fixed KL divergence privacy budget for a single-output neural network under the following additional assumption (it is only required for utility analysis and not needed for our privacy bound).

**Assumption 2.4** ([40, 20, 21]). The training data $\boldsymbol{x}_1, \cdots, \boldsymbol{x}_n$ are i.i.d. samples from a distribution $P_x$ that satisfies $\mathbb{E}[\boldsymbol{x}] = 0, \|\boldsymbol{x}\|_2 = \sqrt{d}$ for $\boldsymbol{x} \sim P_x$, and with probability one for any $i \neq j$, $\boldsymbol{x}_i \nparallel \boldsymbol{x}_j$.

Our ultimate goal is to precisely understand how the excess empirical risk bounds (given fixed KL privacy budget) are affected by increasing width and depth under different initialization distributions.

## 3   KL Privacy for Training Fully Connected ReLU Neural Networks

In this section, we perform the composition-based KL privacy analysis for Langevin Diffusion given random Gaussian initialization distribution under Eq. (5) for fully connected ReLU network. More specifically, we prove upper bound for the KL divergence between distribution of output model parameters when running Langevin diffusion on an arbitrary pair of neighboring datasets $\mathcal{D}$ and $\mathcal{D}'$.

Our first insight is that by a Bayes rule decomposition for density function, KL privacy under a relaxed gradient sensitivity condition can be proved (that could hold *without* gradient clipping).

**Theorem 3.1** (KL composition under possibly unbounded gradient difference). *The KL divergence between running Langevin diffusion* (4) *for DNN* (2) *on neighboring datasets $\mathcal{D}$ and $\mathcal{D}'$ satisfies*

$$\mathrm{KL}(\boldsymbol{W}_{[0:T]} \| \boldsymbol{W}'_{[0:T]}) = \frac{1}{2\sigma^2} \int_0^T \mathbb{E}\left[\|\nabla\mathcal{L}(\boldsymbol{W}_t; \mathcal{D}) - \nabla\mathcal{L}(\boldsymbol{W}_t; \mathcal{D}')\|_2^2\right] \mathrm{d}t. \tag{6}$$

***Proof sketch***. We compute the partial derivative of KL divergence with regard to time $t$, and then integrate it over $t \in [0, T]$ to compute the KL divergence during training with time $T$. For computing the limit of differentiation, we use Girsanov's theorem to compute the KL divergence between the trajectory of Langevin diffusion processes on $\mathcal{D}$ and $\mathcal{D}'$. The complete proof is in Appendix B.1.

Theorem 3.1 is an extension of the standard additivity [51] of KL divergence (also known as chain rule [1]) for a finite sequence of distributions to continuous time processes with (possibly) unbounded drift difference. The key extension is that Theorem 3.1 does not require bounded sensitivity between the drifts of Langevin Diffusion on neighboring datasets. Instead, it only requires finite second-order moment of drift difference (in the $\ell_2$-norm sense) between neighboring datasets $\mathcal{D}, \mathcal{D}'$, which can be proved by the following Lemma. We prove that this expectation of squared gradient difference incurs closed-form upper bound under deep neural network (under mild assumptions), for running Langevin diffusion (without gradient clipping) on any neighboring dataset $\mathcal{D}$ and $\mathcal{D}'$.

**Lemma 3.2** (Drift Difference in Noisy Training). *Let $M_T$ be the subspace spanned by gradients $\{\nabla\ell(\boldsymbol{f}_{\boldsymbol{W}_t}(\boldsymbol{x}_i; \boldsymbol{y}_i) : (\boldsymbol{x}_i, \boldsymbol{y}_i) \in \mathcal{D}, t \in [0, T]\}_{i=1}^n$ throughout Langevin diffusion $(\boldsymbol{W}_t)_{t \in [0,T]}$. Denote $\|\cdot\|_{M_T}$ as the $\ell_2$ norm of the projected input vector onto $M_T$. Suppose that there exists constants $c, \beta > 0$ such that for any $\boldsymbol{W}, \boldsymbol{W}'$ and $(\boldsymbol{x}, \boldsymbol{y})$, we have $\|\nabla\ell(f_{\boldsymbol{W}}(\boldsymbol{x}); \boldsymbol{y})) - \nabla\ell(f_{\boldsymbol{W}'}(\boldsymbol{x}); \boldsymbol{y})\|_2 < \max\{c, \beta\|\boldsymbol{W} - \boldsymbol{W}'\|_{M_T}\}$. Then running Langevin diffusion Eq.* (4) *with Gaussian initialization distribution* (5) *satisfies $\varepsilon$-KL privacy with $\varepsilon = \frac{\max_{\mathcal{D}, \mathcal{D}'} \int_0^T \mathbb{E}\left[\|\nabla\mathcal{L}(\boldsymbol{W}_t; \mathcal{D}) - \nabla\mathcal{L}(\boldsymbol{W}_t; \mathcal{D}')\|_2^2\right] \mathrm{d}t}{2\sigma^2}$ where*

$$\int_0^T \mathbb{E}\left[\|\nabla\mathcal{L}(\boldsymbol{W}_t; \mathcal{D}) - \nabla\mathcal{L}(\boldsymbol{W}_t; \mathcal{D}')\|_2^2\right] \mathrm{d}t \leq 2T \cdot \underbrace{\mathbb{E}\left[\|\nabla\mathcal{L}(\boldsymbol{W}_0; \mathcal{D}) - \nabla\mathcal{L}(\boldsymbol{W}_0; \mathcal{D}')\|_2^2\right]}_{\text{gradient difference at initialization}}$$

$$+ \underbrace{\frac{2\beta^2}{n^2(2+\beta^2)}\left(\frac{e^{(2+\beta^2)T}-1}{2+\beta^2} - T\right) \cdot \left(\mathbb{E}\left[\|\nabla\mathcal{L}(\boldsymbol{W}_0; \mathcal{D})\|_2^2\right] + 2\sigma^2 rank(M_T) + c^2\right)}_{\text{gradient difference fluctuation during training}} + \underbrace{\frac{2c^2T}{n^2}}_{\text{non-smoothness}}.$$

***Proof sketch***. The key is to reduce the problem of upper bounding the gradient difference at any training time $T$, to analyzing its two subcomponents: $\|\nabla\ell(f_{\boldsymbol{W}_t}(\boldsymbol{x}); \boldsymbol{y})) - \nabla\ell(f_{\boldsymbol{W}_t}(\boldsymbol{x}'); \boldsymbol{y}')\|_2^2 \leq \underbrace{2\|\nabla\ell(f_{\boldsymbol{W}_0}(\boldsymbol{x}); \boldsymbol{y})) - \nabla\ell(f_{\boldsymbol{W}_0}(\boldsymbol{x}'); \boldsymbol{y}')\|_2^2}_{\text{gradient difference at initialization}} + 2\beta^2 \underbrace{\|\boldsymbol{W}_t - \boldsymbol{W}_0\|_{M_T}^2}_{\text{parameters' change after time } T} + 2c^2$, where $(\boldsymbol{x}, \boldsymbol{y})$ and $(\boldsymbol{x}', \boldsymbol{y}')$ are the differing data between neighboring datasets $\mathcal{D}$ and $\mathcal{D}'$. This inequality is by the Cauchy-Schwartz inequality. In this way, the second term in Lemma 3.2 uses the change of parameters

to bound the gradient difference between datasets $\mathcal{D}$ and $\mathcal{D}'$ at time $T$, via the relaxed smoothness assumption of loss function (that is explained in Remark 3.5 in details). The complete proof is in Appendix B.2.

*Remark* 3.3 (Gradient difference at initialization). The first term and in our upper bound linearly scales with the difference between gradients on neighboring datasets $\mathcal{D}$ and $\mathcal{D}'$ at initialization. Under different initialization schemes, this gradient difference exhibits different dependency on the network depth and width, as we will prove theoretically in Theorem 4.1.

*Remark* 3.4 (Gradient difference fluctuation during training). The second term in Lemma 3.2 bounds the change of gradient difference during training, and is proportional to the the rank of a subspace $M_T$ spanned by gradients of all training data. Intuitively, this fluctuation is because Langevin diffusion adds per-dimensional noise with variance $\sigma^2$, thus perturbing the training parameters away from the initialization at a scale of $O(\sigma\sqrt{\text{rank}(M_T)})$ in the expected $\ell_2$ distance.

*Remark* 3.5 (Relaxed smoothness of loss function). The third term in Lemma 3.2 is due to the assumption $\|\nabla\ell(f_{\boldsymbol{W}}(\boldsymbol{x});\boldsymbol{y})) - \nabla\ell(f_{\boldsymbol{W}'}(\boldsymbol{x});\boldsymbol{y})\|_2 < \max\{c, \beta\|\boldsymbol{W} - \boldsymbol{W}'\|_{M_T}\}$. This assumption is similar to smoothness of loss function, but is more relaxed as it allows non-smoothness at places where the gradient is bounded by $c$. Therefore, this assumption is general to cover commonly-used smooth, non-smooth activation functions, e.g., sigmoid, ReLU.

***Growth of KL privacy bound with increasing training time*** $T$. The first and third terms in our upper bound Lemma 3.2 grow linearly with the training time $T$, while the second term grows exponentially with regard to $T$. Consequently, for learning tasks that requires a long training time to converge, the second term will become the dominating term and the KL privacy bound suffers from exponential growth with regard to the training time. Nevertheless, observe that for small $T \to 0$, the second component in Lemma 3.2 contains a small factor $\frac{e^{(2+\beta^2)T}-1}{2+\beta^2} - T = o(T)$ by Taylor expansion.Therefore, for small training time, the second component is smaller than the first and the third components in Lemma 3.2 that linearly scale with $T$, and thus does not dominate the privacy bound. Intuitively, this phenomenon is related to lazy training [19]. In Section 5 and Figure 2, we also numerically validate that the second component does not have a high effect on the KL privacy loss in the case of small training time.

***Dependence of KL privacy bound on network over-parameterization***. Under a fixed training time $T$ and noise scale $\sigma^2$, Lemma 3.2 predicts that the KL divergence upper bound in Theorem 3.1 is dependent on the gradient difference and gradient norm at initialization, and the rank of gradient subspace $\text{rank}(M_T)$ throughout training. We now discuss the how these two terms change under increasing width and depth, and whether there are possibilities to improve them under over-parameterization.

1. The gradient norm at initialization crucially depends on how the per-layer variance in the Gaussian initialization distribution scales with the network width and depth. Therefore, it is possible to reduce the gradient difference at initialization (and thus improve the KL privacy bound) by using specific initialization schemes, as we later show in Section 4 and Section 5.

2. Regarding the rank of gradient subspace $\text{rank}(M_T)$: when the gradients along the training trajectory span the whole optimization space, $\text{rank}(M_T)$ would equal the dimension of the learning problem. Consequently, the gradient fluctuation upper bound (and thus the KL privacy bound) worsens with increasing number of model parameters (over-parameterization) in the worst-case. However, if the gradients are low-dimensional [45, 32, 43] or sparse [37], $\text{rank}(M_T)$ could be dimension-independent and thus enables better bound for gradient fluctuation (and KL privacy bound). We leave this as an interesting open problem.

## 4 KL privacy bound for Linearized Network under over-parameterization

In this section, we focus on the training of linearized networks (3), which fosters a refined analysis on the interplay between KL privacy and over-parameterization (increasing width and depth). Analysis of DNNs via linearization is a commonly used technique in both theory [19] and practice [43, 41]. We hope our analysis for linearized network serves as an initial attempt that would open a door to theoretically understanding the relationship between over-parameterization and privacy.

To derive a composition-based KL privacy bound for training a linearized network, we apply Theorem 3.1 which requires an upper bound for the norm of gradient difference between the training

processes on neighboring datasets $\mathcal{D}$ and $\mathcal{D}'$ at any time $t$. Note that the empirical risk function for training linearized models enjoys convexity, and thus a relatively short amount of training time is enough for convergence. In this case, intuitively, the gradient difference between neighboring datasets does not change a lot during training, which allows for a tighter upper bound for the gradient difference norm for linearized networks (than Lemma 3.2).

In the following theorem, we prove that for a linearized network, the gradient difference throughout training has a uniform upper bound that only depends on the network width, depth and initialization.

**Theorem 4.1** (Gradient Difference throughout training linearized network). *Under Assumption 2.2, taking over the randomness of the random initialization and the Brownian motion, for any $t \in [0, T]$, running Langevin diffusion on a linearized network in Eq. (3) satisfies that*

$$\mathbb{E}\left[\|\nabla\mathcal{L}(\boldsymbol{W}_t^{lin};\mathcal{D}) - \mathcal{L}(\boldsymbol{W}_t^{lin};\mathcal{D}')\|_2^2\right] \leq \frac{4B}{n^2}, \text{ where } B := d \cdot o \cdot \left(\prod_{i=1}^{L-1}\frac{\beta_i m_i}{2}\right)\sum_{l=1}^{L}\frac{\beta_L}{\beta_l}, \quad (7)$$

*where $n$ is the training dataset size, and $B$ is a constant that only depends on the data dimension $d$, the number of classes $o$, the network depth $L$, the per-layer network width $\{m_i\}_{i=1}^L$, and the per-layer variances $\{\beta_i\}_{i=1}^L$ of the Gaussian initialization distribution.*

Theorem 4.1 provides a precise analytical upper bound for the gradient difference during training linearized network, by tracking the gradient distribution for fully connected feed-forward ReLU network with Gaussian weight matrices. Our proof borrows some techniques from [3, 54] for computing the gradient distribution, refer to Appendix C.1 and C.2 for the full proofs. By plugging Eq. (7) into Theorem 3.1, we obtain the following KL privacy bound for training a linearized network.

**Corollary 4.2** (KL privacy bound for training linearized network). *Under Assumption 2.2 and neural networks (3) initialized by Gaussian distribution with per-layer variance $\{\beta_i\}_{i=1}^L$, running Langevin diffusion for linearized network with time $T$ on any neighboring datasets satisfies that*

$$\text{KL}(\boldsymbol{W}_{[0:T]}^{lin}\|\boldsymbol{W}'^{lin}_{[0:T]}) \leq \frac{2BT}{n^2\sigma^2}, \quad (8)$$

*where $B$ is the constant that specifies the gradient norm upper bound, given by Eq. (7).*

***Over-parameterization affects privacy differently under different initialization***. Corollary 4.2 and Theorem 4.1 prove the role of over-parameterization in our KL privacy bound, crucially depending on how the per-layer Gaussian initialization variance $\beta_i$ scales with the per-layer network width $m_i$ and depth $L$. We summarize our KL privacy bound for the linearized network under different width, depth and initialization schemes in Table 1, and elaborate the comparison below.

**(1) LeCun initialization** uses small, width-independent variance for initializing the first layer $\beta_1 = \frac{1}{d}$ (where $d$ is the number of input features), and width-dependent variance $\beta_2 = \cdots = \beta_L = \frac{1}{m}$ for initializing all the subsequent layers. Therefore, the second term $\sum_{l=1}^{L}\frac{\beta_L}{\beta_l}$ in the constant $B$ of Eq. (7) increases linearly with the width $m$ and depth $L$. However, due to $\frac{m_l \cdot \beta_l}{2} < 1$ for all $l = 2, \cdots, L$, the first product term $\prod_{l=1}^{L-1}\frac{\beta_l m_l}{2}$ in constant $B$ decays with the increasing depth. Therefore, by combining the two terms, we prove that the KL privacy bound worsens with increasing width, but improves with increasing depth (as long as the depth is large enough). Similarly, under **Xavier initialization** $\beta_l = \frac{2}{m_{l-1}+m_l}$, we prove that the KL privacy bound (especially the constant $B$ (7)) improves with increasing depth as long as the depth is large enough.

**(2) NTK and He initializations** use large per-layer variance $\beta_l = \begin{cases} \frac{2}{m_l} & l = 1, \cdots, L-1 \\ \frac{1}{o} & l = L \end{cases}$ (for NTK) and $\beta_l = \frac{2}{m_{l-1}}$ (for He). Consequently, the gradient difference under NTK or He initialization is significantly larger than that under LeCun initialization. Specifically, the gradient norm constant $B$ in Eq. (7) grows linearly with the width $m$ and the depth $L$ under He and NTK initializations, thus indicating a worsening of KL privacy bound under increasing width and depth.

## 5  Numerical validation of our KL privacy bounds

To understand the relation between privacy and over-parameterization in *practical* DNNs training (and to validate our KL privacy bounds Lemma 3.2 and Corollary 4.2), we perform experiments for

DNNs training via noisy GD to numerically estimate the KL privacy loss. We will show that if the total training time is small, it is indeed possible to obtain numerical KL privacy bound estimates that does not grow with the total number of parameter (under carefully chosen initialization distributions).

***Numerical estimation procedure***. Theorem 3.1 proves that the exact KL privacy loss scales with the expectation of squared gradient norm during training. This could be estimated by empirically average of gradient norm across training runs. For training dataset $\mathcal{D}$, we consider all 'car' and 'plane' images of the CIFAR-10. For neighboring dataset, we consider all possible $\mathcal{D}'$ that removes a record from $\mathcal{D}$, or adds a test record to $\mathcal{D}$, i.e., the standard "add-or remove-one" neighboring notion [2]. We run noisy gradient descent with constant step-size $0.01$ for $50$ epochs on both datasets.

***Numerically validate the growth of KL privacy loss with regard to training time***. Figure 1 shows numerical KL privacy loss under different initializations, for fully connected networks with width $1024$ and depth $10$. We observe that the KL privacy loss grows linearly at the beginning of training ($< 10$ epochs), which validates the first and third term in the KL privacy bound Lemma 3.2. Moreover, the KL privacy loss under LeCun and Xavier initialization is close to zero at the beginning of training ($< 10$ epochs). This shows LeCun and Xavier initialization induce small gradient norm at small training time, which is consistent with Theorem 4.1. However, when the number of epochs is large, the numerical KL privacy loss grows faster than linear accumulation under all initializations, thus validating the second term in Lemma 3.2.

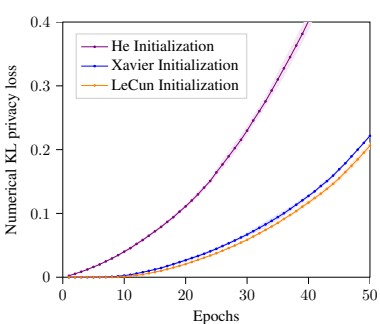

Figure 1: Numerically estimated KL privacy loss for noisy GD with constant step-size $0.001$ on deep neural network with width $1024$ and depth $10$. We report the mean and standard deviation across 6 training runs, taking worst-case over all neighboring datasets. The numerical KL privacy loss grows with the number of training epochs under all initializations. The growth rate is close to linear at beginning of training (epochs $< 10$) and is faster than linear at epochs $\geq 10$.

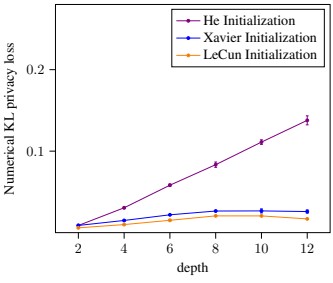
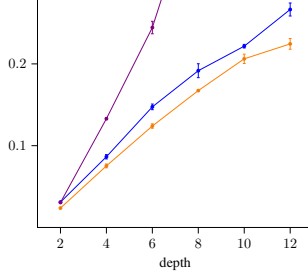
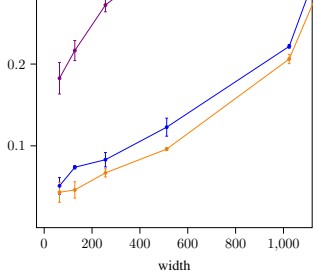

(a) Privacy vs depth (20 epochs, $\sigma = 0.01$, width = 1024)

(b) Privacy vs depth (50 epochs, $\sigma = 0.01$, width = 1024)

(c) Privacy vs width (50 epochs, $\sigma = 0.01$, depth = 10)

Figure 2: Numerically estimated KL privacy loss for noisy GD with constant step-size on fully connected ReLU network with different width, depth and initializations. We report the mean and standard deviation across 6 training runs, taking worst-case over all neighboring datasets. Under increasing width, the KL privacy loss always grows under all evaluated initializations. Under increasing depth, at the beginning of training (20 epochs), the KL privacy loss worsens with depth under He initialization, but first worsens with depth ($\leq 8$) and then improves with depth ($\geq 8$) under Xavier and LeCun initializations. At later phases of the training (50 epochs), KL privacy worsens (increases) with depth under all evaluated initializations.

***Numerically validate the dependency of KL privacy loss on network width, depth and initializations***. Figure 2 shows the numerical KL privacy loss under different network depth, width and initializations, for a fixed training time. In Figure 2c, we observe that increasing width and training time always increases KL privacy loss. This is consistent with Theorem 4.1, which shows that increasing width worsens the gradient norm at initialization (given fixed depth), thus harming KL privacy bound Lemma 3.2 at the beginning of training. We also observe that the relationship between KL privacy

and network depth depends on the initialization distributions and the training time. Specifically, in Figure 2a, when the training time is small (20 epochs), for LeCun and Xavier initializations, the numerical KL privacy loss improves with increasing depth when depth $> 8$. Meanwhile, when the training time is large (50 epochs) in Figure 2b, KL privacy loss worsens with increasing depth under all intializations. This shows that given small training time, the choice of initialization distribution affects the dependency of KL privacy loss on increasing depth, thus validating Lemma 3.2 and Theorem 4.1.

## 6 Utility guarantees for Training Linearized Network

Our privacy analysis suggests that training linearized network under certain initialization schemes (such as LeCun initialization) allows for significantly better privacy bounds under over-parameterization by increasing depth. In this section, we further prove utility bounds for Langevin diffusion under initialization schemes and investigate the effect of over-parameterization on the privacy utility trade-off. In other words, we aim to understand whether there is any utility degradation for training linearized networks when using the more privacy-preserving initialization schemes.

***Convergence of training linearized network***.  We now prove convergence of the excess empirical risk in training linearized network via Langevin diffusion. This is a well-studied problem in the literature for noisy gradient descent. We extend the convergence theorem to continuous-time Langevin diffusion below and investigate factors that affect the convergence under over-parameterization. The proof is deferred to Appendix D.1.

**Lemma 6.1** (Extension of [42, Theorem 2] and [45, Theorem 3.1])**.** *Let $\mathcal{L}_0^{lin}(\boldsymbol{W}; \mathcal{D})$ be the empirical risk function of a linearized network in Eq. (3) expanded at initialization vector $\boldsymbol{W}_0^{lin}$. Let $\boldsymbol{W}_0^*$ be an $\alpha$-near-optimal solution for the ERM problem such that $\mathcal{L}_0^{lin}(\boldsymbol{W}_0^*; \mathcal{D}) - \min_{\boldsymbol{W}} \mathcal{L}_0^{lin}(\boldsymbol{W}; \mathcal{D}) \leq \alpha$. Let $\mathcal{D} = \{\boldsymbol{x}_i\}_{i=1}^n$ be an arbitrary training dataset of size $n$, and denote $M_0 = \left(\nabla \boldsymbol{f}_{\boldsymbol{W}_0^{lin}}(\boldsymbol{x}_1), \cdots, \nabla \boldsymbol{f}_{\boldsymbol{W}_0^{lin}}(\boldsymbol{x}_n)\right)^\top$ as the NTK feature matrix at initialization. Then running Langevin diffusion (4) on $\mathcal{L}_0^{lin}(\boldsymbol{W})$ with time $T$ and initialization vector $\boldsymbol{W}_0^{lin}$ satisfies*

$$\mathbb{E}[\mathcal{L}_0^{lin}(\bar{\boldsymbol{W}}_T^{lin})] - \min_{\boldsymbol{W}} \mathcal{L}_0^{lin}(\boldsymbol{W}; \mathcal{D}) \leq \alpha + \frac{R}{2T} + \frac{1}{2}\sigma^2 rank(M_0),$$

*where the expectation is over Brownian motion $B_T$ in Langevin diffusion in Eq. (4), $\bar{\boldsymbol{W}}_T^{lin} = \frac{1}{T}\int \bar{\boldsymbol{W}}_t^{lin}\mathrm{d}t$ is the average of all iterates, and $R = \|\boldsymbol{W}_0^{lin} - \boldsymbol{W}_0^*\|_{M_0}^2$ is the gap between initialization parameters $\boldsymbol{W}_0^{lin}$ and solution $\boldsymbol{W}_0^*$.*

*Remark* 6.2.  The excess empirical risk bound in Lemma 6.1 is smaller if data is low-rank, e.g., image data, then rank$(M_0)$ is small. This is consistent with the prior dimension-independent private learning literature [32, 33, 37] and shows the benefit of low-dimensional gradients on private learning.

Lemma 6.1 highlights that the excess empirical risk scales with the gap $R$ between initialization and solution (denoted as lazy training distance), the rank of the gradient subspace, and the constant $B$ that specifies upper bound for expected gradient norm during training. Specifically, the smaller the lazy training distance $R$ is, the better is the excess risk bound given fixed training time $T$ and noise variance $\sigma^2$. We have discussed how over-parameterization affects the gradient norm constant $B$ and the gradient subspace rank rank$(M_0)$ in Section 3. Therefore, we only still need to investigate how the lazy training distance $R$ changes with the network width, depth, and initialization, as follows.

***Lazy training distance $R$ decreases with model over-parameterization***.  It is widely observed in the literature [19, 55, 38] that under appropriate choices of initializations, gradient descent on fully connected neural network falls under a lazy training regime. That is, with high probability, there exists a (nearly) optimal solution for the ERM problem that is close to the initialization parameters in terms of $\ell_2$ distance. Moreover, this lazy training distance $R$ is closely related to the smallest eigenvalue of the NTK matrix, and generally decreases as the model becomes increasingly overparameterized. In the following proposition, we compute a near-optimal solution via the pseudo inverse of the NTK matrix, and prove that it has small distance to the initialization parameters via existing lower bounds for the smallest eigenvalue of the NTK matrix [40].

**Lemma 6.3** (Bounding lazy training distance via smallest eigenvalue of the NTK matrix)**.** *Under Assumption 2.4 and single-output linearized network Eq. (3) with $o = 1$, assume that the per-layer network widths $m_0, \cdots, m_L = \tilde{\Omega}(n)$ are large. Let $\mathcal{L}_0^{lin}(\boldsymbol{W})$ be the empirical risk Eq. (1) for*

*linearized network expanded at initialization vector $\boldsymbol{W}_0^{lin}$. Then for any $\boldsymbol{W}_0^{lin}$, there exists a corresponding solution $\boldsymbol{W}_0^{\frac{1}{n^2}}$, s.t. $\mathcal{L}_0^{lin}(\boldsymbol{W}_0^{\frac{1}{n^2}}) - \min_{\boldsymbol{W}} \mathcal{L}_0^{lin}(\boldsymbol{W}; \mathcal{D}) \leq \frac{1}{n^2}$, $rank(M_0) = n$ and*

$$R \leq \tilde{\mathcal{O}}\left(\max\left\{\frac{1}{d\beta_L\left(\prod_{i=1}^{L-1}\beta_i m_i\right)}, 1\right\}\frac{n}{\sum_{l=1}^{L}\beta_l^{-1}}\right), \tag{9}$$

*with high probability over training data sampling and random initialization Eq. (5), where $\tilde{\mathcal{O}}$ ignores logarithmic factors with regard to $n$, $m$, $L$, and tail probability $\delta$.*

The full proof is deferred to Appendix D.2. By using Lemma 6.3, we provide a summary of bounds for $R$ under different initializations in Table 1. We observe that the lazy training distance $R$ decreases with increasing width and depth under LeCun, He and NTK initializations, while under Xavier initialization $R$ only decreases with increasing depth.

***Privacy & Excess empirical risk tradeoffs for Langevin diffusion under linearized network***. We now use the lazy training distance $R$ to prove empirical risk bound and combine it with our KL privacy bound Section 4 to show the privacy utility trade-off under over-parameterization.

**Corollary 6.4** (Privacy utility trade-off for linearized network). *Assume that all conditions in Lemma 6.3 holds. Let $B$ be the gradient norm constant in Eq. (7), and let $R$ be the lazy training distance bound in Lemma 6.3. Then for $\sigma^2 = \frac{2BT}{\varepsilon n^2}$ and $T = \sqrt{\frac{\varepsilon n R}{2B}}$, releasing all iterates of Langevin diffusion with time $T$ satisfies $\varepsilon$-KL privacy, and has empirical excess risk upper bounded by*

$$\mathbb{E}[\mathcal{L}_0^{lin}(\bar{\boldsymbol{W}}_T^{lin})] - \min_{\boldsymbol{W}} \mathcal{L}_0^{lin}(\boldsymbol{W}; \mathcal{D}) \leq \tilde{\mathcal{O}}\left(\frac{1}{n^2} + \sqrt{\frac{BR}{\varepsilon n}}\right) \tag{10}$$

$$= \tilde{\mathcal{O}}\left(\frac{1}{n^2} + \sqrt{\frac{\max\{1, d\beta_L\prod_{l=1}^{L-1}\beta_l m_l\}}{2^{L-1}\varepsilon}}\right) \tag{11}$$

*with high probability over random initiailization Eq. (5), where the expectation is over Brownian motion $B_T$ in Langevin diffusion, and $\tilde{\mathcal{O}}$ ignores logarithmic factors with regard to width $m$, depth $L$, number of training data $n$ and tail probability $\delta$.*

See Appendix D.3 for the full proof. Corollary 6.4 proves that the excess empirical risk worsens in the presence of a stronger privacy constraint, i.e., a small privacy budget $\varepsilon$, thus contributing to a trade-off between privacy and utility. However, the excess empirical risk also scales with the lazy training distance $R$ and the gradient norm constant $B$. These constants depend on network width, depth and initialization distributions, and we prove privacy utility trade-offs for training linearized network under commonly used initialization distributions, as summarized in Table 1.

We would like to highlight that our privacy utility trade-off bound under LeCun and Xavier initialization strictly improves with increasing depth as long as the data satisfy Assumption 2.4 and the hidden-layer width is large enough. To our best knowledge, this is the first time that a strictly improving privacy utility trade-off under over-parameterization is shown in literature. This shows the benefits of precisely bounding the gradient norm (Appendix C.1) in our privacy and utility analysis.

## 7 Conclusion

We prove new KL privacy bound for training fully connected ReLU network (and its linearized variant) using the Langevin diffusion algorithm, and investigate how privacy is affected by the network width, depth and initialization. Our results suggest that there is a complex interplay between privacy and over-parameterization (width and depth) that crucially relies on what initialization distribution is used and the how much the gradient fluctuates during training. Moreover, for a linearized variant of fully connected network, we prove KL privacy bounds that improve with increasing depth under certain initialization distributions (such as LeCun and Xavier). We further prove excess empirical risk bounds for linearized network under KL privacy, which similarly improve as depth increases under LeCun and Xavier initialization. This shows the gain of our new privacy analysis for capturing the effect of over-parameterization. We leave it as an important open problem as to whether our privacy utility trade-off results for linearized network could be generalized to deep neural networks.

## Acknowledgments and Disclosure of Funding

The authors would like to thank Yaxi Hu and anonymous reviewers for helpful discussions on drafts of this paper. This work was supported by Hasler Foundation Program: Hasler Responsible AI (project number 21043), and the Swiss National Science Foundation (SNSF) under grant number 200021_205011, Google PDPO faculty research award, Intel within the www.private-ai.org center, Meta faculty research award, the NUS Early Career Research Award (NUS ECRA award number NUS ECRA FY19 P16), and the National Research Foundation, Singapore under its Strategic Capability Research Centres Funding Initiative. Any opinions, findings and conclusions or recommendations expressed in this material are those of the author(s) and do not reflect the views of National Research Foundation, Singapore.

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

## Contents

## A   Symbols and definitions

### A.1   Additional notations

Vecorization $\mathrm{Vec}(\cdot)$ denotes the transformation that takes an input matrix $\boldsymbol{A} = (a_{ij})_{i \in [r], j \in [c]} \in \mathbb{R}^{r \times c}$ (with $r$ rows and $c$ columns) and outputs a $rc$-dimensional column vector: $\mathrm{Vec}(\boldsymbol{A}) = (a_{1,1}, \cdots, a_{r,1}, a_{1,2}, \cdots, a_{r,2}, \cdots, a_{1,c}, \cdots, a_{r,c})^{\top}$.

Softmax function: $\mathrm{softmax}(\boldsymbol{y}) = \frac{e^{\boldsymbol{y}^{[j]}}}{\sum_{j=1}^{o} e^{\boldsymbol{y}^{[j]}}}$ where $o$ is the number of output classes.

$o$: number of output classes for the neural network.

## A.2 Relation between KL privacy Definition 2.3 to differential privacy

KL privacy is a more relaxed, yet closely connected privacy notion to $(\varepsilon, \delta)$ differential privacy [22].

1. KL privacy and differential privacy are both worst-case privacy notions over all possible neighboring datasets, by requiring bounded distinguishability between the algorithm's output distributions on neighboring datasets in statistical divergence. The difference is that KL privacy requires bounded KL divergence, while $(\varepsilon, \delta)$-differential privacy is equivalent to bounded Hockey-stick divergence [6].

2. KL privacy and differential privacy are both definitions based on the privacy loss random variable $\log \frac{P(A(D)=o)}{P(A(D')=o)}, o \sim A(D)$ (following the definition in [2, Equation 1]). KL privacy implies that the privacy loss random variable has a bounded first order moment, while differential privacy requires a high probability argument that the privacy loss random variable is bounded by $\varepsilon$ with probability $1 - \delta$. Therefore, KL privacy is generally a more relaxed notion than differential privacy.

3. Translation to each other: For $\varepsilon = 0$, KL privacy (bounded first-order moment of privacy loss random variable) implies $(0, \delta)$-differential privacy with $\delta = \sqrt{\frac{\bar{\varepsilon}}{2}}$ by Pinsker inequality. Higher order moments of the privacy loss random variable suffice to prove $(\varepsilon, \delta)$-differential privacy for $\varepsilon > 0$. Note that $(\varepsilon, \delta)$-DP with $\delta > 0$ does not necessarily imply KL privacy, as the privacy loss random variable may be large at the tail event with $\delta$ probability.

4. Due to the connection to the privacy loss random variable (which is closely connected to the likelihood ratio test for membership hypothesis testing), both differential privacy and KL privacy incur upper bound on the performance curve of inference attacks, such as the membership inference and attribute inference [39, 28], as we discuss in Footnote 3.

## B  Deferred proofs for Section 3

### B.1  Deferred proofs for Theorem 3.1

To prove the new composition theorem, we will use the Girsanov's Theorem. Here we follow the presentation of [18, Theorem 6].

**Theorem B.1** (Implication of Girsanov's theorem [18, Theorem 6])**.** *Let $(X_t)_{t \in [0,\eta]}$ and $(\tilde{X}_t)_{t \in [0,\eta]}$ be two continuous-time processes over $\mathbb{R}^r$. Let $P_T$ be the probability measure that corresponds to the trajectory of $(X_t)_{t \in [0,\eta]}$, and let $Q_T$ be the probability measure that corresponds to the trajectory of $(\tilde{X}_t)_{t \in [0,\eta]}$. Suppose that the process $(X_t)_{t \in [0,\eta]}$ follows*

$$dX_t = b_t dt + \sigma dB_t,$$

*where $(B_t)_{t \in [0,T]}$ is a standard Brownian motion over $P_T$, and the process $(\tilde{X}_t)_{t \in [0,\eta]}$ follows*

$$d\tilde{X}_t = \tilde{b}_t dt + \sigma d\tilde{B}_t,$$

*where $(\tilde{B}_t)_{t \in [0,T]}$ is a standard Brownian motion over $Q_T$ with $d\tilde{B}_t = dB_t + \frac{1}{\sigma}\left(b_t - \tilde{b}_t\right)$. Assume that $\sigma$ is a $r \times r$ symmetric positive definite matrix. Then, provided that Novikov's condition holds,*

$$\mathbb{E}_{Q_T} \exp \left( \frac{1}{2} \int_0^\eta \|\sigma^{-1}(b_t - \tilde{b}_t)\|_2^2 dt \right) < \infty, \tag{12}$$

*we have that*

$$\frac{dP_T}{dQ_T} = \exp \left( \int_0^\eta \sigma^{-1}(b_t - \tilde{b}_t) d\tilde{B}_t - \frac{1}{2} \int_0^\eta \|\sigma^{-1}(b_t - \tilde{b}_t)\|_2^2 dt \right).$$

We are now ready to apply Girsanov's theorem to prove the following new KL privacy composition theorem for Langevin diffusion processes on neighboring datasets $\mathcal{D}$ and $\mathcal{D}'$. For ease of description, we repeat Theorem 3.1 as below.

**Theorem 3.1.** *[KL composition under possibly unbounded gradient difference] The KL divergence between running Langevin diffusion (4) for DNN (2) on neighboring datasets $\mathcal{D}$ and $\mathcal{D}'$ satisfies*

$$\text{KL}(\boldsymbol{W}_{[0:T]}\|\boldsymbol{W}'_{[0:T]}) = \frac{1}{2\sigma^2}\int_0^T \mathbb{E}\left[\|\nabla\mathcal{L}(\boldsymbol{W}_t;\mathcal{D}) - \nabla\mathcal{L}(\boldsymbol{W}_t;\mathcal{D}')\|_2^2\right]\mathrm{d}t. \quad (13)$$

*Proof.* Recall that $\boldsymbol{W}_t$ denotes the model parameter after running Langevin diffusion on dataset $\mathcal{D}$ with time $t$, and $\boldsymbol{W}'_t$ denotes the model parameter after running Langevin diffusion on dataset $\mathcal{D}'$ with time $t$. To avoid confusion, we further denote $p_{[t_1:t_2]}$ and $p'_{[t_1:t_2]}$ as the distributions of model parameters trajectories $\boldsymbol{W}_{[t_1:t_2]}$ and $\boldsymbol{W}'_{[t_1:t_2]}$ during time inteval $[t_1, t_2]$ respectively. By definition of partial derivative, we have that

$$\frac{\partial\text{KL}(\boldsymbol{W}_{[0:t]}\|\boldsymbol{W}'_{[0:t]})}{\partial t} = \lim_{\eta\to 0}\frac{\text{KL}(\boldsymbol{W}_{[0:t+\eta]}\|\boldsymbol{W}'_{[0:t+\eta]}) - \text{KL}(\boldsymbol{W}_{[0:t]}\|\boldsymbol{W}'_{[0:t]})}{\eta}. \quad (14)$$

Now we compute the term $\text{KL}(\boldsymbol{W}_{[0:t+\eta]}\|\boldsymbol{W}'_{[0:t+\eta]})$ as follows.

$$\text{KL}(\boldsymbol{W}_{[0:t+\eta]}\|\boldsymbol{W}'_{[0:t+\eta]}) = \mathbb{E}_{w_{[0:t+\eta]}\sim p_{[0:t+\eta]}}\left[\log\left(\frac{p_{[t:t+\eta]|[0:t]}\left(w_{[t:t+\eta]}|w_{[0:t]}\right)p_{[0:t]}(w_{[0:t]})}{p'_{[t:t+\eta]|[0:t]}\left(w_{[t:t+\eta]}|w_{[0:t]}\right)p'_{[0:t]}(w_{[0:t]})}\right)\right]$$

$$=\mathbb{E}_{w_{[0:t+\eta]}\sim p_{[0:t+\eta]}}\left[\log\left(\frac{p_{[t:t+\eta]|[0:t]}(w_{[t:t+\eta]}|w_{[0:t]})}{p'_{[t:t+\eta]|[0:t]}(w_{[t:t+\eta]}|w_{[0:t]})}\right) + \log\left(\frac{p_{[0:t]}(w_{[0:t]})}{p'_{[0:t]},(w_{[0:t]})}\right)\right] \quad (15)$$

where $p_{[t:t+\eta]|[0:t]}(\,\cdot\,|w_{[0:t]})$ is the conditional distribution during running Langevin diffusion on dataset $\mathcal{D}$, given fixed values for model parameters trajectory $\boldsymbol{W}_{[0:t]} = w_{[0:t]}$. Similarly, $p'_{[t:t+\eta]|[0:t]}(\,\cdot\,|w_{[0:t]})$ is the conditional distribution during running Langevin diffusion on dataset $\mathcal{D}'$.

Therefore by using the Markov property of the Langevin diffusion process and the definition of KL divergence in Eq. (15), we have that

$$\text{KL}(\boldsymbol{W}_{[0:t+\eta]}\|\boldsymbol{W}'_{[0:t+\eta]}) = \mathbb{E}_{w_t\sim p_t}\left[\text{KL}\left(p_{[t:t+\eta]|t}(\,\cdot\,|w_t)\Big\|p'_{[t:t+\eta]|t}(\,\cdot\,|w_t)\right)\right] + \text{KL}(p_t, p'_t). \quad (16)$$

Now to compute the term $\text{KL}\left(p_{[t:t+\eta]|t}(\,\cdot\,|w_t)\Big\|p'_{[t:t+\eta]|t}(\,\cdot\,|w_t)\right)$, we only need to apply Theorem B.1 to the following two Langevin diffusion processes $(\boldsymbol{W}_{t+s|t})_{s\in[0,\eta]}$ and $(\boldsymbol{W}'_{t+s|t})_{s\in[0,\eta]}$, conditioning on the observation $\boldsymbol{W}_{t|t} = \boldsymbol{W}'_{t|t} = w_t$ at time $t$.

$$\mathrm{d}\boldsymbol{W}_{t+s|s} = -\nabla\mathcal{L}(\boldsymbol{W}_{t+s|t};\mathcal{D})\mathrm{d}t + \sqrt{2\sigma^2}\mathrm{d}B_s.$$
$$\mathrm{d}\boldsymbol{W}'_{t+s|t} = -\nabla\mathcal{L}(\boldsymbol{W}'_{t+s|t};\mathcal{D}')\mathrm{d}t + \sqrt{2\sigma^2}\mathrm{d}\tilde{B}_s.$$

Note that when $\eta$ is small enough, we have that the Novikov's condition in Eq. (12) holds because the exponent inside integration $\frac{1}{2}\int_0^\eta\|\sigma_s^{-1}(b_s - \tilde{b}_s)\|_2^2\mathrm{d}s$ scales linearly with $\eta$ that can be arbitrarily small. Therefore, by applying Girsanov's theorem, we have that

$$\text{KL}\left(p_{[t:t+\eta]|t}(\,\cdot\,|w_t)\Big\|p'_{[t:t+\eta]|t}(\,\cdot\,|w_t)\right) = \mathbb{E}\left[\int_0^\eta \sigma^{-1}(b_s - \tilde{b}_s)\mathrm{d}\tilde{B}_s - \frac{1}{2}\int_0^T\|\sigma^{-1}(b_s - \tilde{b}_s)\|_2^2\mathrm{d}s\right],$$

where $b_s - \tilde{b}_s = -\nabla\mathcal{L}(\boldsymbol{W}_{t+s|t};\mathcal{D}) + \nabla\mathcal{L}(\boldsymbol{W}_{t+s|t};\mathcal{D}')$. By $d\tilde{B}_s = \mathrm{d}B_s + \frac{1}{\sigma}\left(b_s - \tilde{b}_s\right)$ and Itô integration with regard to $\boldsymbol{W}_{[t:t+\eta]|t}$, we have that

$$\text{KL}\left(p_{[t:t+\eta]|t}(\,\cdot\,|w_t)\Big\|p'_{[t:t+\eta]|t}(\,\cdot\,|w_t)\right) = \frac{\mathbb{E}\left[\int_0^\eta\|\nabla\mathcal{L}(\boldsymbol{W}_{t+s|t};\mathcal{D}) - \nabla\mathcal{L}(\boldsymbol{W}_{t+s|t};\mathcal{D}')\|_2^2\mathrm{d}s\right]}{2\sigma^2}. \quad (17)$$

By plugging Eq. (17) into Eq. (16), we have that

$$\text{KL}(p_{[t:t+\eta]}, p'_{[t:t+\eta]}) = \frac{\mathbb{E}\left[\int_0^\eta \|\nabla\mathcal{L}(\boldsymbol{W}_{t+s}; \mathcal{D}) - \nabla\mathcal{L}(\boldsymbol{W}_{t+s}; \mathcal{D}')\|_2^2 \, ds\right]}{2\sigma^2} + \text{KL}(p_t, p'_t). \quad (18)$$

By plugging Eq. (18) into Eq. (14), and by exchanging the order of expectation and integration, we have that

$$\begin{aligned}
\frac{\partial \text{KL}(p_t, p'_t)}{\partial t} &= \frac{1}{2\sigma^2} \lim_{\eta \to 0} \frac{\int_0^\eta \mathbb{E}_{p_{t+s}}\left[\|\nabla\mathcal{L}(\boldsymbol{W}_{t+s}; \mathcal{D}) - \nabla\mathcal{L}(\boldsymbol{W}_{t+s}; \mathcal{D}')\|_2^2\right] ds}{\eta} \\
&= \frac{1}{2\sigma^2} \mathbb{E}_{p_t}\left[\|\nabla\mathcal{L}(\boldsymbol{W}_t; \mathcal{D}) - \nabla\mathcal{L}(\boldsymbol{W}_t; \mathcal{D}')\|_2^2\right].
\end{aligned} \quad (19)$$

Note that here we exchange the order of expectation and integration by using the Tonelli's Theorem [49] for non-negative integrand function $\|\nabla\mathcal{L}(\boldsymbol{W}_{t+s}; \mathcal{D}) - \nabla\mathcal{L}(\boldsymbol{W}_{t+s}; \mathcal{D}')\|_2^2$. Integrating Eq. (19) on $t \in [0, T]$ finishes the proof. $\qquad\square$

## B.2  Deferred proofs for Lemma 3.2

**Lemma 3.2.** *Let $M_T$ be the subspace spanned by gradients $\{\nabla\ell(f_{\boldsymbol{W}_t}(\boldsymbol{x}_i; \boldsymbol{y}_i) : (\boldsymbol{x}_i, \boldsymbol{y}_i) \in \mathcal{D}, t \in [0, T]\}_{i=1}^n$ throughout Langevin diffusion $(\boldsymbol{W}_t)_{t \in [0,T]}$. Denote $\|\cdot\|_{M_T}$ as the $\ell_2$ norm of the projected input vector onto $M_T$. Suppose that there exists constants $c, \beta > 0$ such that for any $\boldsymbol{W}$, $\boldsymbol{W}'$ and $(\boldsymbol{x}, \boldsymbol{y})$, we have $\|\nabla\ell(f_{\boldsymbol{W}}(\boldsymbol{x}); \boldsymbol{y})) - \nabla\ell(f_{\boldsymbol{W}'}(\boldsymbol{x}); \boldsymbol{y})\|_2 < \max\{c, \beta\|\boldsymbol{W} - \boldsymbol{W}'\|_{M_T}\}$. Then running Langevin diffusion Eq. (4) with Gaussian initialization distribution (5) satisfies $\varepsilon$-KL privacy with $\varepsilon = \frac{\max_{\mathcal{D}, \mathcal{D}'} \int_0^T \mathbb{E}\left[\|\nabla\mathcal{L}(\boldsymbol{W}_t; \mathcal{D}) - \nabla\mathcal{L}(\boldsymbol{W}_t; \mathcal{D}')\|_2^2\right] dt}{2\sigma^2}$ where*

$$\int_0^T \mathbb{E}\left[\|\nabla\mathcal{L}(\boldsymbol{W}_t; \mathcal{D}) - \nabla\mathcal{L}(\boldsymbol{W}_t; \mathcal{D}')\|_2^2\right] dt \leq 2T \cdot \underbrace{\mathbb{E}\left[\|\nabla\mathcal{L}(\boldsymbol{W}_0; \mathcal{D}) - \nabla\mathcal{L}(\boldsymbol{W}_0; \mathcal{D}')\|_2^2\right]}_{\textit{gradient difference at initialization}}$$

$$+ \underbrace{\frac{2\beta^2}{n^2(2+\beta^2)}\left(\frac{e^{(2+\beta^2)T}-1}{2+\beta^2} - T\right) \cdot \left(\mathbb{E}\left[\|\nabla\mathcal{L}(\boldsymbol{W}_0; \mathcal{D})\|_2^2\right] + 2\sigma^2 rank(M_T) + c^2\right)}_{\textit{gradient difference fluctuation during training}} + \underbrace{\frac{2c^2 T}{n^2}}_{\textit{non-smoothness}}.$$

*Proof.* By definition of the neighboring datasets $\mathcal{D}$ and $\mathcal{D}'$, and the definition of empirical risk in Eq. (1), we have that for any $\boldsymbol{W}$, it satisfies that

$$\|\nabla\mathcal{L}(\boldsymbol{W}; \mathcal{D}) - \nabla\mathcal{L}(\boldsymbol{W}; \mathcal{D}')\|_2^2 = \frac{1}{n^2} \|\ell(f_{\boldsymbol{W}}(\boldsymbol{x}); \boldsymbol{y})) - \nabla\ell(f_{\boldsymbol{W}}(\boldsymbol{x}'); \boldsymbol{y}')\|_2^2, \quad (20)$$

where $(\boldsymbol{x}, \boldsymbol{y})$ and $(\boldsymbol{x}', \boldsymbol{y}')$ are the differing records between neighboring datasets $\mathcal{D}$ amd $\mathcal{D}'$. By the assumption that $\|\nabla\ell(f_{\boldsymbol{W}}(\boldsymbol{x}); \boldsymbol{y})) - \nabla\ell(f_{\boldsymbol{W}'}(\boldsymbol{x}); \boldsymbol{y})\|_2 < \max\{c, \beta\|\boldsymbol{W} - \boldsymbol{W}'\|_{M_T}\}$, and by the Cauchy-Schwarz inequality, we further have that for any $\boldsymbol{W}$ and $\boldsymbol{W}_t$, it satisfies that

$$\begin{aligned}
\|\nabla\ell(f_{\boldsymbol{W}_t}(\boldsymbol{x}); \boldsymbol{y})) - \nabla\ell(f_{\boldsymbol{W}_t}(\boldsymbol{x}'); \boldsymbol{y}')\|_2^2 \leq &\, 2\|\nabla\ell(f_{\boldsymbol{W}_0}(\boldsymbol{x}); \boldsymbol{y})) - \nabla\ell(f_{\boldsymbol{W}_0}(\boldsymbol{x}'); \boldsymbol{y}')\|_2^2 \\
&+ 2\beta^2\|\boldsymbol{W}_t - \boldsymbol{W}_0\|_{M_T}^2 + 2c^2.
\end{aligned} \quad (21)$$

The first term $\|\nabla\ell(f_{\boldsymbol{W}_0}(\boldsymbol{x}); \boldsymbol{y})) - \nabla\ell(f_{\boldsymbol{W}_0}(\boldsymbol{x}'); \boldsymbol{y}')\|_2^2$ is constant during training (as it only depends on the initialization). Therefore, we only need to bound the second term $\|\boldsymbol{W}_t - \boldsymbol{W}_0\|_{M_T}^2$. For brevity, we denote the function $\Phi(\boldsymbol{W}) = \|\boldsymbol{W} - \boldsymbol{W}_0\|_{M_T}^2$. Recall our definition, $p_t$ as the distribution of model parameters after running Langevin diffusion on dataset $\mathcal{D}$ with time $t$, and $p'_t$ as the distribution of model parameters after running Langevin diffusion on dataset $\mathcal{D}'$ with time $t$. Then we have that

$$\frac{\partial}{\partial t}\mathbb{E}_{p_t}[\Phi(\boldsymbol{W})] = \lim_{\eta \to 0} \frac{\mathbb{E}_{p_{t+\eta}}[\Phi(\boldsymbol{W})] - \mathbb{E}_{p_t}[\Phi(\boldsymbol{W})]}{\eta}. \quad (22)$$

Denote $\Gamma_s$ as the following random operator on model parameters $\theta$.

$$\Gamma_s(\boldsymbol{W}) = \theta - s\nabla\mathcal{L}(\boldsymbol{W}; \mathcal{D}) + \sqrt{2\sigma^2 s}Z,$$

where $Z \sim \mathcal{N}(0, \mathbb{I})$. We first claim that the following equation holds.

$$\lim_{\eta \to 0} \frac{\mathbb{E}_{p_{t+\eta}}[\Phi(\boldsymbol{W})] - \mathbb{E}_{p_t}[\Phi(\Gamma_\eta(\boldsymbol{W}))]}{\eta} = 0 \,. \tag{23}$$

This is by using Euler-Maruyama discretization method to approximate the solution $\boldsymbol{W}_t$ of SDE Eq. (4). More specifically, the approximation error $\mathbb{E}_{p_{t+\eta}}[\Phi(\boldsymbol{W})] - \mathbb{E}_{p_t}[\Phi(\Gamma_\eta(\boldsymbol{W}))]$ is of size $O(r\eta^2)$ for small $\eta$, where $r$ is the dimension of $\boldsymbol{W}$.

Therefore, by plugging Eq. (23) into Eq. (22), we have that

$$\frac{\partial}{\partial t} \mathbb{E}_{p_t}[\Phi(\boldsymbol{W})] = \lim_{\eta \to 0} \frac{\mathbb{E}_{p_t}[\Phi(\Gamma_\eta(\boldsymbol{W}))] - \mathbb{E}_{p_t}[\Phi(\boldsymbol{W})]}{\eta} \,.$$

Recall that $\nabla^2 \Phi(\boldsymbol{W})$ exists almost everywhere with regard to $\boldsymbol{W} \sim p_t$. Therefore we could approximate the term $\mathbb{E}_{p_t}[\Phi(\Gamma_\eta(\boldsymbol{W}); \mathcal{D}, \mathcal{D}')]$ via its second-order Taylor expansion at $\boldsymbol{W}$ as follows.

$$\frac{\partial}{\partial t} \mathbb{E}_{p_t}[\Phi(\boldsymbol{W})] = \lim_{\eta \to 0} \frac{\mathbb{E}_{p_t}[\langle \nabla \Phi(\boldsymbol{W}), -\eta \nabla \mathcal{L}(\boldsymbol{W}; \mathcal{D}) + \sqrt{2\sigma^2 \eta} Z \rangle + \sigma^2 \eta Z^\top \nabla^2 \Phi(\boldsymbol{W}) Z + o(\eta)]}{\eta}$$
$$= -\mathbb{E}_{p_t}[\langle \nabla \Phi(\boldsymbol{W}), \nabla \mathcal{L}(\boldsymbol{W}; \mathcal{D}) \rangle] + \sigma^2 \mathbb{E}_{p_t}[\text{Tr}\left(\nabla^2 \Phi(\boldsymbol{W})\right)] \,. \tag{24}$$

By plugging $\Phi(\boldsymbol{W}) = \|\boldsymbol{W} - \boldsymbol{W}_0\|_{M_T}^2$ into Eq. (24), we have that

$$\frac{\partial}{\partial t} \mathbb{E}_{p_t}[\|\boldsymbol{W} - \boldsymbol{W}_0\|_{M_T}^2] \leq -2\mathbb{E}_{p_t}[\langle \boldsymbol{W} - \boldsymbol{W}_0, \nabla \mathcal{L}(\boldsymbol{W}; \mathcal{D}) \rangle] + 2\sigma^2 \text{rank}(M_T) \tag{25}$$
$$= -2\mathbb{E}_{p_t}[\langle \boldsymbol{W} - \boldsymbol{W}_0, \nabla \mathcal{L}(\boldsymbol{W}_0; \mathcal{D}) \rangle] + 2\sigma^2 \text{rank}(M_T)$$
$$- 2\mathbb{E}_{p_t}[\langle \boldsymbol{W} - \boldsymbol{W}_0, \nabla \mathcal{L}(\boldsymbol{W}; \mathcal{D}) - \nabla \mathcal{L}(\boldsymbol{W}_0; \mathcal{D}) \rangle]$$
$$\leq \mathbb{E}[\|\nabla \mathcal{L}(\boldsymbol{W}_0; \mathcal{D})\|_2^2] + \mathbb{E}_{p_t}[\|\boldsymbol{W} - \boldsymbol{W}_0\|_{M_t}^2] + 2\sigma^2 \text{rank}(M_T) \tag{26}$$
$$+ \mathbb{E}_{p_t}[\|\boldsymbol{W} - \boldsymbol{W}_0\|_{M_T}^2] + \mathbb{E}_{p_t}[\|\nabla \mathcal{L}(\boldsymbol{W}; \mathcal{D}) - \nabla \mathcal{L}(\boldsymbol{W}_0; \mathcal{D})\|_2^2] \,, \tag{27}$$

where the last inequality is by using the Cauchy-schwartz inequality. By plugging the assumption that $\|\nabla \ell(f_{\boldsymbol{W}}(\boldsymbol{x}); \boldsymbol{y}) - \nabla \ell(f_{\boldsymbol{W}'}(\boldsymbol{x}); \boldsymbol{y})\|_2 < \max\{c, \beta\|\boldsymbol{W} - \boldsymbol{W}'\|_2\}$ into the above inequality, we have that

$$\frac{\partial}{\partial t} \mathbb{E}_{p_t}[\|\boldsymbol{W} - \boldsymbol{W}_0\|_{M_T}^2] \leq (2 + \beta^2) \mathbb{E}_{p_t}[\|\boldsymbol{W} - \boldsymbol{W}_0\|_{M_T}^2] + \mathbb{E}[\|\nabla \mathcal{L}(\boldsymbol{W}_0; \mathcal{D})\|_2^2] + 2\sigma^2 \text{rank}(M_T) + c^2 \,. \tag{28}$$

By solving the above ordinary differential inequality on $t \in [0, T]$, we have that

$$\mathbb{E}_{p_t}[\|\boldsymbol{W} - \boldsymbol{W}_0\|_{M_T}^2] \leq \frac{e^{(2+\beta^2)t} - 1}{2 + \beta^2} \left(\mathbb{E}[\|\nabla \mathcal{L}(\boldsymbol{W}_0; \mathcal{D})\|_2^2] + 2\sigma^2 \text{rank}(M_T) + c^2\right) \,. \tag{29}$$

By plugging Eq. (29) into Eq. (21) and Eq. (20), followed by integration over time $t \in [0, T]$, we have that

$$\int_0^T \mathbb{E}_{p_t}\left[\|\nabla \mathcal{L}(\boldsymbol{W}; \mathcal{D}) - \nabla \mathcal{L}(\boldsymbol{W}; \mathcal{D}')\|_2^2\right] \mathrm{d}t \leq 2T \cdot \mathbb{E}_{p_0}\left[\|\nabla \mathcal{L}(\boldsymbol{W}; \mathcal{D}) - \nabla \mathcal{L}(\boldsymbol{W}; \mathcal{D}')\|_2^2\right] \tag{30}$$

$$+ \frac{2\beta^2}{n^2(2 + \beta^2)} \left(\frac{e^{(2+\beta^2)T} - 1}{2 + \beta^2} - T\right) \cdot \left(\mathbb{E}_{p_0}\left[\|\nabla \mathcal{L}(\boldsymbol{W}; \mathcal{D})\|_2^2\right] + 2\sigma^2 \text{rank}(M_T) + c^2\right) + \frac{2c^2 T}{n^2} \,, \tag{31}$$

which concludes our proof.

$$\square$$

# C Deferred proofs for Section 4

## C.1 Bounding the gradient norm at initialization

To bound the moment of $\ell_2$ norm of the gradient $\frac{\partial f(\boldsymbol{x})}{\partial \boldsymbol{W}_l}$ of network output function, we need the following (extended) lemmas from Zhu et al. [54].

**Lemma C.1** ([54, Lemma 1]). *Let $\boldsymbol{w} \sim \mathcal{N}(0, \sigma^2 \mathbb{I}_n)$, then for two fixed non-zero vectors $\boldsymbol{h}_1, \boldsymbol{h}_2 \in \mathbb{R}^n$ whose correlation is unknown, define two random variables $X = (\boldsymbol{w}^\top \boldsymbol{h}_1 1_{\{\boldsymbol{w}^\top \boldsymbol{h}_2 \geq 0\}})^2$ and $Y = s(\boldsymbol{w}^\top \boldsymbol{h}_1)^2$, where $s \sim Ber(1, 1/2)$ follows a Bernoulli distribution with 1 trial and $\frac{1}{2}$ success rate, and $s$ and $\boldsymbol{w}$ are independent random variables. Then $X$ and $Y$ have the same distribution.*

**Lemma C.2** (Extension of [54, Lemma 2]). *Given a fixed non-zero matrix $\boldsymbol{H}_1 \in \mathbb{R}^{p \times r}$ and a fixed non-zero vector $\boldsymbol{h}_2 \in \mathbb{R}^p$ and , let $\boldsymbol{W} \in \mathbb{R}^{q \times p}$ be a random matrix with i.i.d. entries $W_{ij} \sim \mathcal{N}(0, \beta)$ and a matrix (or vector) $\boldsymbol{V} = \phi'(\boldsymbol{W} \boldsymbol{h}_2) \boldsymbol{W} \boldsymbol{H}_1 \in \mathbb{R}^{q \times r}$, then, we have $\mathbb{E}[\frac{\|\boldsymbol{V}\|_F^2}{\|\boldsymbol{H}_1\|_F^2}] = \frac{q\beta}{2}$.*

*Proof.* According to the definition of $\boldsymbol{V} = \phi'(\boldsymbol{W} \boldsymbol{h}_2) \boldsymbol{W} \boldsymbol{H}_1 \in \mathbb{R}^{q \times r}$, we have:

$$\|\boldsymbol{V}\|_F^2 = \sum_{i=1}^q \sum_{j=1}^r \left( \boldsymbol{D}_{i,i} \langle \boldsymbol{W}^{[i;]}, \boldsymbol{H}_1^{[;j]} \rangle \right)^2 ,$$

where $\boldsymbol{D}_{i,i} = 1_{\{\langle \boldsymbol{W}^{[i]}, \boldsymbol{h}_2 \rangle \geq 0\}}$, $\boldsymbol{W}^{[i;]}$ is the $i$-th row of $\boldsymbol{W}$, and $\boldsymbol{H}_1^{[;j]}$ is the $j$-th column vector of $\boldsymbol{H}_1$. Therefore by Lemma C.1, with i.i.d. Bernoulli random variable $\rho_1, \cdots, \rho_q \sim Ber(1, 1/2)$, we have

$$\|\boldsymbol{V}\|_F^2 \overset{d}{=} \sum_{i=1}^q \sum_{j=1}^r \rho_i \langle \boldsymbol{W}^{[i;]}, \boldsymbol{H}_1^{[;j]} \rangle^2 = \sum_{i=1}^q \sum_{j=1}^r \rho_i \beta \|\boldsymbol{H}_1^{[;j]}\|_2^2 \tilde{w}_{ij}^2 , \tag{32}$$

where $\tilde{w}_{ij} = \langle \boldsymbol{W}^{[i;]}, \boldsymbol{H}_1^{[;j]} \rangle / \left( \sqrt{\beta \|\boldsymbol{H}_1^{[;j]}\|_2^2} \right)$. By the fact that $\boldsymbol{W}^{[i;]}$ has i.i.d. Gaussian entries, for any fixed $j$, we have that $\tilde{w}_{ij} \sim \mathcal{N}(0, 1), i = 1, \cdots, q$ independently. Therefore, we have

$$\mathbb{E}\left[\|\boldsymbol{V}\|_F^2\right] = \sum_{i=1}^q \sum_{j=1}^r \mathbb{E}[\rho_i] \beta \|\boldsymbol{H}_1^{[;j]}\|_2^2 \mathbb{E}[\tilde{w}_{ij}^2] = \frac{q\beta}{2} \mathbb{E}[\|\boldsymbol{H}_1\|_F^2] .$$

$\square$

Now, we are ready to prove output gradient expectation at random initialization as follows.

**Lemma C.3** (Output Gradient Expectation Bound at Random Initialization). *Fix any data record $\boldsymbol{x}$, then over the randomness of the initialization distributions for $\boldsymbol{W}_1, \cdots, \boldsymbol{W}_L$, i.e., $\boldsymbol{W}_l \sim \mathcal{N}(0, \beta_l \mathbb{I})$ for $l = 1, \cdots, L - 1$, it satisfies that*

$$\mathbb{E}_{\boldsymbol{W}} \left[ \|\frac{\partial f(\boldsymbol{x})}{\partial Vec(\boldsymbol{W})}\|_F^2 \right] = \|\boldsymbol{x}\|_2^2 o \left( \prod_{i=1}^{L-1} \frac{\beta_i m_i}{2} \right) \sum_{l=1}^L \frac{\beta_L}{\beta_l} . \tag{33}$$

*Proof.* We use $Vec(\boldsymbol{W}_l)$ to denote the concatenation of all row vector of the parameter matrix $\boldsymbol{W}_l$. By chain rule, for $l = 1, \cdots, L - 1$, we have that

$$\frac{\partial \boldsymbol{f}(\boldsymbol{x})}{\partial Vec(\boldsymbol{W}_l)} = \frac{\partial h_L(\boldsymbol{x})}{\partial h_{L-1}(\boldsymbol{x})} \left( \prod_{i=1}^{L-1-l} \frac{\partial h_{L-i}(\boldsymbol{x})}{\partial h_{L-1-i}(\boldsymbol{x})} \right) \frac{\partial h_l}{Vec(\boldsymbol{W}_l)} \tag{34}$$

$$= \boldsymbol{W}_L \left( \prod_{i=1}^{L-1-l} \sigma'_{L-i} \boldsymbol{W}_{L-i} \right) \sigma'_l \begin{pmatrix} h_{l-1}^\top & 0 & \cdots \\ \vdots & \vdots & \vdots \\ 0 & 0 & h_{l-1}^\top \end{pmatrix}_{m_l \times m_l m_{l-1}} . \tag{35}$$

Similarly, for the $L$-th layer, we have that

$$\frac{\partial \boldsymbol{f}(\boldsymbol{x})}{\partial \mathrm{Vec}(\boldsymbol{W}_L)} = \begin{pmatrix} h_{L-1}^\top & 0 & \cdots \\ \vdots & \vdots & \vdots \\ 0 & 0 & h_{L-1}^\top \end{pmatrix}_{o \times om_{L-1}}. \tag{36}$$

By properties of ReLU activation $\phi$, we have $\phi'_{L-i} = \mathrm{diag}[\mathrm{sgn}(W_{L-i}h_{L-1-i})]$, where $\mathrm{sgn}(x) = \begin{cases} 1 & x > 0 \\ 0 & x \le 0 \end{cases}$ operates coordinate-wise with regard to the input matrix. Therefore, we have that for $l = 1, \cdots, L-1$

$$\frac{\partial \boldsymbol{f}(\boldsymbol{x})}{\partial \mathrm{Vec}(\boldsymbol{W}_l)} = \boldsymbol{W}_L \left( \prod_{i=1}^{L-1-l} \mathrm{diag}[\mathrm{sgn}(W_{L-i}h_{L-1-i})]\boldsymbol{W}_{L-i} \right) \cdot \mathrm{diag}[\mathrm{sgn}(W_l h_{l-1})] \begin{pmatrix} h_{l-1}^\top & 0 & \cdots \\ \vdots & \vdots & \vdots \\ 0 & 0 & h_{l-1}^\top \end{pmatrix}_{m_l \times m_l m_{l-1}}.$$

For notational simplicity, we introduce the notation of $\boldsymbol{t}_l^{l'}$ for $l = 1, \cdots, L-1$ and $l \le l' < L$ as follows.

$$\boldsymbol{t}_l^{l'} := \left( \prod_{i=L-l'}^{L-1-l} \mathrm{diag}[\mathrm{sgn}(W_{L-i}h_{L-1-i})]\boldsymbol{W}_{L-i} \right) \cdot \mathrm{diag}[\mathrm{sgn}(W_l h_{l-1})] \begin{pmatrix} h_{l-1}^\top & 0 & \cdots \\ \vdots & \vdots & \vdots \\ 0 & 0 & h_{l-1}^\top \end{pmatrix}_{m_l \times m_l m_{l-1}}.$$

Then by definition, we have that

$$\mathbb{E}_{\boldsymbol{W}} \left[ \|\frac{\partial f(\boldsymbol{x})}{\partial \mathrm{Vec}(\boldsymbol{W}_l)}\|_F^2 \right] = \mathbb{E}_{\boldsymbol{W}} \left[ \|\boldsymbol{W}_L \boldsymbol{t}_l^{L-1}\|_F^2 \right] = \mathbb{E}_{\boldsymbol{W}} \left[ \frac{\|\boldsymbol{W}_L \boldsymbol{t}_l^{L-1}\|_F^2}{\|\boldsymbol{t}_l^{L-1}\|_F^2} \cdot \frac{\|\boldsymbol{t}_l^{L-1}\|_F^2}{\|\boldsymbol{t}_l^{L-2}\|_F^2} \cdots \frac{\|\boldsymbol{t}_l^{l+1}\|_F^2}{\|\boldsymbol{t}_l^l\|_F^2} \cdot \|\boldsymbol{t}_l^l\|_F^2 \right]$$

$$= \mathbb{E}_{\boldsymbol{W}_1, \cdots, \boldsymbol{W}_l} \left[ \|\boldsymbol{t}_l^l\|_F^2 \cdot \mathbb{E}_{\boldsymbol{W}_{l+1}} \left[ \frac{\|\boldsymbol{t}_l^{l+1}\|_F^2}{\|\boldsymbol{t}_l^l\|_F^2} \cdots \mathbb{E}_{\boldsymbol{W}_L} \left[ \frac{\|\boldsymbol{W}_L \boldsymbol{t}_l^{L-1}\|_F^2}{\|\boldsymbol{t}_l^{L-1}\|_F^2} \right] \right] \right].$$

By rotational invariance of Gaussian column vectors, we prove that for any possible value of $\boldsymbol{t}_l^{L-1}$ (which is completely determined by $\boldsymbol{W}_1, \cdots, \boldsymbol{W}_{L-1}$ and $\boldsymbol{x}$), for any $l = 1, \cdots, L-1$, we have that

$$\mathbb{E}_{\boldsymbol{W}_L} \left[ \frac{\|\boldsymbol{W}_L \boldsymbol{t}_l^{L-1}\|_F^2}{\|\boldsymbol{t}_l^{L-1}\|_F^2} \right] = \mathbb{E}_{\boldsymbol{W}_L} \left[ \frac{\|\boldsymbol{W}_L e_1\|_2^2}{\|e_1\|_2^2} \right] = \beta_L o. \tag{37}$$

By Lemma C.2, for any $l = 1, \cdots, L-2$ and $l \le l' \le L-2$, we have that

$$\mathbb{E}_{\boldsymbol{W}_{l+1}} \left[ \frac{\|\boldsymbol{t}_l^{l'+1}\|_F^2}{\|\boldsymbol{t}_l^{l'}\|_F^2} \right] = \frac{\beta_{l'+1}}{2} m_{l'+1}. \tag{38}$$

We now bound the last term $\mathbb{E}_{\boldsymbol{W}_1, \cdots, \boldsymbol{W}_l} \left[ \|\boldsymbol{t}_l^l\|_F^2 \right]$ for $l = 1, \cdots, L-1$. By definition, we have that

$$\mathbb{E}_{\boldsymbol{W}_1, \cdots, \boldsymbol{W}_l} \left[ \|\boldsymbol{t}_l^l\|_F^2 \right] = \mathbb{E}_{\boldsymbol{W}_1, \cdots, \boldsymbol{W}_l} \left[ \sum_{i=1}^{m_l} 1_{\{\boldsymbol{W}_l^{[i;]} h_{l-1} \le 0\}} \cdot \|h_{l-1}\|_2^2 \right] = \frac{m_l}{2} \mathbb{E}_{\boldsymbol{W}_1, \cdots, \boldsymbol{W}_{l-1}} \mathbb{E}[\|h_{l-1}\|_2^2]. \tag{39}$$

To bound the $\mathbb{E}_{\boldsymbol{W}_1, \cdots, \boldsymbol{W}_l} [\|h_{l-1}\|_2^2]$, note that by Lemma C.2, we prove that for any $l = 1, \cdots, L-1$

$$\mathbb{E}_{\boldsymbol{W}_{l-1}} \left[ \frac{\|h_{l-1}(\boldsymbol{x})\|_2^2}{\|h_{l-2}(\boldsymbol{x})\|_2^2} \right] = \frac{\beta_{l-1}}{2} m_{l-1}. \tag{40}$$

Therefore, for any $l = 1, \cdots, L$, we have that

$$\mathbb{E}_{\boldsymbol{W}_1, \cdots, \boldsymbol{W}_{l-1}} \left[ \|h_{l-1}(\boldsymbol{x})\|_2^2 \right] = \mathbb{E}_{\boldsymbol{W}_1, \cdots, \boldsymbol{W}_{l-1}} \left[ \frac{\|h_{l-1}(\boldsymbol{x})\|_2^2}{\|h_{l-2}(\boldsymbol{x})\|_2^2} \cdots \frac{\|h_1(\boldsymbol{x})\|_2^2}{\|\boldsymbol{x}\|_2^2} \right] \cdot \|\boldsymbol{x}\|_2^2 \tag{41}$$

$$= \left( \prod_{i=1}^{l-1} \frac{\beta_i}{2} m_i \right) \|\boldsymbol{x}\|_2^2 . \tag{42}$$

By plugging (42) into (39), we have that

$$\mathbb{E}_{\boldsymbol{W}_1, \cdots, \boldsymbol{W}_l} \left[ \|\boldsymbol{t}_l^l\|_2^2 \right] = \frac{m_l}{2} \left( \prod_{i=1}^{l-1} \frac{\beta_i}{2} m_i \right) \|\boldsymbol{x}\|_2^2 . \tag{43}$$

By combining (37), (38) and (43), we have for any $l = 1, \cdots, L-1$

$$\mathbb{E}_{\boldsymbol{W}} \left[ \|\frac{\partial f(\boldsymbol{x})}{\partial \mathrm{Vec}(\boldsymbol{W}_l)}\|_F^2 \right]$$

$$= \frac{m_l}{2} \left( \prod_{i=1}^{l-1} \frac{\beta_i}{2} m_i \right) \cdot \left( \prod_{i=l+1}^{L-1} \frac{\beta_i}{2} m_i \right) \cdot \beta_L o \cdot \|\boldsymbol{x}\|_2^2 = \frac{\beta_L}{\beta_l} \|\boldsymbol{x}\|_2^2 o \left( \prod_{i=1}^{L-1} \frac{\beta_i m_i}{2} \right) .$$

On the other hand, by plugging Eq. (42) (under $\ell = L$) into Eq. (36), we have that

$$\mathbb{E}_{\boldsymbol{W}} \left[ \|\frac{\partial f(\boldsymbol{x})}{\partial \mathrm{Vec}(\boldsymbol{W}_L)}\|_2^2 \right] = o \left( \prod_{i=1}^{L-1} \frac{\beta_i}{2} m_i \right) \|\boldsymbol{x}\|_2^2$$

Therefore,

$$\mathbb{E}_{\boldsymbol{W}} \left[ \|\frac{\partial f(\boldsymbol{x})}{\partial \mathrm{Vec}(\boldsymbol{W})}\|_F^2 \right] = \sum_{l=1}^{L} \|\frac{\partial f(\boldsymbol{x})}{\partial \boldsymbol{W}_l}\|_F^2 = \|\boldsymbol{x}\|_2^2 o \left( \prod_{i=1}^{L-1} \frac{\beta_i m_i}{2} \right) \sum_{l=1}^{L} \frac{\beta_L}{\beta_l} ,$$

which suffices to prove Eq. (33). $\qquad\square$

## C.2 Deferred proof for Theorem 4.1

Finally, we prove that the gradient difference between two training datasets under linearized network is bounded by a constant throughout training (which only depends on the network width, depth and initialization distribution).

**Theorem 4.1.** *Under Assumption 2.2, taking over the randomness of the random initialization and the Brownian motion, for any $t \in [0, T]$, running Langevin diffusion on linearized network Eq. (3) satisfies that*

$$\mathbb{E} \left[ \|\nabla \mathcal{L}(\boldsymbol{W}_t^{lin}; \mathcal{D}) - \mathcal{L}(\boldsymbol{W}_t^{lin}; \mathcal{D}')\|_2^2 \right] \le \frac{4B}{n^2} , \tag{44}$$

*where $n$ is the training dataset size, and $B$ is a constant that only depends on the data dimension $d$, the number of classes $o$, the network depth $L$, the per-layer network width $\{m_i\}_{i=1}^L$, and the per-layer variances $\{\beta_i\}_{i=1}^L$ of the Gaussian initialization distribution as follows.*

$$B := d \cdot o \cdot \left( \prod_{i=1}^{L-1} \frac{\beta_i m_i}{2} \right) \sum_{l=1}^{L} \frac{\beta_L}{\beta_l} , \tag{45}$$

*Proof.* Denote $\boldsymbol{W}$ as the initialization parameters and denote $\boldsymbol{W}_t^{lin}$ as the parameters for linearized network after training time $t$. Then the gradient difference under linearized network and cross-entropy loss function is as follows.

$$\|\nabla \mathcal{L}(W_t; \mathcal{D}) - \nabla \mathcal{L}(W_t; \mathcal{D}')\|_F^2$$

$$= \left\| \frac{\nabla \boldsymbol{f_W}(\boldsymbol{x})^\top (\mathrm{softmax}(\boldsymbol{f_{W_t}}(\boldsymbol{x})) - \boldsymbol{y})}{n} - \frac{\nabla \boldsymbol{f_W}(\boldsymbol{x}')^\top (\mathrm{softmax}(\boldsymbol{f_{W_t}}(\boldsymbol{x}')) - \boldsymbol{y}')}{n} \right\|_F^2$$

$$\le \frac{2}{n^2} \left( \|\nabla \boldsymbol{f_W}(\boldsymbol{x})\|_F^2 + \|\nabla \boldsymbol{f_W}(\boldsymbol{x}')\|_F^2 \right) .$$

Plugging Lemma C.3 into the above equation with data Assumption 2.2 suffice to prove the result. $\quad\square$

# D  Deferred proofs for Section 6

## D.1  Deferred proof for Lemma 6.1 on convergence of training linearized network

In this section, we prove empirical risk bound for the average of all iterates of Langevin diffusion, building on standard results [42, Theorem 2] and [45, Theorem 3.1] for the (discrete time) stochastic gradient descent algorithm.

**Lemma 6.1.** (Extension of [42, Theorem 2] and [45, Theorem 3.1]) *Let $\mathcal{L}_0^{lin}(\boldsymbol{W}; \mathcal{D})$ be the empirical risk function of linearized newtork Eq. (3) expanded at initialization vector $\boldsymbol{W}_0^{lin}$. Let $\boldsymbol{W}_0^*$ be an $\alpha$-near-optimal solution for the ERM problem such that $\mathcal{L}_0^{lin}(\boldsymbol{W}_0^*; \mathcal{D}) - \min_{\boldsymbol{W}} \mathcal{L}_0^{lin}(\boldsymbol{W}; \mathcal{D}) \leq \alpha$. Let $\mathcal{D} = \boldsymbol{x}_1, \cdots, \boldsymbol{x}_n$ be an arbitrary training dataset of size $n$, and denote $M_0 = \left(\nabla \boldsymbol{f}_{\boldsymbol{W}_0^{lin}}(\boldsymbol{x}_1), \cdots, \nabla \boldsymbol{f}_{\boldsymbol{W}_0^{lin}}(\boldsymbol{x}_n)\right)^{\top}$ as the NTK feature matrix at initialization. Then running Langevin diffusion (4) on $\mathcal{L}_0^{lin}(\boldsymbol{W})$ with time $T$ and initialization vector $\boldsymbol{W}_0^{lin}$ satisfies*

$$\mathbb{E}[\mathcal{L}_0^{lin}(\bar{\boldsymbol{W}}_T^{lin})] - \min_{\boldsymbol{W}} \mathcal{L}_0^{lin}(\boldsymbol{W}; \mathcal{D}) \leq \alpha + \frac{R}{2T} + \frac{1}{2}\sigma^2 rank(M_0),$$

*where the expectation is over Brownian motion $B_T$ in Langevin diffusion Eq. (4), $\bar{\boldsymbol{W}}_T^{lin} = \frac{1}{T}\int \bar{\boldsymbol{W}}_t^{lin} dt$ is the average of all iterates, and $R = \|\boldsymbol{W}_0^{lin} - \boldsymbol{W}_0^*\|_{M_0}^2$ is the gap between initialization parameters $\boldsymbol{W}_0^{lin}$ and solution $\boldsymbol{W}_0^*$.*

*Proof.* Our proofs are heavily based on the idea in [45, Theorem 3.1] to work only in the parameter space spanned by the input feature vectors. And our proof serves as an extension of their bound to the continuous-time Langevin diffusion algorithm. We begin by using convexity of the empirical loss function $\mathcal{L}_0^{lin}(\boldsymbol{W}; \mathcal{D})$ for linearized network to prove the following standard results

$$\mathcal{L}_0^{lin}(\bar{\boldsymbol{W}}_T^{lin}; \mathcal{D}) - \mathcal{L}_0^{lin}(\boldsymbol{W}_0^*; \mathcal{D}) \leq \langle \bar{\boldsymbol{W}}_T^{lin} - \boldsymbol{W}_0^*, \nabla \mathcal{L}_0^{lin}(\bar{\boldsymbol{W}}_T^{lin}; \mathcal{D})\rangle. \tag{46}$$

Denote $M_0 = \left(\nabla \boldsymbol{f}_{\boldsymbol{W}_0^{lin}}(\boldsymbol{x}_1) \quad \cdots \quad \nabla \boldsymbol{f}_{\boldsymbol{W}_0^{lin}}(\boldsymbol{x}_n)\right)$ and compute the gradient under cross entropy loss and linearized network, we have $\nabla \mathcal{L}_0^{lin}(\boldsymbol{W}_T^{lin}; \mathcal{D})$ lies in the column space of $M_0$. Denote $\Pi_{M_0}$ as the projection operator to the column space of $M_0$, then (46) can be rewritten as

$$\mathcal{L}_0^{lin}(\bar{\boldsymbol{W}}_T^{lin}; \mathcal{D}) - \mathcal{L}_0^{lin}(\boldsymbol{W}_0^*; \mathcal{D}) \leq \langle \Pi_{M_0}(\bar{\boldsymbol{W}}_T^{lin} - \boldsymbol{W}_0^*), \nabla \mathcal{L}_0^{lin}(\bar{\boldsymbol{W}}_T^{lin}; \mathcal{D})\rangle. \tag{47}$$

By taking expectation over the randomness of Brownian motion in Langevin diffusion, we have

$$\mathbb{E}[\mathcal{L}_0^{lin}(\bar{\boldsymbol{W}}_T^{lin})] - \mathcal{L}_0^{lin}(\boldsymbol{W}_0^*; \mathcal{D}) \leq \frac{1}{T}\int_0^T \mathbb{E}\left[\langle \Pi_{M_0}(\boldsymbol{W}_t^{lin} - \boldsymbol{W}_0^*), \nabla \mathcal{L}_0^{lin}(\boldsymbol{W}_t^{lin}; \mathcal{D})\rangle\right] dt. \tag{48}$$

We now rewrite $\mathbb{E}\left[\langle \Pi_{M_0}(\bar{\boldsymbol{W}}_t^{lin} - \boldsymbol{W}_0^*), \nabla \mathcal{L}_0^{lin}(\boldsymbol{W}_t^{lin}; \mathcal{D})\rangle\right]$ by computing $\frac{\partial}{\partial t}\mathbb{E}[\|\boldsymbol{W}_t^{lin} - \boldsymbol{W}_0^*\|_{M_0}^2]$, where $\|\boldsymbol{W}_t^{lin} - \boldsymbol{W}_0^*\|_{M_0}^2 = \Pi_{M_0}\left(\boldsymbol{W}_t^{lin} - \boldsymbol{W}_0^*\right)^{\top}\Pi_{M_0}\left(\boldsymbol{W}_t^{lin} - \boldsymbol{W}_0^*\right)$. By applying (24) with $p_t$ being the distribution for $\boldsymbol{W}_t^{lin}$ in Langevin diffusion for linearized network starting from point initialization $\boldsymbol{W}_0^{lin}$, and with function $d(\boldsymbol{W}) = \|\boldsymbol{W} - \boldsymbol{W}_0^*\|_{M_0}^2$, we have that

$$\frac{\partial}{\partial t}\mathbb{E}[\|\boldsymbol{W}_t^{lin} - \boldsymbol{W}_0^*\|_{M_0}^2] \leq -2\mathbb{E}[\langle \Pi_{M_0}\left(\boldsymbol{W}_t^{lin} - \boldsymbol{W}_0^*\right), \nabla \mathcal{L}_0^{lin}(\boldsymbol{W}_t^{lin}; \mathcal{D})\rangle] + \sigma^2 rank(M_0). \tag{49}$$

Therefore by plugging (49) into (48), we have that

$$\mathbb{E}[\mathcal{L}_0^{lin}(\bar{\boldsymbol{W}}_T^{lin})] - \mathcal{L}_0^{lin}(\boldsymbol{W}_0^*; \mathcal{D}) \leq -\frac{1}{2T}\int_0^T \frac{\partial}{\partial t}\mathbb{E}[\|\boldsymbol{W}_t - \boldsymbol{W}_0^*\|_{M_0}^2] dt + \frac{1}{2}\sigma^2 rank(M_0) \tag{50}$$

$$\leq \frac{1}{2T}\|\boldsymbol{W}_0^{lin} - \boldsymbol{W}_0^*\|_{M_0}^2 + \frac{1}{2}\sigma^2 rank(M_0) \tag{51}$$

$\square$

## D.2 Deferred proof for Lemma 6.3

To bound lazy training distance of training linearized network, we would need the following auxiliary Lemma about high probability upper bound for the final layer output of linearized network at initialization.

**Lemma D.1.** *Fix any data record $\boldsymbol{x}$, then with high probability $1 - \delta$ over random initialization Eq. (5) of model weight matrices $W^1, \cdots, W^L$ for layer $1, \cdots, L$, i.e., $W_l \sim \mathcal{N}(0, \beta_l \mathbb{I})$, it satisfies that*

$$\|\boldsymbol{f_W}(\boldsymbol{x})\|_2 \leq \|\boldsymbol{x}\|_2^2 \tilde{\mathcal{O}} \left( \beta_L \left( \prod_{i=1}^{L-1} \beta_i m_i \right) \right) \tag{52}$$

*where $\tilde{\mathcal{O}}$ ignores logarithmic terms with regard to width $m$, depth $L$ and tail probability $\delta$.*

*Proof.* To bound the term $\|\boldsymbol{f_{W_0^{lin}}}(\boldsymbol{x}_i)\|_2^2$, by definition Eq. (2), for any $\boldsymbol{x}$, we have that

$$\|\boldsymbol{f_{W_0^{lin}}}(\boldsymbol{x})\|_2^2 = \frac{\|h_L(\boldsymbol{x})\|_2^2}{\|h_{L-1}(\boldsymbol{x})\|_2^2} \cdots \frac{\|h_1(\boldsymbol{x})\|_2^2}{\|h_0(\boldsymbol{x})\|_2^2} \cdot \frac{\|h_0(\boldsymbol{x})\|_2^2}{\|\boldsymbol{x}\|_2^2} \|\boldsymbol{x}\|_2^2 \tag{53}$$

We now bound the terms in the right-hand-side of Eq. (53) one by one.

Regarding the first term $\frac{\|h_L(\boldsymbol{x})\|_2^2}{\|h_{L-1}(\boldsymbol{x})\|_2^2}$ in Eq. (53), observe that by the network output definition Eq. (2), we have that

$$\|h_L(\boldsymbol{x})\|_2^2 = \|\boldsymbol{W}^L h_{L-1}(\boldsymbol{x})\|_2^2 \stackrel{d}{=} \beta_L \|h_{L-1}(\boldsymbol{x})\|_2^2 \tilde{w}^2 \tag{54}$$

where $\tilde{w} \sim \mathcal{N}(0, 1)$ and the last equality is by rotaional invariance of Gaussian distribution used for initializing $L$-th layer weight matrix $\boldsymbol{W}^L \in \mathbb{R}^{1 \times m}$. Therefore, by tail probabilty expression for standard Gaussian random variable, we have that with high probability $1 - \frac{\delta}{L}$ over random initialization of $\boldsymbol{W}^L \in \mathbb{R}^{1 \times m}$, it satisfies that

$$\frac{\|h_L(\boldsymbol{x})\|_2^2}{\|h_{L-1}(\boldsymbol{x})\|_2^2} \leq 2\beta_L \log \frac{L}{\delta} \tag{55}$$

Regarding the terms $\frac{\|h_l(\boldsymbol{x})\|_2^2}{\|h_{l-1}(\boldsymbol{x})\|_2^2}$ in Eq. (53) for layer $l = 1, \cdots, L-1$, by setting $\boldsymbol{H}_1 = \boldsymbol{h}_2 = h_{l-1}$ in Eq. (32), we immediately prove that over random initialization of weight matrix $\boldsymbol{W}^l \in \mathbb{R}^{m_l \times m_{l-1}}$, it satisfies that

$$\frac{\|h_l(\boldsymbol{x})\|_2^2}{\|h_{l-1}(\boldsymbol{x})\|_2^2} \stackrel{d}{=} \sum_{i=1}^{m_l} \rho_i \beta_l \tilde{w}_i^2 \tag{56}$$

where $\rho_1, \cdots, \rho_{m_l} \stackrel{i.i.d.}{\sim} \text{Ber}(1, \frac{1}{2})$ and $\tilde{w}_1, \cdots, \tilde{w}_{m_l} \stackrel{i.i.d.}{\sim} \mathcal{N}(0, 1)$ and $\rho_i$ and $\tilde{w}_i$ are independent.

By tail probabilty expression for Gaussian random variable $\tilde{w}_i$, we have that $P(\tilde{w}_i^2 \geq t) \leq e^{-t/2}$ for any $t > 0$. By union bound over $i = 1, \cdots, m_l$, we prove that for any layer $l = 1, \cdots, L-1$, with high probability $1 - \frac{\delta}{2L}$ over random initialization of weight matrix $\boldsymbol{W}^l \in \mathbb{R}^{m_l \times m_{l-1}}$, it satisfies that

$$\max_i \tilde{w}_i^2 \leq 2 \log \frac{2m_l L}{\delta} \tag{57}$$

Moreover, by applying Hoeffding's inequality to i.i.d. Bernoulli r.v.s $\rho_1, \cdots, \rho_m$, we prove with high probability $1 - \frac{\delta}{2L}$, it satisfies that $\sum_{i=1}^m \rho_i \leq \frac{m_l}{2}(1 + \log(2L/\delta))$. By combining it with Eq. (57) via union bound, and plugging the result into Eq. (59), we prove for any $l = 1, \cdots, L-1$, it satisfies with high probability $1 - \frac{\delta}{L}$ over random initialization of weight matrix $\boldsymbol{W}^l \in \mathbb{R}^{m_l \times m_{l-1}}$ that

$$\frac{\|h_l(\boldsymbol{x})\|_2^2}{\|h_{l-1}(\boldsymbol{x})\|_2^2} = m_l \beta_l \log \frac{2m_l L}{\delta} \cdot (1 + \log(2L/\delta)) \tag{58}$$

By using union bound over Eq. (58) for layer $l = 1, \cdots, L - 1$ and Eq. (55), we have that with high probability $1 - \delta$ over random initialization Eq. (5), it satisfies that

$$\frac{\|h_L(\boldsymbol{x})\|_2^2}{\|h_0(\boldsymbol{x})\|_2^2} \le \tilde{\mathcal{O}}(\prod_{i=1}^{L-1} \beta_i m_i) \tag{59}$$

where $\tilde{\mathcal{O}}$ ignores logarithmic factors with regard to $m$, $L$ and $\delta$.

By plugging Eq. (59) into Eq. (53), we prove that with high probability $1 - \delta$ over initialization Eq. (5), the following bound holds.

$$\|\boldsymbol{f_W}(\boldsymbol{x})\|_2 \le \|\boldsymbol{x}\|_2^2 \tilde{\mathcal{O}} \left( \beta_L \left( \prod_{i=1}^{L-1} \beta_i m_i \right) \right), \tag{60}$$

where $\tilde{\mathcal{O}}$ ignores logarithmic terms with regard to width $m$, depth $L$ and tail probability $\delta$. $\qquad \square$

**Lemma 6.3.** (Bounding lazy training distance via smallest eigenvalue of the NTK matrix) *Under Assumption 2.4 and single-output linearized network Eq. (3) with $o = 1$, assume that the per-layer network widths $m_0, \cdots, m_L = \tilde{\Omega}(n)$ are large. Let $\mathcal{L}_0^{lin}(\boldsymbol{W})$ be the empirical risk Eq. (1) for linearized network expanded at initialization vector $\boldsymbol{W}_0^{lin}$. Then for any $\boldsymbol{W}_0^{lin}$, there exists a corresponding solution $\boldsymbol{W}_0^{\frac{1}{n^2}}$, s.t. $\mathcal{L}_0^{lin}(\boldsymbol{W}_0^{\frac{1}{n^2}}) - \min_{\boldsymbol{W}} \mathcal{L}_0^{lin}(\boldsymbol{W}; \mathcal{D}) \le \frac{1}{n^2}$, $rank(M_0) = n$ and*

$$R \le \tilde{\mathcal{O}} \left( \max \left\{ \frac{1}{d\beta_L \left( \prod_{i=1}^{L-1} \beta_i m_i \right)}, 1 \right\} \frac{n}{\sum_{l=1}^{L} \beta_l^{-1}} \right), \tag{61}$$

*with high probability over training data sampling and random initialization Eq. (5), where $\tilde{\mathcal{O}}$ ignores logarithmic factors with regard to $n$, $m$, $L$, and tail probability $\delta$.*

*Proof.* Given an arbitrary initialization parameter vector $\boldsymbol{W}_0^{lin}$, we first construct an solution $\boldsymbol{W}_0^{\frac{1}{n^2}}$ that is nearly optimal for the ERM problem over $\mathcal{L}_0^{lin}(\boldsymbol{W})$, as follows.

$$\boldsymbol{W}_0^{\frac{1}{n^2}} - \boldsymbol{W}_0^{lin} = M_0^{\dagger} \begin{pmatrix} 2 \ln n \cdot y_1 - \boldsymbol{f_{W_0^{lin}}}(\boldsymbol{x}_1) \\ \vdots \\ 2 \ln n \cdot y_n - \boldsymbol{f_{W_0^{lin}}}(\boldsymbol{x}_n) \end{pmatrix} \tag{62}$$

where $M_0 = \begin{pmatrix} \nabla \boldsymbol{f_{W_0^{lin}}}(\boldsymbol{x}_1)^{\top} \\ \vdots \\ \nabla \boldsymbol{f_{W_0^{lin}}}(\boldsymbol{x}_n)^{\top} \end{pmatrix}$ is the gradient matrix at initialization and $\dagger$ denotes pseudo-inverse.

We now prove that the solution $\boldsymbol{W}_0^{\frac{1}{n^2}}$ is close to the initialization parameters $\boldsymbol{W}_0^{lin}$ in $\ell_2$ distance with high probability. By applying the holder inequality in (62), we have that

$$R = \|\boldsymbol{W}_0^{\frac{1}{n^2}} - \boldsymbol{W}_0^{lin}\|_2^2 \le \|M_0^{\dagger}\|_2^2 \cdot \left\| \begin{pmatrix} 2 \ln n \cdot y_1 - \boldsymbol{f_{W_0^{lin}}}(\boldsymbol{x}_1) \\ \vdots \\ 2 \ln n \cdot y_n - \boldsymbol{f_{W_0^{lin}}}(\boldsymbol{x}_n) \end{pmatrix} \right\|_2^2 \tag{63}$$

$$\le \frac{1}{\lambda_0} \cdot \left\| \begin{pmatrix} 2 \ln n \cdot y_1 - \boldsymbol{f_{W_0^{lin}}}(\boldsymbol{x}_1) \\ \vdots \\ 2 \ln n \cdot y_n - \boldsymbol{f_{W_0^{lin}}}(\boldsymbol{x}_n) \end{pmatrix} \right\|_2^2 \tag{64}$$

where $\lambda_0$ is the smallest non-zero eigenvalue of the PSD matrix $M_0 M_0^{\top}$. When the data regularity assumption Assumption 2.4 holds and the per-layer width $m_0, \cdots, m_L = \tilde{\Omega}(n)$, by applying existing bounds for the smallest eigenvalue of the NTK matrix $M_0 M_0^{\top}$ for single-output network in [40,

Theorem 4.1], we prove that with high probability over data sampling and random initialization Eq. (5), it satisfies that

$$O\left(\left(d\prod_{l=1}^{L-1}m_l\right)\cdot\left(\prod_{l=1}^{L}\beta_l\right)\cdot\left(\sum_{l=1}^{L}\beta_l^{-1}\right)\right)\geq\lambda_0$$

$$\geq\Omega\left(\left(d\prod_{l=1}^{L-1}m_l\right)\cdot\left(\prod_{l=1}^{L}\beta_l\right)\cdot\left(\sum_{l=1}^{L}\beta_l^{-1}\right)\right),\tag{65}$$

where we have set the auxiliary variables $\xi_l=1$ in [40, Theorem 4.1] because the per-layer width $m_l=\tilde{\Omega}(n)$ are large enough for $l=0,\cdots,L$ (where $n$ is the number of training data).

Therefore, by using Eq. (65), we have that with high probability over data sampling and random initialization Eq. (5), it satisfies that

$$\frac{1}{\lambda_0}\leq O\left(\frac{1}{\left(d\prod_{l=1}^{L-1}m_l\right)\cdot\left(\prod_{l=1}^{L}\beta_l\right)\cdot\left(\sum_{l=1}^{L}\beta_l^{-1}\right)}\right),\tag{66}$$

where $m_l$ is the width of layer $l$ and $n$ is the number of training data.

For the second term in Eq. (64), by Cauchy-Schwarz inequality, we have that

$$\left\|\begin{pmatrix}2\ln n\cdot y_1-\boldsymbol{f}_{\boldsymbol{W}_0^{lin}}(\boldsymbol{x}_1)\\\vdots\\2\ln n\cdot y_n-\boldsymbol{f}_{\boldsymbol{W}_0^{lin}}(\boldsymbol{x}_n)\end{pmatrix}\right\|_2^2\leq 2\cdot(2\ln n)^2\sum_{i=1}^{n}y_i^2+2\sum_{i=1}^{n}\|\boldsymbol{f}_{\boldsymbol{W}_0^{lin}}(\boldsymbol{x}_i)\|_2^2\tag{67}$$

By Lemma D.1, we have that $\|\boldsymbol{f}_{\boldsymbol{W}_0^{lin}}(\boldsymbol{x})\|_2\leq\tilde{\mathcal{O}}\left(d\cdot\beta_L\left(\prod_{i=1}^{L-1}\beta_im_i\right)\right)$ with high probability over the random initialization Eq. (5). By plugging this result and $y_i\in\{-1,1\}$ into Eq. (67), we have that with high probability over random initialization Eq. (5), it satisfies that

$$\left\|\begin{pmatrix}2\ln n\cdot y_1-\boldsymbol{f}_{\boldsymbol{W}_0^{lin}}(\boldsymbol{x}_1)\\\vdots\\2\ln n\cdot y_n-\boldsymbol{f}_{\boldsymbol{W}_0^{lin}}(\boldsymbol{x}_n)\end{pmatrix}\right\|_2^2=\tilde{\mathcal{O}}\left(n+nd\beta_L\left(\prod_{i=1}^{L-1}\beta_im_i\right)\right),\tag{68}$$

where $\tilde{O}$ ingores logarithmic factors with regard to $n$, $m$, $L$, and tail probability $\delta$. Therefore, by combining Eq. (66) and Eq. (68) with union bound, and by plugging the result into Eq. (64), we have that with high probability over data sampling and random initialization Eq. (5), it satisfies that

$$R\leq\tilde{\mathcal{O}}\left(\frac{n+nd\beta_L\left(\prod_{i=1}^{L-1}\beta_im_i\right)}{\left(d\beta_L\prod_{l=1}^{L-1}\beta_lm_l\right)\cdot\left(\sum_{l=1}^{L}\beta_l^{-1}\right)}\right)$$

$$\leq\tilde{\mathcal{O}}\left(\max\left\{\frac{1}{d\beta_L\left(\prod_{i=1}^{L-1}\beta_im_i\right)},1\right\}\frac{n}{\sum_{l=1}^{L}\beta_l^{-1}}\right).$$

where $m_l$ is the width of layer $l$, $n$ is the number of training data, and $\tilde{\mathcal{O}}$ ignores logarithmic factors with regard to $n$, $m$, $L$, and tail probability $\delta$.

We finally prove that $\boldsymbol{W}_0^{\frac{1}{n^2}}$ is a $\frac{1}{n^2}$-near-optimal solution. Note that Eq. (65) implies that with high probability $M_0M_0^{\top}$ is full rank, i.e., $\text{rank}(M_0M_0^{\top})=n$. Therefore $M_0^{\dagger}=M_0^{\top}(M_0M_0^{\top})^{-1}$ and $\begin{pmatrix}\boldsymbol{f}_{\boldsymbol{W}_0^{\frac{1}{n^2}}}(\boldsymbol{x}_1)\\\vdots\\\boldsymbol{f}_{\boldsymbol{W}_0^{\frac{1}{n^2}}}(\boldsymbol{x}_n)\end{pmatrix}=\begin{pmatrix}2\ln n\cdot y_1\\\vdots\\2\ln n\cdot y_n\end{pmatrix}$ with high probability over the training data sampling and random initialization Eq. (5). By plugging it into the cross-entropy loss for the single-output network

defined below Eq. (1), we have that with high probability over the training data sampling and random initialization Eq. (5), the solution $\boldsymbol{W}_0^{\frac{1}{n^2}}$ satisfies

$$\mathcal{L}_0^{lin}(\boldsymbol{W}_0^{\frac{1}{n^2}}) - \min_{\boldsymbol{W}} \mathcal{L}_0^{lin}(\boldsymbol{W};\mathcal{D}) \le \log(1 + \exp(-2\ln n)) < \frac{1}{n^2}. \tag{69}$$

$\square$

### D.3  Deferred proof for Corollary 6.4

**Corollary 6.4.** (Privacy utility trade-off for linearized network) *Assume that all conditions in Lemma 6.3 holds. Let $B = d\left(\prod_{i=1}^{L-1} \frac{\beta_i m_i}{2}\right)\sum_{l=1}^{L} \frac{\beta_L}{\beta_l}$ be the gradient norm constant proved in Eq. (7), and let $R \le \tilde{\mathcal{O}}\left(\max\left\{\frac{1}{d\beta_L\left(\prod_{i=1}^{L-1} \beta_i m_i\right)}, 1\right\} \frac{n}{\sum_{l=1}^{L} \beta_l^{-1}}\right)$ be the lazy training distance bound proved in Lemma 6.3. Then for $\sigma^2 = \frac{2BT}{\varepsilon n^2}$ and $T = \sqrt{\frac{\varepsilon n R}{2B}}$, releasing all iterates of Langevin diffusion with time $T$ satisfies $\varepsilon$-KL privacy, and has empirical excess risk upper bounded by*

$$\mathbb{E}[\mathcal{L}_0^{lin}(\bar{\boldsymbol{W}}_T^{lin})] - \min_{\boldsymbol{W}} \mathcal{L}_0^{lin}(\boldsymbol{W};\mathcal{D}) \le \tilde{\mathcal{O}}\left(\frac{1}{n^2} + \sqrt{\frac{BR}{\varepsilon n}}\right) \tag{70}$$

$$= \tilde{\mathcal{O}}\left(\frac{1}{n^2} + \sqrt{\frac{1}{2^{L-1}\varepsilon}\max\{1, d\beta_L \prod_{l=1}^{L-1} \beta_l m_l\}}\right) \tag{71}$$

*with high probability over random initiailization Eq. (5), where the expectation is over Brownian motion $B_T$ in Langevin diffusion, and $\tilde{\mathcal{O}}$ ignores logarithmic factors with regard to width $m$, depth $L$, number of training data $n$ and tail probability $\delta$. A summary of our excess empirical risk bounds under different initializations is in Table 1.*

*Proof.* By setting $\boldsymbol{W}_0^* = \boldsymbol{W}_0^{\frac{1}{n^2}}$ in Lemma 6.1 with $\boldsymbol{W}_0^{\frac{1}{n^2}}$ constructed as Lemma 6.3, we have with high probability over random initialization Eq. (5), we have

$$\mathbb{E}[\mathcal{L}_0^{lin}(\bar{\boldsymbol{W}}_T^{lin})] - \min_{\boldsymbol{W}} \mathcal{L}_0^{lin}(\boldsymbol{W};\mathcal{D}) \le \frac{1}{n^2} + \frac{R}{2T} + \frac{\sigma^2 n}{2}. \tag{72}$$

where $R \le \tilde{\mathcal{O}}\left(\max\left\{\frac{1}{d\beta_L\left(\prod_{i=1}^{L-1} \beta_i m_i\right)}, 1\right\} \frac{n}{\sum_{l=1}^{L} \beta_l^{-1}}\right)$ by Lemma 6.3.

Meanwhile, to ensure $\varepsilon$-KL privacy, by Corollary 4.2, we only need to set $\sigma^2 = \frac{2BT}{\varepsilon n^2}$ where $B = d\left(\prod_{i=1}^{L-1} \frac{\beta_i m_i}{2}\right)\sum_{l=1}^{L} \frac{\beta_L}{\beta_l}$ by for single-output network with $o = 1$. By plugging $\sigma^2 = \frac{2BT}{\varepsilon n^2}$ into (72), we prove that

$$\mathbb{E}[\mathcal{L}_0^{lin}(\bar{\boldsymbol{W}}_T^{lin})] - \min_{\boldsymbol{W}} \mathcal{L}_0^{lin}(\boldsymbol{W};\mathcal{D}) \le \frac{1}{n^2} + \frac{R}{2T} + \frac{BT}{\varepsilon n}. \tag{73}$$

Setting $T = \sqrt{\frac{\varepsilon n R}{2B}}$ in (73) and elaborating the computations suffice to prove the result. $\square$

## E  Discussion on extending our results to Noisy GD with constant step-size

In this section, we discuss how to extend our privacy analyses to noisy GD with constant step-size. Specifically, we only need to extend the KL composition theorem under possibly unbounded gradient difference, i.e., Theorem 3.1, to the noisy GD algorithm.

**Theorem E.1** (KL composition for noisy GD under possibly unbounded gradient difference)**.** *Let the iterative update in noisy GD algorithm be defined by:* $\boldsymbol{W}_{(k+1)} = \boldsymbol{W}_{(k)} - \eta\nabla\mathcal{L}(\boldsymbol{W}_{(k)};\mathcal{D}) +$

$\sqrt{2\eta\sigma^2}Z_k$, where $Z_k \sim \mathcal{N}(0, \mathbb{I})$. *Then the KL divergence between running noisy GD for DNN* (2) *on neighboring datasets $\mathcal{D}$ and $\mathcal{D}'$ satisfies*

$$KL(\boldsymbol{W}_{(1:K)}, \boldsymbol{W}'_{(1:K)}) = \frac{1}{2\sigma^2}\sum_{k=0}^{K-1}\eta \cdot \mathbb{E}\left[\left\|\nabla\mathcal{L}(\boldsymbol{W}_{(k)};\mathcal{D}) - \nabla\mathcal{L}(\boldsymbol{W}_{(k)};\mathcal{D}')\right\|_2^2\right]. \qquad (74)$$

*Proof.* Denote $p_{(k)}$ as the distribution of model parameters after running noisy GD on dataset $\mathcal{D}$ with $k$ steps, and similarly denote $p'_{(k)}$ as the distribution of model parameters after running noisy GD on dataset $\mathcal{D}'$ with $k$ steps. Similarly, denote $p_{(1:k)}$ as the joint distribution of $(\boldsymbol{W}_{(1)}, \cdots, \boldsymbol{W}_{(k)})$, and denote $p'_{(1:k)}$ as the joint distribution of $(\boldsymbol{W}'_{(1)}, \cdots, \boldsymbol{W}'_{(k)})$. Now we expand the term $KL(p_{(1,k+1)}, p'_{(1,k+1)})$ by the Bayes rule as follows.

$$KL(p_{(1:k+1)}, p'_{(1:k+1)}) \qquad (75)$$

$$=\mathbb{E}_{p_{(1:k+1)}(\boldsymbol{W}_{(1:k+1)})}\left[\log\left(\frac{p_{(k+1)|(1:k)}(\boldsymbol{W}_{(k+1)}|\boldsymbol{W}_{(1:k)})p_{(1:k)}(\boldsymbol{W}_{(1:k)})}{p'_{(k+1)|(1:k)}(\boldsymbol{W}_{(k+1)}|\boldsymbol{W}_{(1:k)})p'_{(1:k)}(\boldsymbol{W}_{(1:k)})}\right)\right]$$

$$=\mathbb{E}_{p_{(1:k+1)}(\boldsymbol{W}_{(1:k+1)})}\left[\log\left(\frac{p_{(k+1)|(1:k)}(\boldsymbol{W}_{(k+1)}|\boldsymbol{W}_{(1:k)})}{p'_{(k+1)|(1:k)}(\boldsymbol{W}_{(k+1)}|\boldsymbol{W}_{(1:k)})}\right)\right] + \mathbb{E}_{p_{(1:k)}(\boldsymbol{W}_{(1:k)})}\left[\log\left(\frac{p_{(1:k)}(\boldsymbol{W}_{(1:k)})}{p'_{(1:k)}(\boldsymbol{W}_{(1:k)})}\right)\right]$$

$$=\mathbb{E}_{p_{(1:k)}(\boldsymbol{W}_{(1:k)})}\left[KL(p_{(k+1)|(1:k)}, p'_{(1:k+1)|(1:k)})\right] + KL(p_{(1:k)}, p'_{(1:k)}) \qquad (76)$$

Observe that conditioned on fixed model parameters $\boldsymbol{W}_{(1:k)}$ at iteration $1, \cdots, k$, the distributions $p_{(k+1)|(1:k)}, p'_{(k+1)|(1:k)}$ are Gaussian with per-dimensional variance $\sigma^2$. Therefore, by computing the KL divergence between two multivariate Gaussians, we have that

$$KL(p_{(k+1)|(1:k)}, p'_{(k+1)|(1:k)}) = \frac{1}{2\sigma^2} \cdot \eta \cdot \left\|\nabla\mathcal{L}(\boldsymbol{W}_{(k)};\mathcal{D}) - \nabla\mathcal{L}(\boldsymbol{W}_{(k)};\mathcal{D}')\right\|_2^2 \qquad (77)$$

Therefore, by plugging Eq. (77) into Eq. (76), we have that

$$KL(p_{(1:k+1)}, p'_{(1:k+1)}) = \frac{\eta}{2\sigma^2}\mathbb{E}\left[\left\|\nabla\mathcal{L}(\boldsymbol{W}_{(k)};\mathcal{D}) - \nabla\mathcal{L}(\boldsymbol{W}_{(k)};\mathcal{D}')\right\|_2^2\right] + KL(p_{(1:k)}, p'_{(1:k)}) \quad (78)$$

By summing (78) over $k = 0, \cdots, K-1$ and observing that $KL(p_{(0)}, p'_{(0)}) = 0$ (as the initialization distribution is the same between noisy GD on $\mathcal{D}$ and $\mathcal{D}'$), we finish the proof for Eq. (74). $\qquad\square$

