# Initialization Matters: Privacy-Utility Analysis of Overparameterized Neural Networks

## Abstract

We analytically investigate how overparameterization of models in randomized machine learning algorithms impacts the information leakage about their training data. Specifically, we prove a privacy bound for the KL divergence between model distributions on worst-case neighboring datasets, and explore its dependence on the initialization, width, and depth of fully connected neural networks. We find that this KL privacy bound is largely determined by the expected squared gradient norm relative to model parameters during training. Notably, for the special setting of linearized network, our analysis indicates that the squared gradient norm (and therefore the escalation of privacy loss) is tied directly to the per-layer variance of the initialization distribution. By using this analysis, we demonstrate that privacy bound improves with increasing depth under certain initializations (LeCun and Xavier), while degrades with increasing depth under other initializations (He and NTK). Our work reveals a complex interplay between privacy and depth that depends on the chosen initialization distribution. We further prove excess empirical risk bounds under a fixed KL privacy budget, and show that the interplay between privacy utility trade-off and depth is similarly affected by the initialization.

## 1   Introduction

Deep neural networks (DNNs) in the over-parameterized regime (i.e., more parameters than data) perform well in practice but the model predictions can easily leak private information about the training data under inference attacks such as membership inference attacks [38] and reconstruction attacks. [14, 5, 23] This leakage can be mathematically measured in terms of how much the algorithm's output distribution changes if it were trained on a neighboring dataset (that only differs in one record), following the differential privacy (DP) framework [18].

To train differential private model, a typical way is randomize each gradient update in neural networks training, e.g., stochastic gradient descent (SGD), which leads to the most widely applied differentially private training algorithm in the literature – DP-SGD [1]. In each step, DP-SGD employs gradient clipping and adds calibrated Gaussian noise, and thus it comes with a differential privacy guarantee that scales with the noise multiplier (i.e., per-dimensional Gaussian noise standard deviation divided by the clipping threshold) and number of training epochs. However, this privacy bound [1] is overly general due to its independence on the network properties (e.g., width and depth) and training schemes (e.g., initializations). Accordingly, a natural question arises in the community:

*How does overparameterization (e.g., increasing width and depth) of neural networks affect the (worst-case) privacy bound of the training algorithm?*

Submitted to 37th Conference on Neural Information Processing Systems (NeurIPS 2023). Do not distribute.

Table 1: Our results for the privacy utility trade-off of training linearized network (3) via Langevin diffusion, under different width $m$, depth $L$ and initializations. We set per-layer width $m_0 = d$, $m_1, \cdots, m_{L-1} = m$ and $m_L = o$. We prove privacy bound in KL divergence, and obtain excess empirical risk bounds given KL privacy budget $\varepsilon$. For the excess risk bounds, we assume the network width $m = \Omega(n)$ is sufficiently large, and the data and network satisfy regularity assumption Assumption 2.1. For NTK, He and LeCun initialization, we observe that the privacy utility trade-off improves with overparameterization (increasing depth).

| Init | Variance $\beta_l$ for layer $l$ | Gradient norm constant $B$ (9) | Approximate lazy training distance $\tilde{R}$ (12) | Excess Empirical risk under KL privacy bound $\varepsilon$ (Corollary 5.4) |
|---|---|---|---|---|
| LeCun | $1/m_{l-1}$ | $\frac{om(L-1+\frac{d}{m})}{2^{L-1}d}$ | $\tilde{O}(\frac{n}{m(L-1)})$ | $\tilde{O}\left(\frac{1}{n^2} + \sqrt{\frac{1}{2^L d\varepsilon}\left(1+\frac{d}{m(L-1)}\right)}\right)$ |
| He | $2/m_{l-1}$ | $\frac{om(L-1+\frac{d}{m})}{d}$ | $\tilde{O}(\frac{n}{2^L m(L-1)})$ | $\tilde{O}\left(\frac{1}{n^2} + \sqrt{\frac{1}{2^L d\varepsilon}\left(1+\frac{d}{m(L-1)}\right)}\right)$ |
| NTK | $2/m_l, l < L$ $1/o, l = L$ | $\frac{m(L-1)}{2} + o$ | $\tilde{O}(\frac{n}{d\cdot 2^L \cdot (m(L-2)+1)})$ | $\tilde{O}\left(\frac{1}{n^2} + \sqrt{\frac{1}{d\cdot 2^L \varepsilon}\cdot\frac{m(L-1)+2}{m(L-2)+1}}\right)$ |
| Xavier | $\frac{2}{m_{l-1}+m_l}$ | $\frac{L-1+\frac{d+o}{2m}}{2^{L-1}(1+\frac{d}{m})(1+\frac{o}{m})}$ | - | - |

- The Xavier initialization makes neural networks fall into non-lazy training regime [31], so we do not include the lazy training distance nor privacy-utility trade-off analysis here.

To answer this question, we would need new algorithmic framework and (or) new privacy analyses. In this paper, we focus on analyzing privacy for the Langevin diffusion algorithm [1]. This is to avoid artificially setting a sensitivity constraint on the gradient update and thus making the privacy bound insensitive to the network overparameterization (as in DP-SGD analysis). Instead, we prove a KL privacy bound for Langevin diffusion that scales with the expected gradient difference between the training on any two worst-case neighboring datasets (Theorem 3.1). [2] By proving precise upper bounds on the expected $\ell_2$-norm of this gradient difference, we obtain KL privacy bounds for fully connected neural network (Lemma 3.2) and its linearized variant (Corollary 4.2) that changes with the network width, depth and per-layer variance for the initialization distribution. We summarized the details of our KL privacy bounds in Table 1, and highlight our key observations below.

- Width always worsen privacy, under all the considered initialization distributions. Meanwhile, the interplay between network depth and privacy is much more complex and crucially depends on what initialization distribution is used and how long the training time is.

- Specifically, when the initialization distribution has small per-layer variance (such as LeCun and Xavier initialization), our KL privacy bound for training fully connected network (with a small amount of time) and for training linearized network (with finite time) decay exponentially with increasing depth, as long as the depth is large enough. To the best of our knowledge, this is the first time that an improvement of privacy bound under overparameterization is observed for randomized training algorithm.

To further understand how the privacy utility trade-off is affected by overparameterization, we also analyze the excess empirical risk and excess population risk of training linearized network using Langevin diffusion. Our risk bounds scale with the lazy training distance $R$ (i.e., how close is the initialization vector to an optimal solution for the empirical risk minimization problem), as well as a constant $B$ for expected gradient norm in Langevin diffusion. By analyzing these two terms precisely under overparameterization, we prove that given any fixed KL privacy budget $\varepsilon$, our risk bounds strictly improves with increasing depth and width for linearized network under LeCun and He initialization. To our best knowledge, this is the first time that such a gain in privacy-utility trade-off

---

[1] A key difference between this paper and existing privacy utility analysis of Langevin diffusion [21] is that we analyze in the absence of gradient clipping or Lipschitz assumption on loss function. Our results also readily extend to discretized noisy GD with constant step-size (as discussed in Appendix E).

[2] We focus on KL privacy loss because it is a more relaxed distinguishability notion than standard $(\varepsilon, \delta)$-DP, and therefore could be upper bounded even without gradient clipping. Moreover, KL divergence enables upper bound for the advantage (relative success) of various inference attacks, as studied in recent works [32, 22].

61 due to overparameterization (increasing depth) is shown. Meanwhile, prior results only prove (nearly)
62 dimension-independent privacy utility trade-off for such linear models in the literature [39, 26, 30].
63 Our improvement demonstrates the unique benefits of our new KL privacy analysis in understanding
64 the effect of overparameterization.

## 1.1 Related Works

66 ***Overparameterization in DNNs and NTK.*** Theoretical demonstration on the benefit of over-
67 parameterization in DNNs occurs in global convergence [2, 17], generalization [3, 13]. Under
68 proper initialization, the training dynamics of over-parameterized DNNs can be described by a kernel
69 function, termed as neural tangent kernel (NTK) [25], which stimulates a series of analysis in DNNs.
70 Accordingly, over-parameterization has been demonstrated to be beneficial/harmful to several topics
71 in deep learning, e.g., robustness [12, 47], covariate shift [44]. However, the relationship between
72 overparameterization and privacy (based on the differential privacy framework) remains largely an
73 unsolved problem, as the training dynamics typically change [11] after adding new components in
74 the privacy-preserving learning algorithm (such as DP-SGD [1]) to enforce privacy constraints.

75 ***Membership inference privacy risk under overparameterization.*** A recent line of works [42, 43]
76 investigates how overparameterization affects the theoretical and empirical privacy in terms of
77 membership inference advantage, and proves novel trade-off between privacy and generalization error.
78 These are the closest works in the literature to our objective of investigating the interplay between
79 privacy and overparameterization. However, Tan et al. [42, 43] focus on proving upper bounds for
80 an average-case privacy risk defined by the advantage (relative success) of membership inference
81 attack on models trained from randomly sampled training dataset from a population distribution. By
82 contrast, our KL privacy bound is heavily based on the strongest adversary model in the differential
83 privacy definition, and holds under an arbitrary *worst-case* pair of neighboring datasets that only
84 differ in one record. Our setting for model (fully connected network) is also very different from that
85 considered in Tan et al. [42, 43], thus requiring very different analysis tools.

86 ***Differentially private learning in high dimension.*** Standard results for private empirical risk
87 minimization [6, 41] and private stochastic convex optimization [7, 9, 4] under $\ell_1$ and $\ell_2$ constraints
88 suggest that there is an unavoidable factor $d$ in the empirical risk and population risk that depends
89 on the model dimension. However, for unconstrained optimization, it is possible to go across the
90 dimension-dependency in proving risk bounds for certain class of problems (such as generalized
91 linear model [39]). Recently, there is a growing line of works that prove dimension-independent
92 excess risk bounds for differentially private learning, by utilizing the low-rank structure of data
93 features [39] or gradient matrices [26, 30] in training. Several follow-up works [27, 10] further
94 explore techniques to enforce the low-rank property (via random projection) and boost privacy utility
95 trade-off. However, all the works focus on investigating a general high-dimensional problem for
96 private learning, rather than separating the study for different network choices such as structure,
97 width, depth and initializaiton. On the contrary, our study focus on the fully connected neural network
98 and its linearized variant, which enables us to prove more precise privacy utility trade-off bounds for
99 these particular networks under overparameterization.

## 2 Problem and Methodology

101 We consider the following standard multi-class supervised learning setting. Let $\mathcal{D} = (\boldsymbol{z}_1, \cdots, \boldsymbol{z}_n)$ be
102 a finite input dataset of size $n$, where each input data record $\boldsymbol{z}_i = (\boldsymbol{x}_i, \boldsymbol{y}_i)$ contains a $d$-dimensional
103 input feature vector $\boldsymbol{x}_i \in \mathbb{R}^d$ and a label vector $\boldsymbol{y}_i \in \mathcal{Y} = \{0, 1\}^o$ on $o$ possible classes. The goal
104 of learning is to learn a neural network output function $\boldsymbol{f_W}(\cdot) : \mathcal{X} \to \mathcal{Y}$ parameterized by $\boldsymbol{W}$ that
105 achieves high prediction performance on the training dataset $\mathcal{D}$. Formally, we consider the learning
106 objective to be the empirical risk defined as follows.

$$\min_{\boldsymbol{W}} \mathcal{L}(\boldsymbol{W}; \mathcal{D}) := \frac{1}{n} \sum_{i=1}^{n} \ell(\boldsymbol{f_W}(\boldsymbol{x}_i); \boldsymbol{y}_i), \tag{1}$$

107 where $\ell(\boldsymbol{f_W}(\boldsymbol{x}_i); \boldsymbol{y}_i)$ is a loss function that reflects the approximation quality of model predic-
108 tion $\boldsymbol{f_W}(\boldsymbol{x}_i)$ compared to the ground truth label $\boldsymbol{y}_i$. For simplicity, we assume that the cross-

entropy loss is used, i.e., $\ell(\boldsymbol{f}_{\boldsymbol{W}}(\boldsymbol{x}); \boldsymbol{y}) = -\langle \boldsymbol{y}, \log \text{softmax}(\boldsymbol{f}_{\boldsymbol{W}}(\boldsymbol{x})) \rangle$ for multi-output network, and $\ell(\boldsymbol{f}_{\boldsymbol{W}}(\boldsymbol{x}); \boldsymbol{y}) = \log(1 + \exp(-\boldsymbol{y}\boldsymbol{f}_{\boldsymbol{W}}(\boldsymbol{x})))$ for single-output network.

***Network***.  We focus on the *multi-output, fully connected, deep neural network (DNN) with ReLU activation* with depth $L$ (i.e., $L-1$ hidden layers). Denote the width of each hidden layer with $m_1, \cdots, m_{L-1}$. The output function $\boldsymbol{f}_{\boldsymbol{W}}(\boldsymbol{x}) \coloneqq \boldsymbol{h}_L(\boldsymbol{x})$ is defined recursively as follows.

$$\boldsymbol{h}_0(\boldsymbol{x}) = \boldsymbol{x}; \quad \boldsymbol{h}_l(\boldsymbol{x}) = \phi(\boldsymbol{W}_l \boldsymbol{x}) \text{ for } l = 1, \cdots, L-1; \quad \boldsymbol{h}_L(\boldsymbol{x}) = \boldsymbol{W}_L \boldsymbol{h}_{L-1}(\boldsymbol{x}) \qquad (2)$$

where $h_l(\boldsymbol{x})$ denotes the output after activation at the $l$-th layer, the parameter matrices of each layer of the neural network satisfy $\boldsymbol{W}_1 \in \mathbb{R}^{m_1 \times d}$, $\boldsymbol{W}_l \in \mathbb{R}^{m_l \times m_{l-1}}$, $l = 2, \ldots, L-1$ and $\boldsymbol{W}_L \in \mathbb{R}^{o \times m_{L-1}}$. The model parameter $\boldsymbol{W} \coloneqq (\text{Vec}(\boldsymbol{W}_1), \ldots, \text{Vec}(\boldsymbol{W}_L)) \in \mathbb{R}^{m_1 \cdot d + m_2 \cdot m_1 + \cdots + o \cdot m_{L-1}}$ consists of concatenation of vectorizations for parameters of all the layers. For consistency, we also denote $m_0 = d$ and $m_L = o$.

We also analyze the following ***linearized network***, which is used in prior works [28, 2, 34] as an important tool to (approximately and qualitatively) analyze the training dynamics of deep neural networks. More formally, the linearized network $\boldsymbol{f}_{\boldsymbol{W}}^{lin,0}(\boldsymbol{x})$ is a first-order Taylor expansion of the fully connected ReLU network at initialization parameter $\boldsymbol{W}_0^{lin}$, as follows.

$$\boldsymbol{f}_{\boldsymbol{W}}^{lin,0}(\boldsymbol{x}) \equiv \boldsymbol{f}_{\boldsymbol{W}_0^{lin}}(\boldsymbol{x}) + \frac{\partial \boldsymbol{f}_{\boldsymbol{W}}(\boldsymbol{x})}{\partial \boldsymbol{W}}\Big|_{\boldsymbol{W} = \boldsymbol{W}_0^{lin}} \left(\boldsymbol{W} - \boldsymbol{W}_0^{lin}\right), \qquad (3)$$

where $\boldsymbol{f}_{\boldsymbol{W}_0^{lin}}(\boldsymbol{x})$ is the output function of the fully connected ReLU network (2) at initialization $\boldsymbol{W}_0^{lin}$. We denote $\mathcal{L}_0^{lin}(\boldsymbol{W}; \mathcal{D}) = \sum_{i=1}^n \ell\left(\boldsymbol{f}_{\boldsymbol{W}_0^{lin}}(\boldsymbol{x}_i) + \frac{\partial \boldsymbol{f}_{\boldsymbol{W}}(\boldsymbol{x})}{\partial \boldsymbol{W}}|_{\boldsymbol{W} = \boldsymbol{W}_0^{lin}} (\boldsymbol{W} - \boldsymbol{W}_0^{lin}); \boldsymbol{y}_i\right)$ as the empirical loss function for training linearized network, by plugging (3) into (1).

***Langevin Diffusion***.   In terms of optimization algorithm, we focus on the *Langevin diffusion* algorithm [29] with per-dimensional noise variance $\sigma^2$. Note that we aim to *avoid gradient clipping* while still proving KL privacy bounds. After initializing the model parameters $\boldsymbol{W}_0$ at time zero, the model parameters $\boldsymbol{W}_t$ at subsequent time $t$ evolves as the below stochastic differential equation.

$$d\boldsymbol{W}_t = -\nabla \mathcal{L}(\boldsymbol{W}_t; \mathcal{D})dt + \sqrt{2\sigma^2}d\boldsymbol{B}_t. \qquad (4)$$

***Initialization Distribution***.  The initialization of parameters $\boldsymbol{W}_0$ crucially affects the convergence of Langevin diffusion, as observed in prior literatures [46, 20, 19]. Moreover, when the network function depends on the initialization parameters (as in linearized network (3)), the stationary distribution of Langevin diffusion also depends on the initialization distribution (as discussed in Section 5). In this work, we investigate the following general class of Gaussian initialization distribution with (possibly depth-dependent) variance for the parameters in each layer. For any layer $l = 1, \cdots, L$, we have that

$$[\boldsymbol{W}^l]_{ij} \sim \mathcal{N}(0, \beta_l), \text{ for } (i, j) \in [m_l] \times [m_{l-1}] \qquad (5)$$

where $\beta_1, \cdots, \beta_L > 0$ are the per-layer variance for Gaussian initialization. By choosing different variance, we recover many common initialization schemes in the literature, as summarized in Table 1.

## 2.1   Our objective and methodology

We aim to understand the relation between privacy, utility and over-parameterization (depth and width) for the Langevin diffusion algorithm (under different initializaiton distributions). To understand privacy, we prove a KL divergence upper bound for running Langevin diffusion on any two *worst-case* neighboring datasets $\mathcal{D}$ and $\mathcal{D}'$ of size $n$ that only differ in one record, denoted as $(\boldsymbol{x}, \boldsymbol{y}) \in \mathcal{D}$ and $(\boldsymbol{x}, \boldsymbol{y}') \in \mathcal{D}'$. For brevity, in later sections, we denote $\boldsymbol{W}_t$ (with distribution $p_t$) and $\boldsymbol{W}_t'$ (with distribution $p_t'$) as the trained model parameters after running Langevin diffusion (4) for time $T$ on $\mathcal{D}$ and $\mathcal{D}'$ respectively. We make the following assumptions for privacy analysis in this paper.

**Assumption 2.1** (Bounded Data)**.**  We assume that all $\boldsymbol{x}$ in the data domain is bounded s.t. $\|\boldsymbol{x}\|_2 \leq 1$.

To understand utility (under a given KL privacy budget), we aim to prove upper bounds for excess empirical risk and excess population risk given an arbitrarily fixed KL divergence privacy budget $\varepsilon$. Finally, we also investigate how trade-off between our KL privacy bound and risk bounds is affected by the network width and depth. For utility analysis, we additionally make the following fair and attainable assumptions on data and network regularity.

152 **Assumption 2.2** (Data and network regularity [33, Assumption 2.1]). For any training data $\boldsymbol{x}_i \in \mathcal{D}$,
153 it satisfies that $\|\boldsymbol{x}_i\|_2 = 1$. Moreover, $\boldsymbol{x}_i \in \mathcal{D}$ are i.i.d. samples from a data distribution $P_x$ that
154 satisfies $\int \|\boldsymbol{x}\|_2^2 dP_x(\boldsymbol{x}) = 1$. We also assume that the network only has single output.

155 Note that Assumption 2.2 is only required for utility analysis, and is not need for our privacy bound.

# 3 KL Privacy for Training Fully Connected ReLU Neural Networks

157 In this section, we perform the composition-based privacy analysis in KL divergence for Langevin
158 Diffusion on deep ReLU neural networks, under Gaussian initialization distribution specified by
159 Eq. (5). More specifically, we prove upper bound for the KL divergence between distribution of output
160 model parameters when running Langevin diffusion on an arbitrary pair of neighboring datasets $\mathcal{D}$
161 and $\mathcal{D}'$.

162 Our first key insight is that by the joint convexity of KL divergence, it is possible to prove composition-
163 based KL privacy bound under more relaxed condition regarding the sensitivity of gradient computa-
164 tion (i.e., *without* gradient clipping).

165 **Theorem 3.1** (KL composition under possibly unbounded gradient difference). *The KL divergence*
166 *between running Langevin diffusion* (4) *for DNN* (2) *on neighboring datasets $\mathcal{D}$ and $\mathcal{D}'$ satisfies*

$$KL(\boldsymbol{W}_T, \boldsymbol{W}_T') \leq \frac{1}{2\sigma^2} \int_0^T \mathbb{E}\left[\|\nabla\mathcal{L}(\boldsymbol{W}_t; \mathcal{D}) - \nabla\mathcal{L}(\boldsymbol{W}_t; \mathcal{D}')\|_2^2\right] dt. \tag{6}$$

167 ***Proof sketch***. We compute the partial derivative of KL divergence with regard to time $t$, and then
168 compute integral to bound the KL divergence. During computing the limit of differentiation, we
169 upper bound KL divergence at time $t + \eta$ for small enough step-size with the divergence on path
170 $[t, t+\eta]$. Then we use Girsanov's theorem to compute the KL divergence between the path of coupled
171 Langevin diffusion processes. The complete proof is in Appendix B.1.

172 Theorem 3.1 is an extension of the standard additivity [45] of KL divergence (also known as chain
173 rule [40]) for a finite sequence of distributions to continuous time processes with (possibly) unbounded
174 drift difference. The key extension is that Theorem 3.1 does not require bounded sensitivity between
175 Langevin Diffusion on neighboring datasets. Instead, it only requires finite second-order moment of
176 drift difference (in the $\ell_2$-norm sense) between neighboring datasets $\mathcal{D}, \mathcal{D}'$. By using this extended
177 KL composition Theorem 3.1, we prove KL privacy bound for running Langevin diffusion algorithm
178 (without gradient clipping) on deep neural networks, by tracking the upper bound for $\ell_2$ norm of the
179 gradient difference throughout training (under mild assumptions) as follows.

180 **Lemma 3.2** (Drift Difference in Noisy Training). *Let $M_T$ be the subspace spanned by gradients*
181 $\{\nabla\ell(\boldsymbol{f}_{\boldsymbol{W}_t}(\boldsymbol{x}_i; \boldsymbol{y}_i) : (\boldsymbol{x}_i, \boldsymbol{y}_i) \in \mathcal{D}, t \in [0, T]\}$ *on each training data record throughout Langevin*
182 *diffusion* $(\boldsymbol{W}_t)_{t\in[0,T]}$. *Denote $\|\cdot\|_{M_T}$ as the $\ell_2$ norm of the projection of the input vector onto linear*
183 *space $M_T$. Suppose that $\exists c, \beta > 0$ s.t. for any $\boldsymbol{W}, \boldsymbol{W}'$ and $\boldsymbol{x}, \boldsymbol{y}$ we have $\|\nabla\ell(\boldsymbol{f}_{\boldsymbol{W}}(\boldsymbol{x}); \boldsymbol{y})) -$*
184 $\nabla\ell(\boldsymbol{f}_{\boldsymbol{W}'}(\boldsymbol{x}); \boldsymbol{y})\|_2 < \max\{c, \beta\|\boldsymbol{W} - \boldsymbol{W}'\|_{M_T}\}$. *Then over the randomness of the Brownian motion*
185 $\boldsymbol{B}_t$ *and initialization distribution* (5) *in Langevin diffusion* $(\boldsymbol{W}_t)_{t\in[0,T]}$, *it satisfies that*

$$\int_0^T \mathbb{E}_{p_t}\left[\|\nabla\mathcal{L}(\boldsymbol{W}; \mathcal{D}) - \nabla\mathcal{L}(\boldsymbol{W}; \mathcal{D}')\|_2^2\right] dt \leq 2 \cdot T \cdot \underbrace{\mathbb{E}_{p_0}\left[\|\nabla\mathcal{L}(\boldsymbol{W}; \mathcal{D}) - \nabla\mathcal{L}(\boldsymbol{W}; \mathcal{D}')\|_2^2\right]}_{\text{gradient difference at initialization}}$$

$$+ 2\left(\frac{e^{(2+\beta^2)T} - (2+\beta^2)T}{2+\beta^2}\right) \cdot \underbrace{\left(\mathbb{E}_{p_0}\left[\|\nabla\mathcal{L}(\boldsymbol{W}; \mathcal{D})\|_2^2\right] + \sigma^2 rank(M_T) + c^2\right)}_{\text{gradient difference fluctuation during training}} + \underbrace{2c^2 \cdot T}_{\text{non-smoothness cost}}. \tag{7}$$

186 *Remark* 3.3. The assumption $\|\nabla\ell(\boldsymbol{f}_{\boldsymbol{W}}(\boldsymbol{x}); \boldsymbol{y})) - \nabla\ell(\boldsymbol{f}_{\boldsymbol{W}'}(\boldsymbol{x}); \boldsymbol{y})\|_2 < \max\{c, \beta\|\boldsymbol{W} - \boldsymbol{W}'\|_{M_T}\}$ is
187 similar to smoothness condition for the loss function, but is more relaxed as it allows non-smoothness
188 at places where the gradient is bounded. Therefore, the assumption holds under ReLU activation.

189 *Remark* 3.4 (Gradient difference at initialization). The first term and in our upper bound linearly
190 scales with the difference between gradients on neighboring datasets $\mathcal{D}$ and $\mathcal{D}'$ at initialization. Under
191 different initializations, this gradient difference exhibits different dependency on the network depth
192 and width, as we will prove theoretically in Lemma 4.1.

*Remark* 3.5 (Gradient difference fluctuation during training). The second term in our upper bound is to bound the change of gradient difference norm during training, and is therefore proportional to the the rank of a subspace $M_T$ spanned by gradients of all training data. Intuitively, this fluctuation is because Langevin diffusion adds per-dimensional noise with variance $\sigma^2$, thus perturbing the training parameters away from the initialization at a rate of $O(\sigma\sqrt{\text{rank}(M_T)})$ in expected $\ell_2$ distance.

***Growth of KL privacy bound with regard to training time*** $T$. The first term in the gradient difference bound Lemma 3.2 grows linearly with the training time $T$, while the second term grows exponentially with regard to $T$. Consequently, for learning tasks that requires a long training time to converge, the second term will become the dominating term and the KL privacy bound suffers from exponential growth with regard to the training time. Nevertheless, if the total amount of required training time (for convergence) is small enough e.g. $T \leq \frac{1}{2(2+\beta^2)}$, then we have that $\frac{e^{(2+\beta^2)T}-(2+\beta^2)T}{2+\beta^2} < T$ and therefore the second term in the gradient difference upper bound accumulates at a lower than linear rate with increasing training time.

***Dependence of KL privacy bound on network overparameterization***. Under a fixed training time $T$ and noise scale $\sigma^2$, Lemma 3.2 predicts that the KL divergence upper bound in Theorem 3.1 is dependent on the gradient difference and gradient norm at initialization, and the rank of gradient subspace $\text{rank}(M_T)$ throughout training. We now discuss the how these two terms change under increasing width and depth, and whether there are possibilities to improve them under overparameterization.

1. The gradient norm at initialization crucially depend on how the per-layer variance in the Gaussian initialization distribution scales with the network width and depth. Therefore, it is possible to improve the KL privacy bound by using initialization distributions that enable smaller gradient difference at initialization, as we will theoretically show in Section 4.

2. Regarding the rank of gradient subspace $\text{rank}(M_T)$: when the gradients along the training trajectory span the whole optimization space, $\text{rank}(M)$ would equal the dimension of the learning problem. Consequently, the gradient fluctuation upper bound (and thus the KL privacy bound) worsens with increasing number of model parameters (overparameterization) in the worst-case. However, if the gradients are low-dimensional [39, 26, 37] or sparse [30], it is possible that $\text{rank}(M_T)$ will be dimension-independent and thus enable better bound for gradient fluctuation (and KL privacy bound). We leave this as an interesting open problem.

# 4 KL privacy bound for Linearized Network under overparameterization

In this section, we restrict ourselves to the training of linearized networks as described in (3), and investigate the interplay between KL privacy and overparameterization (increasing width and depth). The analysis of DNN via linearization is a commonly used technique in both theory and practice. In theory, DNN can work in the lazy training regime [16] (also called linear regime), under which the linearized network well approximates the training dynamics for deep neural entwork [28] and has been well studied by NTK. In practice, linearized network can still achieve decent performance, which provides a good justification of linearized networks. [37, 34]. We hope our analysis for linearized network serve as an initial attempt that would open a door to theoretically understanding the relationship between overparameterization and privacy.

To derive a composition-based KL privacy bound for training linearized network, we apply Theorem 3.1 which requires an upper bound for the norm of gradient difference between the training processes on neighboring datasets $\mathcal{D}$ and $\mathcal{D}'$ at any time $t$. Note that the empirical risk function for training linearized models enjoys convexity, and therefore requires a relatively short amount of training time for convergence. Therefore intuitively, the gradient difference between neighboring datasets does not change a lot during training, thus allowing us to prove tighter upper bound for the gradient difference norm for linearized networks (than Lemma 3.2).

In the following lemma, we prove that for linearized network, the gradient difference throughout training has a uniform upper bound that only depends on the network width, depth and initialization.

**Lemma 4.1** (Gradient Difference throughout training linearized network)**.** *Under Assumption 2.1, taking over the randomness of the random initialization and the Brownian motion in Langevin*

diffusion, for any $t \in [0, T]$, it satisfies that

$$\mathbb{E}\left[\|\nabla\mathcal{L}(\boldsymbol{W}_t; \mathcal{D}) - \mathcal{L}(\boldsymbol{W}_t; \mathcal{D}')\|^2\right] \leq \frac{4B}{n^2}, \tag{8}$$

where $n$ is the training dataset size, and $B$ is a constant that only depends on the network width, depth and initialization distribution as follows.

$$B := o\left(\prod_{i=1}^{L-1} \frac{\beta_i m_i}{2}\right) \sum_{l=1}^{L} \frac{\beta_L}{\beta_l}, \tag{9}$$

where $o$ is the number of output classes, $\{m_i\}_{i=1}^L$ are the per-layer network widths, and $\{\beta_i\}_{i=1}^L$ are the variances of Gaussian initialization at each layer.

Lemma 4.1 provides an precise analytical upper bound for the gradient difference during training linearized network, by tracking the gradient distribution under fully connected feed-forward ReLU network with Gaussian weight matrices. The full proof is in Appendix C.1 and is heavily based on similar techniques for computing the gradient distribution in the NTK literature [2, 47]. By plugging Eq. (8) into Theorem 3.1, we have the following KL privacy bound for training linearized network.

**Corollary 4.2** (KL privacy bound for training linearized network)**.** *Under Assumption 2.1 and neural networks* (3) *initialized by Gaussian distribution with per-layer variance* $\{\beta_i\}_{i=1}^L$, *running Langevin diffusion for linearized network with time $T$ on neighboring datasets satisfies that*

$$KL(\boldsymbol{W}_t^{lin} \| \boldsymbol{W'}_T^{lin}) \leq \frac{2BT}{n^2\sigma^2}. \tag{10}$$

where $B$ is the constant that specifies the gradient norm upper bound, given by (9).

***Overparameterization affects privacy differently under different initialization***. Corollary 4.2 and Lemma 4.1 suggest that the effect of network overparameterization on KL privacy bound crucially relies on how the per-layer Gaussian initialization variance $\beta_i$ is scaled with the per-layer network width $m_i$ and depth $L$. We summarize our KL privacy bound for linearized network under different width, depth and initialization schemes in Table 1, and elaborate the comparison below.

**(1) LeCun initialization** uses small, width-independent variance for initializing the first layer $\beta_1 = \frac{1}{d}$ (where $d$ is the number of input features), and width-dependent variance $\beta_2 = \cdots = \beta_L = \frac{1}{m}$ for initializing all the subsequent layers. Therefore, the second term $\sum_{l=1}^L \frac{\beta_L}{\beta_l}$ in the constant $B$ (9) increases linearly with the width $m$ and depth $L$. However, due to $\frac{m_l \cdot \beta_l}{2} < 1$ for all $l = 2, \cdots, L$, the first product term $\prod_{l=1}^{L-1} \frac{\beta_l m_l}{2}$ in constant $B$ decays with the increasing depth. Therefore, by combining the two terms, we prove that the KL privacy bound worsens with increasing width, but improves with increasing depth (as long as the depth is large enough). Similarly, under Xavier initialization $\beta_l = \frac{2}{m_{l-1} + m_l}$, we prove that the KL privacy bound (especially the constant $B$ (9)) improves with increasing depth as long as the depth is large enough.

**(2) NTK and He initializations** user large per-layer variance $\beta_l = \begin{cases} \frac{2}{m_l} & l = 1, \cdots, L-1 \\ \frac{1}{o} & l = L \end{cases}$ (for NTK) and $\beta_l = \frac{2}{m_{l-1}}$ (for He). Consequently, the gradient difference under NTK or He initialization is significantly larger than that under LeCun initialization. Specifically, the gradient norm constant $B$ (9) grows linearly with the width $m$ and the depth $L$ under He and NTK initializations, thus suggesting a worsening of KL privacy bound under increasing width and depth.

## 5   Utility guarantees for Training Linearized Network

Our privacy analysis suggest that training linearized network under certain initialization schemes (such as LeCun initialization) enable significantly better privacy bounds under overparameterization by increasing depth. In this section, we further prove utility bounds for Langevin diffusion under initialization schemes, and investigate the effect of overparameterization on the privacy utility trade-off. In other words, we aim to understand whether there are any utility degradation for training linearized network when using the more privacy-preserving initialization schemes.

**Convergence of training linearized network**. We now prove convergence of excess empirical risk in training linearized network via Langevin diffusion. This is well-studied problem in the literature for noisy gradient descent. We extend the convergence theorem to continuous-time Langevin diffusion below, and investigate factors that affect the convergence under overparameterization.

**Proposition 5.1** (Extension of [36, Theorem 2] and [39, Theorem 3.1]). *Let $\boldsymbol{W}_0^{lin}$ be a randomly initialized parameter vector by* (5). *Let the empirical NTK feature mapping matrix for dataset training $\mathcal{D}$ at initialization be $M_0 = \left(\nabla \boldsymbol{f}_{\boldsymbol{W}_0^{lin}}(\boldsymbol{x}_1) \quad \cdots \quad \nabla \boldsymbol{f}_{\boldsymbol{W}_0^{lin}}(\boldsymbol{x}_n)\right)$. Let $\mathcal{L}_0^{lin}(\boldsymbol{W}; \mathcal{D})$ be the empirical loss for linearized network* (3) *expanded at initialization vector $\boldsymbol{W}_0^{lin}$. Then running Langevin diffusion* (4) *under empirical loss $\mathcal{L}_0^{lin}(\boldsymbol{W}; \mathcal{D})$ and initialization $\boldsymbol{W}_0^{lin}$ for time $T$ satisfies the following excess empirical risk bound*

$$\mathbb{E}[\mathcal{L}(\boldsymbol{W}_T^{lin})] - \mathbb{E}[\mathcal{L}(\boldsymbol{W}_0^*; \mathcal{D})] \leq \frac{2R}{T} + \frac{1}{2}\sigma^2 \mathbb{E}[rank(M_0)]\left(1 + \log\frac{2BT^2}{R}\right) \qquad (11)$$

*where $\boldsymbol{W}_0^*$ is an (exact or approximate) solution for the ERM problem on $\mathcal{L}_0^{lin}(\boldsymbol{W}; \mathcal{D})$, and $R = \mathbb{E}[\|\boldsymbol{W}_0^{lin} - \boldsymbol{W}_0^*\|_{M_0}^2]$ is the expected gap between initialization parameters $\boldsymbol{W}_0$ and solution $\boldsymbol{W}_0^*$.*

*Remark* 5.2. The excess empirical risk bound Proposition 5.1 is smaller if data is low-rank, e.g., image data, then $rank(M_0)$ is small. This is consistent with the prior dimension-independent private learning literature [26, 27, 30] and show benefit of low-dimensional gradients on private learning.

Proposition 5.1 highlights that the excess empirical risk scales with the expected gap $R$ between initialization and optima (which we refer as the lazy training distance), the rank of the gradient subspace rank$(M_0)$, and the constant $B$ that specifies upper bound for expected gradient norm during training. Specifically, the smaller is the lazy training distance $R$, the better is the excess risk bound for Langevin diffusion given fixed training time $T$ and noise variance $\sigma^2$. We have discussed how overparameterization affects the the gradient norm constant $B$ and the gradient subspace rank rank$(M_0)$ in Section 3. Therefore, we only still need to investigate how the lazy training distance $R$ changes with the network width, depth and initialization, as follows.

**Lazy training distance $R$ decreases with increasing depth**. It is widely observed in the literature [16, 48, 31] that under appropriate choices of initializations, gradient descent on fully connected neural network falls under a lazy training regime. That is, with high probability, there exists a (nearly) optimal solution for the ERM problem that is close to the initialization parameters in terms of $l_2$ distance. Moreover, this lazy training distance $R$ is closely related to the smallest eigenvalue of the NTK matrix. In the following proposition, we compute a near optimal solution via the pseudo inverse of the NTK matrix, and prove that it has small distance to the initialization parameters via existing lower bounds for the smallest eigenvalue of the NTK matrix [33].

**Proposition 5.3** (Bounding lazy training distance via smallest eigenvalue of the NTK matrix). *Under the data and network regularity Assumption 2.1, if the width $m_1 = \cdots = m_{L-1} = \Omega(n)$ is sufficiently large, then there exists an optimal solution $\boldsymbol{W}_0^{\frac{1}{n^2}}$ that satisfies $\mathcal{L}_0^{lin}(\boldsymbol{W}_0^{\frac{1}{n^2}}) \leq \frac{1}{n^2}$ and satisfies*

$$\tilde{R} = \mathbb{E}[\|\boldsymbol{W}_0^{\frac{1}{n}} - \boldsymbol{W}_0\|_2^2] \leq \begin{cases} \tilde{O}(\frac{n}{d \cdot 2^L \cdot (m(L-2)+1)}) & \textit{for NTK initialization} \\ \tilde{O}(\frac{n}{2^L m(L-1)}) & \textit{for He initialization} \\ \tilde{O}(\frac{n}{m(L-1)}) & \textit{for LeCun initialization} \end{cases} \qquad (12)$$

We refer to $\tilde{R}$ as the approximate lazy training distance because $\boldsymbol{W}_0^{\frac{1}{n^2}}$ is only a nearly optimal solution for the ERM problem. Proposition 5.3 shows that this approximate lazy training distance improves with overparameterization (width and depth) under LeCun, He and NTK initializations.

**Privacy Excess empirical risk Tradeoff for training linearized network via Langevin diffusion**. We now use the approximate lazy training distance $\tilde{R}$ to prove empirical risk bound, and combine it with our KL privacy bound Section 4 to show the privacy utility trade-off under overparameterization.

**Corollary 5.4** (Privacy utility trade-off for last iterate). *Assume that the data and network regularity Assumption 2.2 holds. Assume that all the conditions and definition for constants in Proposition 5.1*

holds. Then by setting $\sigma^2 = \frac{2BT}{\varepsilon n^2}$ and $T = \sqrt{\frac{2\varepsilon n \tilde{R}}{B}}$, we have that running Langevin diffusion for time $T$ satisfies bound KL divergence $\epsilon$, and has ecess empirical risk upper bounded by

$$\mathbb{E}[\mathcal{L}(\boldsymbol{W}_T^{lin})] \leq \mathcal{O}\left(\frac{1}{n^2} + \sqrt{\frac{B\tilde{R}}{\varepsilon n}} \log(\varepsilon n)\right) \tag{13}$$

where $B$ is the gradient norm constant Eq. (9), and $\tilde{R}$ is the approximate lazy training distance in Eq. (12). A summary of $B$ and $\tilde{R}$ under different initializations is in Table 1.

Corollary 5.4 suggests that the excess empirical risk worsens in the presence of a stronger privacy constraint, i.e., under a small privacy budget $\varepsilon$, thus contributing to a trade-off between privacy and utility. However, the excess empirical risk also scales with constants such as the approximate lazy training distance $\tilde{R}$ and the gradient norm constant $B$. These constants depend on network width, depth and initialization distributions, and therefore we prove privacy utility trade-offs for training linearized network that changes with overparameterization, as sumarized in Table 1.

We would like to highlight that our privacy utility trade-off bound under LeCun and He initialization strictly improves with increasing width and depth as long as the data and network satisfy regularity Assumption 2.2 and the network width is large enough. To our best knowledge, this is the first time that a strictly improving privacy utility trade-off under overparameterization is shown in literature. This shows the benefits of our precise KL privacy analysis under overparameterization.

***Extension of privacy utility results to excess population risk***. Our privacy utility trade-off results can be generalized to to excess population risk, by additionally bounding the generalization error with standard stability-based arguments [35, 24, 8]. We elaborate the details below.

**Proposition 5.5** (Extended excess population risk bound for training linearized network)**.** *Denote* $R_0(\boldsymbol{W}) = \mathbb{E}_{(\boldsymbol{x},\boldsymbol{y}) \in pop}[\ell(f_{\boldsymbol{W}_0}(\boldsymbol{x}) + \frac{\partial \boldsymbol{f}_{\boldsymbol{W}_0}(\boldsymbol{x})}{\partial \boldsymbol{W}_0}(\boldsymbol{W} - \boldsymbol{W}_0); \boldsymbol{y})]$ *as the population risk of linearized network expanded at initialization vector* $\boldsymbol{W}_0$ *over population data distribution pop. Then under the conditions of Corollary 5.4, we have that for any dataset* $\mathcal{D}$ *of size* $n$*, the following excess population risk upper bound holds.*

$$\mathbb{E}[R_0(\boldsymbol{W}_T)] - \mathbb{E}[\mathcal{L}(\boldsymbol{W}_{pop,0}^*; \mathcal{D})] \leq O\left(\frac{1}{n^2} + \sqrt{\frac{B\tilde{R}}{\varepsilon n}}(\log(\varepsilon n) + \varepsilon)\right) \tag{14}$$

*where the expectation is over the randomness of sampling the training dataset* $\mathcal{D} \sim pop^n$ *from the data population and the random coins for the Langevin diffusion training algorithm, and* $\boldsymbol{W}_{pop,0}^* = \mathrm{argmin}_{\boldsymbol{W}} R_0(\boldsymbol{W})$ *is the optimal solution for the population risk minimization problem.*

Proposition 5.5 shows that the excess population risk bound is almost the same as excess empirical risk, except that there is an additional generalization term that scales with the privacy budget $\varepsilon$. Intuitively, this is because the generalization error is proportional to the stability of model prediction function under different training dataset, which is smaller when the KL privacy loss $\varepsilon$ is small.

# 6 Conclusion

We prove new KL privacy bound for training fully connected ReLu network (and its linearized variant) using the Langevin diffusion algorithm, and investigate how privacy is affected by the network width, depth and initialization. Our results suggest that there is a complex interplay between privacy and overparameterization (width and depth) that crucially relies on what initialization distribution is used and the how much the gradient fluctuates during training. To this end, we show that for training a linearized variant of fully connected network with finite time, it is possible to prove a KL privacy bound that improves with depth, as long as the initialization distribution is set appropriately (such as LeCun). We also study the excess empirical and population risk bounds for linearized network, and prove that the privacy-utility trade-off similarly improves as depth increases under LeCun initialization. This shows the gain of our new privacy analysis for capturing the effect of overparameterization. We leave it as an important open problem as to whether our privacy utility trade-off results for linearized network could be generalized to deep neural networks.

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

# Contents

## A  Symbols and definitions

Vecorization $\text{Vec}(\cdot)$ denotes the transformation that takes an input matrix $\boldsymbol{A} = (a_{ij})_{i \in [r], j \in [c]} \in \mathbb{R}^{r \times c}$ (with $r$ rows and $c$ columns) and outputs a $rc$-dimensional column vector: $\text{Vec}(\boldsymbol{A}) = (a_{1,1}, \cdots, a_{r,1}, a_{1,2}, \cdots, a_{r,2}, \cdots, a_{1,c}, \cdots, a_{r,c})^{\top}$.

Distribution $p_t$ and $p_t'$: we denote $p_t$ as the distribution of model parameters after running Langevin diffusion on dataset $\mathcal{D}$ with time $t$, and similarly denote $p_t'$ as the distribution of model parameters after running Langevin diffusion on dataset $\mathcal{D}'$ with time $t$.

Softmax function: $\text{softmax}(\boldsymbol{y}) = \frac{e^{\boldsymbol{y}^{[j]}}}{\sum_{j=1}^{o} e^{\boldsymbol{y}^{[j]}}}$ where $o$ is the number of output classes.

Neighboring datasets $D$ and $D'$: two dataset with the same number of data records that differ in one record. We also denote the differing records as $(\boldsymbol{x}, \boldsymbol{y}) \in \mathcal{D}$ and $(\boldsymbol{x}', \boldsymbol{y}') \in \mathcal{D}'$.

532  $o$: number of output classes for the neural network.

## B  Deferred proofs for Section 3

### B.1  Deferred proofs for Theorem 3.1

535  To prove the new composition theorem, we will use the Girsanov's Theorem. Here we follow the
536  presentation of [15, Theorem 6].

537  **Theorem B.1** (Implication of Girsanov's theorem [15, Theorem 6])**.** *Let* $(\tilde{X}_t)_{t\in[0,\eta]}$ *and* $(X_t)_{t\in[0,\eta]}$
538  *be two continuous-time processes over* $\mathbb{R}^r$. *Let* $P_T$ *be the probability measure that corresponds to the*
539  *trajectory of* $(\tilde{X}_t)_{t\in[0,\eta]}$, *and let* $Q_T$ *be the probability measure that corresponds to the trajectory of*
540  $(X_t)_{t\in[0,\eta]}$. *Suppose that the process* $(\tilde{X}_t)_{t\in[0,\eta]}$ *follows*

$$d\tilde{X}_t = \tilde{b}_t dt + \sigma_t d\tilde{B}_t,$$

541  *where* $\tilde{B}$ *is a Brownian motion, and the process* $(X_t)_{t\in[0,\eta]}$ *follows*

$$dX_t = b_t dt + \sigma_t dB_t,$$

542  *where* $B$ *is a Brownian motion. We assume that for each* $t > 0$, $\sigma_t$ *is a* $r \times r$ *symmetric positive*
543  *definite matrix. Then, provided that Novikov's condition holds,*

$$\mathbb{E}_{Q_T} \exp\left( \frac{1}{2} \int_0^\eta \|\sigma_t^{-1}(\tilde{b}_t - b_t)\|_2^2 dt \right) < \infty \,, \tag{15}$$

544  *we have that*

$$\frac{dP_T}{dQ_T} = exp\left( \int_0^\eta \sigma_t^{-1}(\tilde{b}_t - b_t) dB_t - \frac{1}{2} \int_0^\eta \|\sigma_t^{-1}(\tilde{b}_t - b_t)\|_2^2 dt \right) \,.$$

545  Now we apply Girsanov's theorem on the coupled Langevin diffusion processes on neighboring
546  datasets, and obtain the following new composition theorem for KL divergence in the context of
547  privacy.

548  ***Theorem 3.1*** (KL composition under possibly unbounded gradient difference)**.**  The KL divergence
549  between running Langevin diffusion (4) for DNN (2) on neighboring datasets $\mathcal{D}$ and $\mathcal{D}'$ satisfies

$$KL(\boldsymbol{W}_T, \boldsymbol{W}_T') \leq \frac{1}{2\sigma^2} \int_0^T \mathbb{E}\left[ \|\nabla\mathcal{L}(\boldsymbol{W}_t; \mathcal{D}) - \nabla\mathcal{L}(\boldsymbol{W}_t; \mathcal{D}')\|_2^2 \right] dt \,. \tag{16}$$

550  *Proof.* Denote $p_t$ as the distribution of model parameters after running Langevin diffusion on dataset
551  $\mathcal{D}$ with time $t$, and similarly denote $p_t'$ as the distribution of model parameters after running Langevin
552  diffusion on dataset $\mathcal{D}'$ with time $t$. Then by definition,

$$\frac{\partial KL(p_t, p_t')}{\partial t} = \lim_{\eta \to 0} \frac{KL(p_{t+\eta}, p_{t+\eta}') - KL(p_t, p_t')}{\eta}$$
$$\leq \lim_{\eta \to 0} \frac{KL(p_{t,t+\eta}, p_{t,t+\eta}') - KL(p_t, p_t')}{\eta}, \tag{17}$$

553  where the last inequality is by the data processing inequality for KL divergence [45, Theorem 9]
554  (with the data processing operation given by $(\boldsymbol{W}_t, \boldsymbol{W}_{t+\eta}) \to \boldsymbol{W}_t$). Now we compute the term
555  $KL(p_{t:t+\eta}, p_{t:t+\eta}')$ as follows.

$$KL(p_{t,t+\eta}, p_{t,t+\eta}') = \mathbb{E}_{p_{t,t+\eta}(\boldsymbol{W}_t, \boldsymbol{W}_{t+\eta})}\left[ \log\left( \frac{p_{t+\eta|t}(\boldsymbol{W}_{t+\eta}|\boldsymbol{W}_t)p_t(\boldsymbol{W}_t)}{p_{t+\eta|t}'(\boldsymbol{W}_{t+\eta}|\boldsymbol{W}_t)p_t'(\boldsymbol{W}_t)} \right) \right]$$
$$= \mathbb{E}_{p_{t,t+\eta}(\boldsymbol{W}_t, \boldsymbol{W}_{t+\eta})}\left[ \log\left( \frac{p_{t+\eta|t}(\boldsymbol{W}_{t+\eta}|\boldsymbol{W}_t)}{p_{t+\eta|t}'(\boldsymbol{W}_{t+\eta}|\boldsymbol{W}_t)} \right) \right] + \mathbb{E}_{p_t(\boldsymbol{W}_t)}\left[ \log\left( \frac{p_t(\boldsymbol{W}_t)}{p_t'(\boldsymbol{W}_t)} \right) \right]$$
$$= \mathbb{E}_{p_t(\boldsymbol{W}_t)}\left[ KL(p_{t+\eta|t}, p_{t+\eta|t}') \right] + KL(p_t, p_t') \tag{18}$$

Now we want to apply Girsanov's theorem to the following langevin diffusion processes $(\boldsymbol{W}_{t+s|t})_{s\in[0,\eta]}$ and $(\boldsymbol{W}'_{t+s|t})_{s\in[0,\eta]}$.

$$d\boldsymbol{W}_{t+s|t} = -\nabla\mathcal{L}(\boldsymbol{W}_{t+s};\mathcal{D})dt + \sqrt{2\sigma^2}dB_s$$

$$d\boldsymbol{W}'_{t+s|t} = -\nabla\mathcal{L}(\boldsymbol{W}'_{t+s};\mathcal{D}')dt + \sqrt{2\sigma^2}dB_s$$

where we have the boundary condition that $\boldsymbol{W}_{t|t} = \boldsymbol{W}'_{t|t}$ due to the conditioning at time $t$. Note that when $\eta$ is small enough, we have that the Novikov's condition in Eq. (15) holds because the exponent inside integration $\frac{1}{2}\int_0^\eta\|\sigma_t^{-1}(\tilde{b}_t - b_t)\|_2^2 dt$ scales linearly with $\eta$ and is small when $\eta$ is small enough. Therefore, by applying Girsanov's theorem, we have that

$$KL(p_{t+\eta|t}, p'_{t+\eta|t}) \leq KL(p_{t:t+\eta|t}, p'_{t:t+\eta|t})$$

$$= \mathbb{E}_{p_{t:t+\eta|t}}\left[\int_0^\eta \sigma^{-1}(\tilde{b}_s - b_s)dB_s - \frac{1}{2}\int_0^T \|\sigma^{-1}(\tilde{b}_s - b_s)\|_2^2 ds\right]$$

where $\tilde{b}_s - b_s = -\nabla\mathcal{L}(\boldsymbol{W}_{t+s};\mathcal{D}) + \nabla\mathcal{L}(\boldsymbol{W}_{t+s};\mathcal{D}')$. By Itô integration with regard to standard Brownian motion, we have that

$$KL(p_{t+\eta|t}, p'_{t+\eta|t}) \leq \frac{1}{2\sigma^2}\mathbb{E}_{p_{t:t+\eta|t}}\left[\int_0^\eta \|\nabla\mathcal{L}(\boldsymbol{W}_{t+s};\mathcal{D}) - \nabla\mathcal{L}(\boldsymbol{W}_{t+s};\mathcal{D}')\|_2^2 ds\right] \qquad (19)$$

By plugging Eq. (19) into Eq. (18), we have that

$$KL(p_{t,t+\eta}, p'_{t,t+\eta}) \leq \frac{1}{2\sigma^2}\mathbb{E}_{p_{t:t+\eta}}\left[\int_0^\eta \|\nabla\mathcal{L}(\boldsymbol{W}_{t+s};\mathcal{D}) - \nabla\mathcal{L}(\boldsymbol{W}_{t+s};\mathcal{D}')\|_2^2 ds\right] + KL(p_t, p'_t) \tag{20}$$

By plugging Eq. (20) into Eq. (17), and by exchanging the order of expectation and integration, we have that

$$\frac{\partial KL(p_t, p'_t)}{\partial t} \leq \frac{1}{2\sigma^2}\lim_{\eta\to 0}\frac{\int_0^\eta \mathbb{E}_{p_{t+s}}\left[\|\nabla\mathcal{L}(\boldsymbol{W}_{t+s};\mathcal{D}) - \nabla\mathcal{L}(\boldsymbol{W}_{t+s};\mathcal{D}')\|_2^2\right]ds}{\eta}$$

$$= \frac{1}{2\sigma^2}\mathbb{E}_{p_t}\left[\|\nabla\mathcal{L}(\boldsymbol{W}_t;\mathcal{D}) - \nabla\mathcal{L}(\boldsymbol{W}_t;\mathcal{D}')\|_2^2\right] \tag{21}$$

Integrating Eq. (21) on $t \in [0, T]$ finishes the proof. $\qquad\square$

## B.2 Deferred proofs for Lemma 3.2

***Lemma 3.2.*** Let $M_T$ be the subspace spanned by gradients $\{\nabla\ell(\boldsymbol{f}_{\boldsymbol{W}_t}(\boldsymbol{x}_i;\boldsymbol{y}_i) : (\boldsymbol{x}_i, \boldsymbol{y}_i) \in \mathcal{D}, t \in [0, T]\}$ on each training data record throughout Langevin diffusion $(\boldsymbol{W}_t)_{t\in[0,T]}$. Denote $\|\cdot\|_{M_T}$ as the $\ell_2$ norm of the projection of the input vector onto linear space $M_T$. Suppose that $\exists c, \beta > 0$ s.t.

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

*Proof.* Denote $\boldsymbol{W}$ as the initialization parameters and denote $\boldsymbol{W}_t^{lin}$ as the parameters for linearized network after training time $t$. Then the gradient difference under linearized network and cross-entropy loss function is as follows.

$$\|\nabla\mathcal{L}(W_t;\mathcal{D}) - \nabla\mathcal{L}(W_t;\mathcal{D}')\|_F^2$$

$$= \left\| \frac{\nabla \boldsymbol{f_W}(\boldsymbol{x})^\top \left(\mathrm{softmax}(\boldsymbol{f_{W_t}}(\boldsymbol{x})) - \boldsymbol{y}\right)}{n} - \frac{\nabla \boldsymbol{f_W}(\boldsymbol{x}')^\top \left(\mathrm{softmax}(\boldsymbol{f_{W_t}}(\boldsymbol{x}')) - \boldsymbol{y}'\right)}{n} \right\|_F^2$$

$$\leq \frac{2}{n^2}\left( \|\nabla \boldsymbol{f_W}(\boldsymbol{x})\|_F^2 + \|\nabla \boldsymbol{f_W}(\boldsymbol{x}')\|_F^2 \right).$$

Plugging Lemma 4.1 into the above equation with data Assumption 2.1 suffice to prove the result. $\square$

# D   Deferred proofs for Section 5

## D.1   Excess empirical risk for training linearized network (average iterate)

To prove empirical risk bound for the last iterate of training linearized network, we will first need to prove the following intermediate result of excess empirical risk bound for average iterate.

**Lemma D.1** (Excess empirical risk for average iterate (Extension of [39, Theorem 3.1]))**.** *Let $\boldsymbol{W}_0^{lin}$ be a randomly initialized parameter vector by (5). Let the empirical NTK feature matrix for dataset training $\mathcal{D}$ at initialization be $M_0 = \left( \nabla \boldsymbol{f_{W_0^{lin}}}(\boldsymbol{x}_1) \quad \cdots \quad \nabla \boldsymbol{f_{W_0^{lin}}}(\boldsymbol{x}_n) \right)$. Let $\mathcal{L}_0^{lin}(\boldsymbol{W};\mathcal{D})$ be the empirical loss for linearized network (3) expanded at initialization vector $\boldsymbol{W}_0^{lin}$. Then running Langevin diffusion (4) under empirical loss $\mathcal{L}_0^{lin}(\boldsymbol{W};\mathcal{D})$ and initialization $\boldsymbol{W}_0^{lin}$ for time $T$ satisfies the following excess empirical risk bound.*

$$\mathbb{E}[\mathcal{L}(\bar{\boldsymbol{W}}_T^{lin})] - \mathbb{E}[\mathcal{L}(\boldsymbol{W}_0^*;\mathcal{D})] \leq \frac{R}{2T} + \frac{1}{2}\sigma^2\mathbb{E}[rank(M_0)]$$

*where $\bar{\boldsymbol{W}}_T^{lin} = \frac{1}{T}\int \bar{\boldsymbol{W}}_t^{lin}dt$ is the average of all iterates, $\boldsymbol{W}_0^*$ is an (exact or approximate) solution for the ERM problem on $\mathcal{L}_0^{lin}(\boldsymbol{W};\mathcal{D})$, and $R = \mathbb{E}[\|\boldsymbol{W}_0^{lin} - \boldsymbol{W}_0^*\|_{M_0}^2]$ is the expected gap between initialization parameters $\bar{\boldsymbol{W}}_0$ and solution $\boldsymbol{W}_0^*$.*

*Proof.* Our proofs are heavily based on the idea in [39, Theorem 3.1] to work only in the parameter space spanned by the input feature vectors. And our proof serves as an extension of their bound to the continuous-time Langevin diffusion algorithm. We begin by using convexity of the empirical loss function $\mathcal{L}^{lin}(\boldsymbol{W};\mathcal{D})$ for linearized network to prove the following standard results

$$\mathcal{L}^{lin}(\bar{\boldsymbol{W}}_T^{lin};\mathcal{D}) - \mathcal{L}^{lin}(\boldsymbol{W}_0^*;\mathcal{D}) \leq \langle \bar{\boldsymbol{W}}_T^{lin} - \boldsymbol{W}_0^*, \nabla\mathcal{L}^{lin}(\bar{\boldsymbol{W}}_T^{lin};\mathcal{D})\rangle \tag{48}$$

Denote $M_0 = \left( \nabla \boldsymbol{f_{W_0^{lin}}}(\boldsymbol{x}_1) \quad \cdots \quad \nabla \boldsymbol{f_{W_0^{lin}}}(\boldsymbol{x}_n) \right)$. By computing the gradient under cross entropy loss and linearized network, we have $\nabla\mathcal{L}^{lin}(\boldsymbol{W}_T^{lin};\mathcal{D})$ lies in the column space of $M_0$. Denote $\Pi_{M_0}$ as the projection operator to the column space of $M_0$, then (48) can be rewritten as

$$\mathcal{L}^{lin}(\bar{\boldsymbol{W}}_T^{lin};\mathcal{D}) - \mathcal{L}^{lin}(\boldsymbol{W}_0^*;\mathcal{D}) \leq \langle \Pi_{M_0}(\bar{\boldsymbol{W}}_T^{lin} - \boldsymbol{W}_0^*), \nabla\mathcal{L}^{lin}(\bar{\boldsymbol{W}}_T^{lin};\mathcal{D})\rangle. \tag{49}$$

By taking expectation over training randomness and initialization distribution, we have

$$\mathbb{E}[\mathcal{L}(\bar{\boldsymbol{W}}_T^{lin})] - \mathbb{E}[\mathcal{L}(\boldsymbol{W}_0^*;\mathcal{D})] \leq \frac{1}{T}\int_0^T \mathbb{E}\left[\langle \Pi_{M_0}(\boldsymbol{W}_t^{lin} - \boldsymbol{W}_0^*), \nabla\mathcal{L}^{lin}(\boldsymbol{W}_t^{lin};\mathcal{D})\rangle\right]dt \tag{50}$$

We now rewrite $\mathbb{E}\left[\langle \Pi_{M_0}(\bar{\boldsymbol{W}}_t^{lin} - \boldsymbol{W}_0^*), \nabla\mathcal{L}^{lin}(\boldsymbol{W}_t^{lin};\mathcal{D})\rangle\right]$ by computing $\frac{\partial}{\partial t}\mathbb{E}[\|\boldsymbol{W}_t^{lin} - \boldsymbol{W}_0^*\|_{M_0}^2]$, where $\|\boldsymbol{W}_t^{lin} - \boldsymbol{W}_0^*\|_{M_0}^2 = \Pi_{M_0}\left(\boldsymbol{W}_t^{lin} - \boldsymbol{W}_0^*\right)^\top \Pi_{M_0}\left(\boldsymbol{W}_t^{lin} - \boldsymbol{W}_0^*\right)$. By applying (27), we have

$$\frac{\partial}{\partial t}\mathbb{E}[\|\boldsymbol{W}_t^{lin} - \boldsymbol{W}_0^*\|_{M_0}^2] = -2\mathbb{E}[\langle \Pi_{M_0}\left(\boldsymbol{W}_t^{lin} - \boldsymbol{W}_0^*\right), \nabla\mathcal{L}^{lin}(\boldsymbol{W}_t^{lin};\mathcal{D})\rangle] + \sigma^2\mathbb{E}[rank(M_0)]$$

$$\tag{51}$$

Therefore by plugging (51) into (50), we have that

$$\mathbb{E}[\mathcal{L}^{lin}(\boldsymbol{W};\mathcal{D}) - \mathcal{L}^{lin}(\boldsymbol{W}_0^*;\mathcal{D})] \leq -\frac{1}{2T}\int_0^T \frac{\partial}{\partial t}\mathbb{E}[\|\boldsymbol{W}_t - \boldsymbol{W}_0^*\|_{M_0}^2]dt + \frac{1}{2}\sigma^2\mathbb{E}[rank(M_0)] \tag{52}$$

$$\leq \frac{1}{2T}\mathbb{E}[\|\boldsymbol{W}_0^{lin} - \boldsymbol{W}_0^*\|_{M_0}^2] + \frac{1}{2}\sigma^2\mathbb{E}[rank(M_0)] \tag{53}$$

$\square$

 **D.2 Deferred proof for Proposition 5.1**

679 We are now ready to prove the last iterate excess empirical risk bound for training linearized network.

680 **Proposition 5.1** (Excess empirical risk for training linearized network (last iterate))**.** Let $\boldsymbol{W}_0^{lin}$ be
681 a randomly initialized parameter vector by (5). Let the empirical NTK feature mapping matrix for
682 dataset training $\mathcal{D}$ at initialization be $M_0 = \left(\nabla \boldsymbol{f}_{\boldsymbol{W}_0^{lin}}(\boldsymbol{x}_1) \quad \cdots \quad \nabla \boldsymbol{f}_{\boldsymbol{W}_0^{lin}}(\boldsymbol{x}_n)\right)$. Let $\mathcal{L}_0^{lin}(\boldsymbol{W}; \mathcal{D})$
683 be the empirical loss for linearized network (3) expanded at initialization vector $\boldsymbol{W}_0^{lin}$. Then running
684 Langevin diffusion (4) under empirical loss $\mathcal{L}_0^{lin}(\boldsymbol{W}; \mathcal{D})$ and initialization $\boldsymbol{W}_0^{lin}$ for time $T$ satisfies
685 the following excess empirical risk bound

$$\mathbb{E}[\mathcal{L}(\boldsymbol{W}_T^{lin})] - \mathbb{E}[\mathcal{L}(\boldsymbol{W}_0^*; \mathcal{D})] \leq \frac{2R}{T} + \frac{1}{2}\sigma^2 \mathbb{E}[rank(M_0)]\left(1 + \log\frac{2BT^2}{R}\right) \tag{54}$$

686 where $\boldsymbol{W}_0^*$ is an (exact or approximate) solution for the ERM problem on $\mathcal{L}_0^{lin}(\boldsymbol{W}; \mathcal{D})$, and $R =$
687 $\mathbb{E}[\|\boldsymbol{W}_0^{lin} - \boldsymbol{W}_0^*\|_{M_0}^2]$ is the expected gap between initialization parameters $\boldsymbol{W}_0$ and solution $\boldsymbol{W}_0^*$.

688 *Proof.* We first define the following potential function.

$$\Phi(t) = \frac{1}{T-t}\int_t^T \mathbb{E}[\mathcal{L}(\boldsymbol{W}_\tau^{lin}; \mathcal{D})]d\tau \tag{55}$$

689 By definition we have that the boundary values are $\Phi(0) = \frac{1}{T}\int_0^T \mathbb{E}[\mathcal{L}(\boldsymbol{W}_\tau^{lin}; \mathcal{D})]d\tau$ and $\Phi(T) =$
690 $\lim_{t\to T}\frac{1}{T-t}\int_t^T \mathbb{E}[\mathcal{L}(\boldsymbol{W}_\tau^{lin}; \mathcal{D})]d\tau = \mathbb{E}[\mathcal{L}(\boldsymbol{W}_T^{lin}; \mathcal{D})]$. Since we have proved upper bound for $\Phi(0)$
691 in the excess empirical risk bound for the average iterate Appendix D.1, to analyze $\Phi(T)$ the
692 loss difference between last iterate and average iterate, we only need to prove upper bound for
693 $\Phi(T) - \Phi(0)$.

694 By definition, we compute the partial derivative of the function $\Phi(t)$ with regard to time $t$ as follows.

$$\frac{\partial}{\partial t}\Phi(t) = \frac{1}{(T-t)^2}\int_t^T \mathbb{E}[\mathcal{L}(\boldsymbol{W}_\tau^{lin}; \mathcal{D})]d\tau - \frac{1}{T-t}\mathbb{E}[\mathcal{L}(\boldsymbol{W}_t^{lin}; \mathcal{D})]$$

$$= \frac{1}{(T-t)^2}\int_t^T \mathbb{E}[\mathcal{L}(\boldsymbol{W}_\tau^{lin}; \mathcal{D}) - \mathcal{L}(\boldsymbol{W}_t^{lin}; \mathcal{D})]d\tau$$

$$\leq \frac{1}{(T-t)^2}\int_t^T \mathbb{E}[\langle\nabla\mathcal{L}(\boldsymbol{W}_\tau^{lin}), \Pi_{M_0}(\boldsymbol{W}_\tau^{lin} - \boldsymbol{W}_t^{lin})\rangle]d\tau \tag{56}$$

695 where the last inequality is by the convexity of loss function for linearized network. Now to control
696 the intergral in (56), we use a similar argument to (51) as follows.

$$\frac{\partial}{\partial \tau}\mathbb{E}\left[\|\boldsymbol{W}_\tau^{lin} - \boldsymbol{W}_t^{lin}\|_{M_0}^2\right] = -2\mathbb{E}[\langle\nabla\mathcal{L}(\boldsymbol{W}_\tau^{lin}), \Pi_{M_0}(\boldsymbol{W}_\tau^{lin} - \boldsymbol{W}_t^{lin})\rangle] + \sigma^2\mathbb{E}[rank(M_0)] \tag{57}$$

697 By plugging (57) into (56), we have that

$$\frac{\partial}{\partial t}\Phi(t) \leq \frac{\int_t^T -\frac{\partial}{\partial\tau}\mathbb{E}[\|\boldsymbol{W}_\tau^{lin} - \boldsymbol{W}_t^{lin}\|_{M_0}^2] + \sigma^2\mathbb{E}[rank(M_0)]d\tau}{2(T-t)^2} \leq \frac{\sigma^2\mathbb{E}[rank(M_0)]}{2(T-t)} \tag{58}$$

698 By intergrating the above equation over $t \in [0, T - \Delta T]$ where $\Delta \in (0,1)$ is a tuning parameter that
699 we will determine later, we have that

$$\Phi(T - \Delta T) \leq \Phi(0) + \sigma^2\mathbb{E}[rank(M_0)]\ln\frac{1}{\Delta} \tag{59}$$

700 Now we proceed to bound $\Phi(T) - \Phi(T - \Delta T)$. By definition of $\Phi(t)$, we have that

$$\Phi(T) - \Phi(T - \Delta T) = \frac{1}{\Delta T}\int_{T-\Delta T}^T \mathbb{E}[\mathcal{L}(\boldsymbol{W}_T^{lin}; \mathcal{D}) - \mathcal{L}(\boldsymbol{W}_\tau^{lin}; \mathcal{D})]d\tau$$

$$\leq \frac{1}{\Delta T}\int_{T-\Delta T}^T \mathbb{E}[\langle\nabla\mathcal{L}(\boldsymbol{W}_T^{lin}; \mathcal{D}), \Pi_{M_0}(\boldsymbol{W}_T^{lin} - \boldsymbol{W}_\tau^{lin})\rangle]d\tau \tag{60}$$

701 where (60) is by convexity of empirical loss function and by that $\nabla\mathcal{L}^{lin}(\boldsymbol{W}_T^{lin};\mathcal{D})$ is in the linear
702 space spanned by the network output gradient at initialization.

703 By applying the Cauchy-Schwartz inequality, we have that for any $\alpha \in (0,+\infty)$, the following
704 inequality holds.

$$\Phi(T)-\Phi(T-\Delta T) \leq \frac{1}{\Delta T}\int_{T-\Delta T}^{T}\frac{1}{\alpha}\mathbb{E}[\|\nabla\mathcal{L}(\boldsymbol{W}_T^{lin};\mathcal{D})\|_2^2+\frac{\alpha}{4}\mathbb{E}[\|\boldsymbol{W}_T^{lin}-\boldsymbol{W}_\tau^{lin}\|_{M_0}^2]d\tau \quad (61)$$

$$\leq \frac{1}{\alpha}\max_{\tau\in[T-\Delta T,T]}\mathbb{E}[\|\nabla\mathcal{L}(\boldsymbol{W}_T^{lin};\mathcal{D})\|_2^2+\frac{\alpha}{4}\max_{\tau\in[T-\Delta T,T]}\mathbb{E}[\|\boldsymbol{W}_T^{lin}-\boldsymbol{W}_\tau^{lin}\|_{M_0}^2] \quad (62)$$

705 By Eq. (8), we have that when the data is normalized,

$$\mathbb{E}[\|\nabla\mathcal{L}(\boldsymbol{W}_T^{lin};\mathcal{D})\|_2^2 \leq 2B \quad (63)$$

706 where $B = |\mathcal{Y}|\left(\prod_{i=1}^{L-1}\frac{\beta_i m_i}{2}\right)\sum_{l=1}^{L}\frac{\beta_L}{\beta_l}$. We now only need to bound $\mathbb{E}[\|\boldsymbol{W}_T^{lin}-\boldsymbol{W}_\tau^{lin}\|_{M_0}^2]$. By
707 similar argument as (57), for any $t \geq \tau$, we have that

$$\frac{\partial}{\partial t}\mathbb{E}\left[\|\boldsymbol{W}_t^{lin}-\boldsymbol{W}_\tau^{lin}\|_{M_0}^2\right] = -2\mathbb{E}[\langle\nabla\mathcal{L}(\boldsymbol{W}_t^{lin}),\Pi_{M_0}(\boldsymbol{W}_t^{lin}-\boldsymbol{W}_\tau^{lin})\rangle]+\sigma^2\mathbb{E}[rank(M_0)]$$

$$\leq \frac{1}{\gamma}\mathbb{E}[\|\nabla\mathcal{L}(\boldsymbol{W}_t^{lin})\|_2^2]+\gamma\mathbb{E}[\|\boldsymbol{W}_t^{lin}-\boldsymbol{W}_\tau^{lin}\|_{M_0}^2]+\sigma^2\mathbb{E}[rank(M_0)]$$

$$\leq \gamma\mathbb{E}[\|\boldsymbol{W}_t^{lin}-\boldsymbol{W}_\tau^{lin}\|_{M_0}^2]+\frac{2B}{\gamma}+\sigma^2\mathbb{E}[rank(M_0)]$$

708 This is a linear ODE and can be solve closed formly as follows for $t \geq \tau$.

$$\mathbb{E}\left[\|\boldsymbol{W}_t^{lin}-\boldsymbol{W}_\tau^{lin}\|_{M_0}^2\right] \leq (\frac{2B}{\gamma^2}+\frac{\sigma^2\mathbb{E}[rank(M_0)]}{\gamma})(e^{\gamma(t-\tau)}-1)$$

709 Since the above equation holds for any $\tau \leq t$, by setting $\tau = T-\Delta T$ and $t = T$ we have that

$$\max_{\tau\in[T-\Delta T,T]}\mathbb{E}[\|\boldsymbol{W}_T^{lin}-\boldsymbol{W}_\tau^{lin}\|_{M_0}^2] \leq (\frac{2B}{\gamma^2}+\frac{\sigma^2\mathbb{E}[rank(M_0)]}{\gamma})(e^{\gamma\Delta T}-1) \quad (64)$$

$$\leq 4B(\Delta T)^2+2\sigma^2\mathbb{E}[rank(M_0)]\Delta T \quad (65)$$

710 where (65) is by setting $\gamma = \frac{1}{\Delta T}$ and by $e^1-1 \leq 2$. By combining (62), (63) and (65), we have that

$$\Phi(T)-\Phi(T-\Delta T) \leq \frac{2B}{\alpha}+\alpha B(\Delta T)^2+\frac{\alpha\sigma^2\mathbb{E}[rank(M_0)]\Delta T}{2}$$

$$= \frac{3R}{2T}+\frac{\sigma^2\mathbb{E}[rank(M_0)]}{2} \quad (66)$$

711 where the last equation (66) is by setting $\alpha = \frac{2B}{R}T$ and $\Delta = \frac{R}{2BT^2}$ (note that here $B$ and $R$ are as
712 given in the proposition statement). By combining (59) and (66) and using that $\Delta = \frac{R}{2BT^2}$, we have

$$\mathbb{E}[\mathcal{L}(\boldsymbol{W}_T^{lin})] = \Phi(T) = \Phi(T)-\Phi(T-\Delta T)+\Phi(T-\Delta T)$$

$$\leq \Phi(0)+\frac{\sigma^2\mathbb{E}[rank(M_0)]}{2}\left(1+\log\frac{2BT^2}{R}\right)+\frac{3R}{2T}$$

713 Observe that $\Phi(0) = \mathbb{E}[\mathcal{L}(\bar{\boldsymbol{W}}_T^{lin})]$, therefore by using Lemma D.1 we have that

$$\mathbb{E}[\mathcal{L}(\boldsymbol{W}_T^{lin})] \leq \mathbb{E}[\mathcal{L}(\boldsymbol{W}_0^*;\mathcal{D})]+\frac{2R}{T}+\frac{\sigma^2\mathbb{E}[rank(M_0)]}{2}\left(1+\log\frac{2BT^2}{R}\right)$$

714 $\qquad\qquad\qquad\qquad\qquad\qquad\qquad\qquad\qquad\qquad\qquad\qquad\qquad\qquad\qquad\qquad\qquad\qquad\qquad\square$

## D.3 Deferred proof for Proposition 5.3

**Proposition 5.3** (Bounding lazy training distance via smallest eigenvalue of the NTK matrix). Under the data and network regularity Assumption 2.1, if the width $m_1 = \cdots = m_{L-1} = \Omega(n)$ is sufficiently large, then there exists an optimal solution $\boldsymbol{W}_0^{\frac{1}{n^2}}$ that satisfies $\mathcal{L}_0^{lin}(\boldsymbol{W}_0^{\frac{1}{n^2}}) \leq \frac{1}{n^2}$ and satisfies

$$\tilde{R} = \mathbb{E}[\|\boldsymbol{W}_0^{\frac{1}{n}} - \boldsymbol{W}_0\|_2^2] \leq \begin{cases} \tilde{O}(\frac{n}{d \cdot 2^L \cdot (m(L-2)+1)}) & \text{for NTK initialization} \\ \tilde{O}(\frac{n}{2^L m(L-1)}) & \text{for He initialization} \\ \tilde{O}(\frac{n}{m(L-1)}) & \text{for LeCun initialization} \end{cases} \tag{67}$$

*Proof.* Given arbitrary initialization parameters $\boldsymbol{W}_0$, we first construct an solution $\boldsymbol{W}_0^{\frac{1}{n^2}}$ that is nearly optimal for the ERM problem over $\mathcal{L}_0^{lin}(\boldsymbol{W})$. Specifically, let $\boldsymbol{W}_0^{\frac{1}{n^2}}$ have the following expression.

$$\boldsymbol{W}_0^{\frac{1}{n^2}} - \boldsymbol{W}_0 = M_0^\dagger \begin{pmatrix} 2\ln n \cdot y_1 - \boldsymbol{f}_{\boldsymbol{W}_0}(\boldsymbol{x}_1) \\ \vdots \\ 2\ln n \cdot y_n - \boldsymbol{f}_{\boldsymbol{W}_0}(\boldsymbol{x}_n) \end{pmatrix} \tag{68}$$

where $M_0 = \begin{pmatrix} \nabla \boldsymbol{f}_{\boldsymbol{W}_0}(\boldsymbol{x}_1)^\top \\ \vdots \\ \nabla \boldsymbol{f}_{\boldsymbol{W}_0}(\boldsymbol{x}_n)^\top \end{pmatrix}$ is the NTK feature matrix at initialization and $\dagger$ denotes the pseudo-inverse. By random Gaussian initialization, and by the data regularity assumption Assumption 2.2, we have that $rank(M_0) = n$ with probability one, therefore $M_0^\dagger = M_0^\top (M_0 M_0^\top)^{-1}$

and $\begin{pmatrix} \boldsymbol{f}_{\boldsymbol{W}_0^{\frac{1}{n^2}}}(\boldsymbol{x}_1) \\ \vdots \\ \boldsymbol{f}_{\boldsymbol{W}_0^{\frac{1}{n^2}}}(\boldsymbol{x}_n) \end{pmatrix} = \begin{pmatrix} 2\ln n \cdot y_1 \\ \vdots \\ 2\ln n \cdot y_n \end{pmatrix}$ with probability one. By further using the definition of cross-entropy loss for the single-output setting, we have that the solution $\boldsymbol{W}_0^{\frac{1}{n^2}}$ satisfies the following inequality.

$$\mathcal{L}_0^{lin}(\boldsymbol{W}_0^{\frac{1}{n^2}}) = \log(1 + \exp(-2\ln n)) < \frac{1}{n^2} \tag{69}$$

We now only need to prove that the solution $\boldsymbol{W}_0^{\frac{1}{n^2}}$ is close to the initialization parameters $\boldsymbol{W}_0$ in expected $\ell_2$ norm. By applying the holder inequality on (68), we have that

$$\tilde{R} = \mathbb{E}[\|\boldsymbol{W}_0^{\frac{1}{n^2}} - \boldsymbol{W}_0\|_2^2] \leq \mathbb{E}\left[ \|M_0^\dagger\|_2^2 \cdot \left\| \begin{pmatrix} 2\ln n \cdot y_1 - \boldsymbol{f}_{\boldsymbol{W}_0}(\boldsymbol{x}_1) \\ \vdots \\ 2\ln n \cdot y_n - \boldsymbol{f}_{\boldsymbol{W}_0}(\boldsymbol{x}_n) \end{pmatrix} \right\|_2^2 \right] \tag{70}$$

$$\leq \mathbb{E}\left[ \frac{1}{\lambda_{min}(M_0 M_0^\top)} \cdot \left\| \begin{pmatrix} 2\ln n \cdot y_1 - \boldsymbol{f}_{\boldsymbol{W}_0}(\boldsymbol{x}_1) \\ \vdots \\ 2\ln n \cdot y_n - \boldsymbol{f}_{\boldsymbol{W}_0}(\boldsymbol{x}_n) \end{pmatrix} \right\|_2^2 \right] \tag{71}$$

We now prove bounds for the two terms on the right hand side separately. For the first term, when the width is sufficiently large $m = \Omega(n)$, by existing bound for the smallest eigenvalue of the NTK matrix $M_0 M_0^\top$ in [33, Theorem 4.1], we have that with high probability

$$\frac{1}{\lambda_{min}(M_0 M_0^\top)} \leq O\left( \frac{1}{\left(d \prod_{l=1}^{L-1} m_l\right) \cdot \left(\prod_{l=1}^{L} \beta_l\right) \cdot \left(\sum_{l=2}^{L} \beta_l^{-1}\right)} \right) \tag{72}$$

For the second term, by Cauchy-Schwarz inequality, we have that

$$\mathbb{E}\left[\left\|\begin{pmatrix} 2\ln n \cdot y_1 - \boldsymbol{f}_{\boldsymbol{W}_0}(\boldsymbol{x}_1) \\ \vdots \\ 2\ln n \cdot y_n - \boldsymbol{f}_{\boldsymbol{W}_0}(\boldsymbol{x}_n) \end{pmatrix}\right\|_2^2\right] \leq 2 \cdot (2\ln n)^2 \sum_{i=1}^n y_i^2 + 2\sum_{i=1}^n \mathbb{E}[\|\boldsymbol{f}_{\boldsymbol{W}_0}(\boldsymbol{x}_1)\|_2^2]$$

$$= 8n(\ln n)^2 + 2\sum_{i=1}^n \mathbb{E}[\|\boldsymbol{f}_{\boldsymbol{W}_0}(\boldsymbol{x}_1)\|_2^2]$$

By further using Eq. (44), we have that

$$\mathbb{E}\left[\left\|\begin{pmatrix} 2\ln n \cdot y_1 - \boldsymbol{f}_{\boldsymbol{W}_0}(\boldsymbol{x}_1) \\ \vdots \\ 2\ln n \cdot y_n - \boldsymbol{f}_{\boldsymbol{W}_0}(\boldsymbol{x}_n) \end{pmatrix}\right\|_2^2\right] = O\left(n(\ln n)^2 + n\prod_{i=1}^L \frac{\beta_i m_i}{2}\right) \qquad (73)$$

Therefore, by plugging Eq. (72) and Eq. (73) into Eq. (71), and by considering input dimension $d$ as const, we have that

$$\tilde{R} = O\left(\frac{n(\ln n)^2 + n\prod_{i=1}^L \frac{\beta_i m_i}{2}}{\left(d\prod_{l=1}^{L-1}\beta_l m_l\right) \cdot \beta_L \cdot \left(\sum_{l=2}^L \beta_l^{-1}\right)}\right)$$

By plugging the choice of initialization variance $\beta_1, \cdots, \beta_L$ for NTK, He and LeCun initialization into the above equation, for single output network with $L \geq 2$, we have that

$$\tilde{R} = \begin{cases} \tilde{O}(\frac{n}{d \cdot 2^L \cdot (m(L-2)+1)}) & \text{for NTK initialization} \\ \tilde{O}(\frac{n}{2^L m(L-1)}) & \text{for He initialization} \\ \tilde{O}(\frac{n}{m(L-1)}) & \text{for LeCun initialization} \end{cases}$$

$\square$

## D.4 Deferred proof for Corollary 5.4

**Corollary 5.4** (Privacy utility trade-off for last iterate)**.** Assume that the data and network regularity Assumption 2.2 holds. Assume that all the conditions and definition for constants in Proposition 5.1 holds. Then by setting $\sigma^2 = \frac{2BT}{\varepsilon n^2}$ and $T = \sqrt{\frac{2\varepsilon n \tilde{R}}{B}}$, we have that running Langevin diffusion for time $T$ satisfies bound KL divergence $\epsilon$, and has empirical excess risk upper bounded by

$$\mathbb{E}[\mathcal{L}(\boldsymbol{W}_T^{lin})] \leq \mathcal{O}\left(\frac{1}{n^2} + \sqrt{\frac{B\tilde{R}}{\varepsilon n}}\log(\varepsilon n)\right) \qquad (74)$$

where $B$ is the gradient norm constant Eq. (9), and $\tilde{R}$ is the approximate lazy training distance in Eq. (12). A summary of $B$ and $\tilde{R}$ under different initializations is in Table 1.

*Proof.* By setting $\boldsymbol{W}_0^* = \boldsymbol{W}_0^{\frac{1}{n^2}}$ in Proposition 5.3, we have that

$$\mathbb{E}[\mathcal{L}(\boldsymbol{W}_T^{lin})] - \mathbb{E}[\mathcal{L}(\boldsymbol{W}_0^{\frac{1}{n^2}}; \mathcal{D})] \leq \frac{2\tilde{R}}{T} + \frac{\sigma^2 \mathbb{E}[rank(M_0)]}{2}\left(1 + \log\frac{2BT^2}{\tilde{R}}\right)$$

By Proposition 5.3, we have that $\mathbb{E}[\mathcal{L}(\boldsymbol{W}_0^{\frac{1}{n^2}}; \mathcal{D})] \leq \frac{1}{n^2}$, therefore

$$\mathbb{E}[\mathcal{L}(\boldsymbol{W}_T^{lin})] \leq \frac{1}{n^2} + \frac{2\tilde{R}}{T} + \frac{\sigma^2 \mathbb{E}[rank(M_0)]}{2}\left(1 + \log\frac{2BT^2}{\tilde{R}}\right) \qquad (75)$$

By plugging $\sigma^2 = \frac{2BT}{\varepsilon n^2}$ and $T = \sqrt{\frac{2\varepsilon n \tilde{R}}{B}}$ into (75), and by rank$(M_0) \leq n$, we have that

$$\mathbb{E}[\mathcal{L}(\boldsymbol{W}_T^{lin})] \leq \frac{1}{n^2} + \sqrt{\frac{2B\tilde{R}}{\varepsilon n}}\left(2 + \log(4\varepsilon n)\right) \qquad (76)$$

$\square$

## D.5 Deferred proofs for Proposition 5.5

We will first need to prove following lemma to bound the uniform stability for the model parameters during training linearized network.

**Lemma D.2** (Uniform stability for training linearized network). *Assume that the data and network regularity Assumption 2.2 holds. Then it satisfies that*

$$\mathbb{E}\left[\|\boldsymbol{W}_T - \boldsymbol{W}_T'\|^2\right] \leq \frac{2B}{n^2}T^2 \tag{77}$$

*where $B = |\mathcal{Y}|\left(\prod_{i=1}^{L-1}\frac{\beta_i m_i}{2}\right)\sum_{l=1}^{L}\frac{\beta_L}{\beta_l}$ is the constant defined in (37) for network output gradient norm bound at initialization.*

*Proof.* By definition, and by coupling the choice of Gaussian noise in the two Langevin diffusion processes $(\boldsymbol{W}_t)_{t \in [0,T]}$ and $(\boldsymbol{W}_t')_{t \in [0,T]}$, we have that

$$\frac{\partial \mathbb{E}\left[\|\boldsymbol{W}_t - \boldsymbol{W}_t'\|^2\right]}{\partial t} = \lim_{\eta \to 0} \frac{\mathbb{E}[\|\boldsymbol{W}_{t+\eta} - \boldsymbol{W}_{t+\eta}'\|^2] - \mathbb{E}[\|\boldsymbol{W}_t - \boldsymbol{W}_t'\|^2]}{\eta}$$

$$= \lim_{\eta \to 0} \frac{\mathbb{E}[\|\boldsymbol{W}_t - \eta\nabla\mathcal{L}(\boldsymbol{W}_t;\mathcal{D}) - \boldsymbol{W}_t' + \eta\nabla\mathcal{L}(\boldsymbol{W}_t';\mathcal{D}')\|^2] - \mathbb{E}[\|\boldsymbol{W}_t - \boldsymbol{W}_t'\|^2]}{\eta}$$

$$= \lim_{\eta \to 0} \frac{\eta^2\mathbb{E}[\|\nabla\mathcal{L}(\boldsymbol{W}_t;\mathcal{D}) - \nabla\mathcal{L}(\boldsymbol{W}_t';\mathcal{D}')\|^2] - 2\eta\mathbb{E}[\langle\nabla\mathcal{L}(\boldsymbol{W}_t;\mathcal{D}) - \nabla\mathcal{L}(\boldsymbol{W}_t';\mathcal{D}'), \boldsymbol{W}_t - \boldsymbol{W}_t'\rangle]}{\eta}$$

$$= -2\mathbb{E}[\langle\nabla\mathcal{L}(\boldsymbol{W}_t;\mathcal{D}) - \nabla\mathcal{L}(\boldsymbol{W}_t';\mathcal{D}'), \boldsymbol{W}_t - \boldsymbol{W}_t'\rangle] \leq 2\sqrt{\frac{2B}{n^2}\cdot\mathbb{E}\left[\|\boldsymbol{W}_T - \boldsymbol{W}_T'\|^2\right]}$$

where the last inequality is by holder's inequality and by using Lemma 4.1. By solving the ordinary differential equation with boundary condition $\mathbb{E}\left[\|\boldsymbol{W}_0 - \boldsymbol{W}_0'\|^2\right] = 0$, we have that

$$\mathbb{E}\left[\|\boldsymbol{W}_T - \boldsymbol{W}_T'\|^2\right] \leq \frac{2B}{n^2}T^2 \tag{78}$$

$\square$

We are now ready to prove our excess population risk bound for training linearized network.

**Proposition 5.5.** Denote $R_0(\boldsymbol{W}) = \mathbb{E}_{(\boldsymbol{x},\boldsymbol{y})\in pop}[\ell(f_{\boldsymbol{W}_0}(\boldsymbol{x}) + \frac{\partial f_{\boldsymbol{W}_0}(\boldsymbol{x})}{\partial \boldsymbol{W}_0}(\boldsymbol{W} - \boldsymbol{W}_0); \boldsymbol{y})]$ as the population risk of linearized network expanded at initialization vector $\boldsymbol{W}_0$ over population data distribution $pop$. Then under the conditions of Corollary 5.4, we have that for any dataset $\mathcal{D}$ of size $n$, the following excess population risk upper bound holds.

$$\mathbb{E}[R_0(\boldsymbol{W}_T)] - \mathbb{E}[\mathcal{L}(\boldsymbol{W}_{pop,0}^*;\mathcal{D})] \leq O\left(\frac{1}{n^2} + \sqrt{\frac{B\tilde{R}}{\varepsilon n}}(\log(\varepsilon n) + \varepsilon)\right) \tag{79}$$

where the expectation is over the randomness of sampling the training dataset $\mathcal{D} \sim pop^n$ from the data population and the random coins for the Langevin diffusion training algorithm, and $\boldsymbol{W}_{pop,0}^* = \arg\min_{\boldsymbol{W}} R_0(\boldsymbol{W})$ is the optimal solution for the population risk minimization problem.

*Proof.* By the uniform stability method [24, Theorem 2.2], we have the following generalization error upper bound holds.

$$\alpha_{gen} = |\mathbb{E}[R_0(\boldsymbol{W}_T)] - \mathbb{E}_{\mathcal{D}\sim pop^n}[\mathcal{L}(\boldsymbol{W}_T;\mathcal{D})]| \leq \max_{z,D,D'}\mathbb{E}\left[\ell(f_{\boldsymbol{W}_T}(\boldsymbol{x}_z);\boldsymbol{y}_z) - \ell(f_{\boldsymbol{W}_T'}(\boldsymbol{x}_z);\boldsymbol{y}_z)\right] \tag{80}$$

where $\boldsymbol{z} = (\boldsymbol{x}_z, \boldsymbol{y}_z)$ is an arbitrary data point in the population data distribution $pop$. By convexity, we further have the following bound for the uniform stability.

$$\ell(f_{\boldsymbol{W}_T}(\boldsymbol{x}_z);\boldsymbol{y}_z) - \ell(f_{\boldsymbol{W}_T'}(\boldsymbol{x}_z);\boldsymbol{y}_z) \leq \nabla\ell(f_{\boldsymbol{W}_T}(\boldsymbol{x}_z);\boldsymbol{y}_z)^\top(\boldsymbol{W}_T - \boldsymbol{W}_T') \tag{81}$$

$$\leq \sqrt{\|\nabla\ell(f_{\boldsymbol{W}_T}(\boldsymbol{x}_z);\boldsymbol{y}_z)\|^2\|\boldsymbol{W}_T - \boldsymbol{W}_T'\|^2} \tag{82}$$

By Eq. (37) and Lemma D.2, we further have that

$$\mathbb{E}[\ell(f_{\boldsymbol{W}_T}(\boldsymbol{x}_z); \boldsymbol{y}_z) - \ell(f_{\boldsymbol{W}_T'}(\boldsymbol{x}_z); \boldsymbol{y}_z)] \leq \sqrt{2B} \cdot \sqrt{\frac{2B}{n^2}T^2} = \frac{2B}{n}T \tag{83}$$

Therefore, by plugging the above equation into (80), we have that the generalization error satisfies

$$\alpha_{gen} = |\mathbb{E}_{\mathcal{D}\sim pop^n}[\mathcal{L}(\boldsymbol{W}_T; \mathcal{D})] - \mathbb{E}[R(\boldsymbol{W}_T)]| \leq \frac{2BT}{n} \leq 2\sqrt{\frac{2\varepsilon BR}{n}} \tag{84}$$

where the last inequality is by plugging our choice of $T = \sqrt{\frac{2\varepsilon n R}{B}}$ into the equation. On the other hand, by Proposition 5.1, we have that the empirical risk is upper bounded as follows.

$$\mathbb{E}_{\mathcal{D}\sim pop^n}[\mathcal{L}(\boldsymbol{W}_T; \mathcal{D})] \leq O\left(\sqrt{\frac{B\tilde{R}}{\varepsilon n}}\log(\varepsilon n)\right)$$

$$\leq \mathbb{E}[\mathcal{L}(\boldsymbol{W}_{pop,0}^*; \mathcal{D})] + O\left(\sqrt{\frac{B\tilde{R}}{\varepsilon n}}\log(\varepsilon n)\right) \tag{85}$$

Combining the generalization error term (84) and the excess empirical risk term (85) suffice to prove the equation in the statement. $\qquad\square$

# E  Discussion on extending our results to Noisy GD with constant step-size

In this section, we discuss how to extend our privacy analyses to noisy GD with constant step-size. Specifically, we only need to extend the KL composition theorem under possibly unbounded gradient difference, i.e., Theorem 3.1, to the noisy GD algorithm.

**Theorem E.1** (KL composition for noisy GD under possibly unbounded gradient difference)**.** *Let the iterative update in noisy GD algorithm be defined by:* $\boldsymbol{W}_{(k+1)} = \boldsymbol{W}_{(k)} - \eta\nabla\mathcal{L}(\boldsymbol{W}_{(k)}; \mathcal{D}) + \sqrt{2\eta\sigma^2}Z_k$, *where* $Z_k \sim \mathcal{N}(0, \mathbb{I})$. *Then the KL divergence between running noisy GD for DNN* (2) *on neighboring datasets* $\mathcal{D}$ *and* $\mathcal{D}'$ *satisfies*

$$KL(\boldsymbol{W}_{(K)}, \boldsymbol{W}_{(K)}') \leq \frac{1}{2\sigma^2}\sum_{k=0}^{K-1}\eta \cdot \mathbb{E}\left[\left\|\nabla\mathcal{L}(\boldsymbol{W}_{(k)}; \mathcal{D}) - \nabla\mathcal{L}(\boldsymbol{W}_{(k)}; \mathcal{D}')\right\|_2^2\right]. \tag{86}$$

*Proof.* Denote $p_{(k)}$ as the distribution of model parameters after running noisy GD on dataset $\mathcal{D}$ with $k$ steps, and similarly denote $p_{(k)}'$ as the distribution of model parameters after running noisy GD on dataset $\mathcal{D}'$ with $k$ steps. Then by the data processing inequality for KL divergence [45, Theorem 9] (with the data processing operation given by $(\boldsymbol{W}_{(k)}, \boldsymbol{W}_{(k+1)}) \to \boldsymbol{W}_{(k)}$), we have that

$$KL(p_{(k+1)}, p_{(k+1)}') \leq KL(p_{(k),(k+1)}, p_{(k),(k+1)}'), \tag{87}$$

where $p_{(k),(k+1)}$ denotes the joint distribution of $(\boldsymbol{W}_{(k)}, \boldsymbol{W}_{(k+1)})$, and $p_{(k),(k+1)}'$ denotes the joint distribution of $(\boldsymbol{W}_{(k)}', \boldsymbol{W}_{(k+1)}')$. Now we expand the term $KL(p_{(k),(k+1)}, p_{(k),(k+1)}')$ by the Bayes rule as follows.

$$KL(p_{(k),(k+1)}, p_{(k),(k+1)}') \tag{88}$$

$$=\mathbb{E}_{p_{(k),(k+1)}(\boldsymbol{W}_{(k)}, \boldsymbol{W}_{(k+1)})}\left[\log\left(\frac{p_{(k+1)|(k)}(\boldsymbol{W}_{(k+1)}|\boldsymbol{W}_{(k)})p_{(k)}(\boldsymbol{W}_{(k)})}{p_{(k+1)|(k)}'(\boldsymbol{W}_{(k+1)}|\boldsymbol{W}_{(k)})p_{(k)}'(\boldsymbol{W}_{(k)})}\right)\right]$$

$$=\mathbb{E}_{p_{(k),(k+1)}(\boldsymbol{W}_{(k)}, \boldsymbol{W}_{(k+1)})}\left[\log\left(\frac{p_{(k+1)|(k)}(\boldsymbol{W}_{(k+1)}|\boldsymbol{W}_{(k)})}{p_{(k+1)|(k)}'(\boldsymbol{W}_{t+\eta}|\boldsymbol{W}_t)}\right)\right] + \mathbb{E}_{p_{(k)}(\boldsymbol{W}_{(k)})}\left[\log\left(\frac{p_{(k)}(\boldsymbol{W}_t)}{p_{(k)}'(\boldsymbol{W}_{(k)})}\right)\right]$$

$$=\mathbb{E}_{p_{(k)}(\boldsymbol{W}_{(k)})}\left[KL(p_{(k+1)|(k)}, p_{(k+1)|(k)}')\right] + KL(p_{(k)}, p_{(k)}') \tag{89}$$

Observe that $p_{(k+1)|(k)}, p'_{(k+1)|(k)}$ are two Gaussian distributions with per-dimensional variance $\sigma^2$, due to the conditioning on the same model parameters $\boldsymbol{W}_{(k)}$ at iteration $k$. Therefore, by computing the KL divergence between two multivariate Gaussians, we have that

$$KL(p_{(k),(k+1)}, p'_{(k),(k+1)}) = \frac{1}{2\sigma^2} \cdot \eta \cdot \left\| \nabla \mathcal{L}(\boldsymbol{W}_{(k)}; \mathcal{D}) - \nabla \mathcal{L}(\boldsymbol{W}_{(k)}; \mathcal{D}') \right\|_2^2 \qquad (90)$$

Therefore, by plugging Eq. (90) into Eq. (89) and Eq. (87), we have that

$$KL(p_{(k+1)}, p'_{(k+1)}) \leq \frac{\eta}{2\sigma^2} \mathbb{E}\left[ \left\| \nabla \mathcal{L}(\boldsymbol{W}_{(k)}; \mathcal{D}) - \nabla \mathcal{L}(\boldsymbol{W}_{(k)}; \mathcal{D}') \right\|_2^2 \right] + KL(p_{(k)}, p'_{(k)}) \qquad (91)$$

By summing (91) over $k = 0, \cdots, K-1$ and observing that $KL(p_{(0)}, p'_{(0)}) = 0$ (as the initialization distribution is the same between noisy GD on $\mathcal{D}$ and $\mathcal{D}'$), we finish the proof for Eq. (86). $\qquad\square$