# OpenReview forum: "Initialization Matters: Privacy-Utility Analysis of Overparameterized Neural Networks"
_NeurIPS.cc/2023/Conference — NeurIPS 2023 poster_

### Official Review · Reviewer_R6b6 · 2023-06-25

**Soundness:** 2 fair
**Presentation:** 3 good
**Contribution:** 3 good
**Rating:** 5
**Confidence:** 4

**Summary:**

This work aims to connect the structure of a neural network and its initialization scheme to the privacy guarantees it can offer. KL divergence is used as privacy metric, under the settings of Langevine diffusion.

**Strengths:**

The paper aims to solve a very important problem, i.e. the effects of overparameterization on privacy for overparametrized deep learning models. The analysis through KL divergence allows the problem to be treated analytically, and to connect the structure of the network to privacy bounds.

**Weaknesses:**

fifth line of caption of Table 1: $n$ and $\phi$ appear here for the first time. It looks like they represent the number of training samples and your regularity condition parameter respectively. Probably better to mention this in the caption as well.

same line: $m = \Omega(n^{14} L^{16} \phi^{-4})$. To me this assumption looks severely irrealistic for any deep learning model.

The trade-offs in the table are simply upper-bounds. It is unclear to me how these upper-bounds can be considered tight, therefore telling anything about a possible comparison between different initializations / widths.

Activation function and regularity parameter have the same notation $\phi$

Assumption 2.1 enforces the samples to respect $\| x_i\|_2 \leq 1$. This scaling is quite innatural considering your initializations schemes. which are designed to renormalize the preactivations when the input norms scale with $\sqrt d$ (more natural assumption).

End of line 151, are the subscripts a typo here?

It looks that the main bound in Lemma 4.1 derives from a worst case, per-layer computation, under the assumption of ReLU activation. This suggests the bound to be non tight, and therefore to not be informative about how different initializations really affect the preservation of privacy.

Lack of experimental validation on the claims in the paper.

**Questions:**

third line of caption of figure 1: you write $d = m$ first, and then you mention that you assume $m \geq d$. Is this a typo?

would your results still hold if we consider a more natural scaling of the data $\| x_i\|_2 \leq \sim d$?

Your results are valid only in the case of ReLU networks. Is the analysis generalizable?

In the discussion after Lemma 3.2 you mention that there are two important keys to control privacy: initialization and training time. You focus your analysis on the first one, and you motivate why the second component is not relevant in the case of small training time. Correct? If yes, how do you verify this hypothesis? Is it a consequence of lazy training?

Your Lemma 4.1 looks to exploit a worst-case bound over the layers of the network, plus the property of the ReLU activation. Can you confirm?

**Limitations:**

No limitations discussion needed

---

> ### Author Rebuttal · Authors · 2023-08-09
>
> We thank the reviewer for pointing out the typos and repeated symbols. Below we clarify the questions and concerns.
>
> > same line: $m = \Omega(n^{14}L^{16}\phi^{-4})$. To me this assumption looks severely irrealistic for any deep learning model.
>
> This condition comes as is from the prior analysis [48] for lazy training distance in the deep ReLU network. This is a standard and fair condition on the polynomial order of $n$ and $L$ in the deep learning theory community.
>
> Since then, there have been various improved analyses on how much over-parameterization is enough in deep learning theory. With some modifications, e.g., the excess risk under a high probability setting, the width requirement in our work can be relaxed from the $\Omega(ploy(n))$ order to $\Omega(n)$, see Appendix D.3 for details.
>
> Relaxation on the width requirement is always an important direction in DL theory and we believe that any relevant advances for the lazy training distance analysis in the literature could be used off-the-shelf in Corollary 5.4.
>
>
> > The trade-offs in the table are simply upper-bounds. It is unclear to me how these upper-bounds can be considered tight, therefore telling anything about a possible comparison between different initializations / widths.
>
> To the best of our knowledge, there are no instance-specific lower bounds for linearized networks derived from fully connected ReLU networks. Nevertheless, our trade-off bounds are based on (and therefore at least as tight as) the prior known tight excess empirical risk bounds for generalized linear model [39]. We leave more fine-grained instance-specific lower bounds as an important open problem.
>
> Additionally, we have added numerical validation experiments in general response point 2 to validate our KL privacy bound under different width, depth and initialization. We observe that the numerical results are consistent with the trends predicted by Lemma 3.2 and Corollary 4.2. See [general response point 2](https://openreview.net/forum?id=IKvxmnHjkL&noteId=jh61g9pgBi) for details.
>
> > Would your results still hold if we consider a more natural scaling of the data $|x_i| \leq d$?
>
> This assumption is only for simplicity of presentation. Our bounds readily extend to $|x_i| \leq d$, up to a multiplicative factor of $d$. Since $d$ remains constant as the network width and depth increases, it does not affect our conclusion of how overparameterization affects privacy and utility under different initializations.
>
> > End of line 151, are the subscripts a typo here?
>
> Yes, it is a typo. It should be $\forall (x_1, y_1), (x_2, y_2)\in D$ with $y_1\neq y_2$.
>
> > third line of caption of figure 1: you write $d=m$ first, and then you mention that you assume $m\geq d$. Is this a typo?
>
> This is not a typo. We use $m_l$ to refer to the width of layers $l$. In Figure 1 caption, we write $d = m_0$, meaning that the input 0-th layer has width equal to the data domain dimension. We also write $m_1 =...=m_{L-1} = m$, which means all hidden layers $1, ..., L-1$ have a same width $m$ that is larger than the data dimension $m\geq d$. We make it clearer in the revised version.
>
> > Your results are only valid for ReLU networks. Is the analysis generalizable?
>
> The primary challenge in extending our privacy constraints to different network functions lies in adapting our gradient norm constraint from Lemma 4.1. While we focus on dissecting ReLU activation for the sake of analytical clarity, the potential exists to broaden our outcomes to encompass alternative activation functions through analogous analytical calculations [a]. Furthermore, the applicability can be extended to other architectural designs such as ResNet [b,c] and CNN [d, e], leveraging the global convergence outcome. We consider the pursuit of these extensions a crucial avenue for future exploration.
>
> > In the discussion after Lemma 3.2 you mention that there are two important keys to control privacy: initialization and training time. You focus your analysis on the first one, and you motivate why the second component is not relevant in the case of small training time. Correct? If yes, how do you verify this hypothesis? Is it a consequence of lazy training?
>
> Yes, the second component does not dominate the privacy bound at small training time $T$, because its factor related to $T$ is $\frac{e^{(2+\beta^2)T} - (2+\beta^2)T}{2+\beta^2}$. This factor is is smaller than (sublinear to) $T$ for small $T\leq \frac{1}{2(2+\beta^2)}$, which is smaller than the first and the third components in Lemma 3.2 that are linear to $T$. We also numerically validate this point in general response point 2 and attached pdf. Intuitively, this phenomenon is related to lazy training.
>
> For the special case of linearized networks, there is no second component because the its gradient during training is upper bounded by the network’s gradient at initialization. (See Lemma 4.1 for more details.)
>
>
> > Your Lemma 4.1 looks to exploit a worst-case bound over the layers of the network, plus the property of the ReLU activation. Can you confirm?
>
> No, Lemma 4.1 is not a worst-case bound over the layers. Instead, it computes the exact expected squared gradient norm at initialization (Lemma C.3). The key insight is that the ratio between the squared norm of gradient at $l+1$-th layer and $l$-th layer follows a Chi-squared distribution multiplied by Bernoulli distribution, thus allowing a recursion equality over the layers. See general response point 1 for more explanations for intuition of this gradient norm bound.
>
> **References**
>
> [a] Generalization properties of NAS under activation and skip connection search. NeurIPS 2022.
>
> [b] Why Do Deep Residual Networks Generalize Better than Deep Feedforward Networks?---A Neural Tangent Kernel Perspective. NeurIPS 2020.
>
> [c] Generalization Ability of Wide Residual Networks
>
> [d] On Exact Computation with an Infinitely Wide Neural Net. NeurIPS 2019.
>
> [e] What Can Be Learnt With Wide Convolutional Neural Networks? . ICML 2023.

---

> > ### Comment · Reviewer_R6b6 · 2023-08-13
> >
> > I thank the authors for their detailed responses. I appreciate the introduction of experiments in support of your bounds, reason for which I increased my previous score.
> >
> > I have three follow-up questions:
> >
> > 1 - Regarding Equation (10), this is an upper-bound. In principle, taken as such, it wouldn't give any information about which initialization would provide smaller KL divergence. However, if considered tight, it might suggest the conclusions you draw at the end of Section 4, after Corollary 4.2. Now, with the new numerical evaluation (in the new experiments), this hypothesis is more supported. Is this correct? If so, I would really suggest to modify the presentation in the revised version, as the word "suggest" at line 257 is at the moment not very motivated.
> >
> > 2 - How do you compute the KL divergence in your experiments?
> >
> > 3 - KL divergence between two distributions in high dimension is expected to increase with the number of dimensions (not a very precise statement, but we would expect divergence to increase as more degrees of freedom are offered to the evolution of the weight distribution). This makes it reasonable to have smaller privacy guarantees (with this metric) for more over-parametrized networks. Can you elaborate on your contribution stated in line 44 based on this?

---

> > > ### Author Response · Authors · 2023-08-15
> > > **Replies to the follow-up questions**
> > >
> > > Thanks for the reply and detailed questions. We answer them below.
> > >
> > > > Q1: Eq (10) and numerical results
> > >
> > > Yes, equation (10) is an **upper bound** for gradient difference **throughout training**. Meanwhile, our gradient norm computation **at initialization** uses an equality (equation (35) in Lemma C.3), and is thus **exact**. Our numerical experiments validate the exact gradient norm computation at initialization, and are consistent with the trend predicted by Lemma 4.1 at small training time (thus supporting its tightness).
> > >
> > > According to your suggestions, we will make our presentation precise in the revised version, e.g., “Corollary 4.2 and Lemma 4.1 prove that the KL privacy upper bound ...". Besides, we will clarify that this trend of upper bound is consistent with numerical experiments at small training time.
> > >
> > > > Q2: How do you compute the KL divergence in your experiments?
> > >
> > > Our detailed numerical estimation procedures for noisy GD with $K$ steps, noise std $\sigma$ and learning rates {$\eta_k$}$_{k=1,\cdots,K}$ are as follows.
> > >
> > > 1. At each noisy GD iteration $k$, we **compute the maximal squared gradient difference norm** $s(k)$ between all possible neighboring datasets in CIFAR10. (See [general response point 1](https://openreview.net/forum?id=IKvxmnHjkL&noteId=jh61g9pgBi) for details about neighboring datasets.) We repeat this for $N$ noisy GD runs, and obtain $N$ squared gradient difference norm estimates  $s_1(k), \cdots, s_N(k)$ (with mean $\hat{s}(k)$ and standard deviation $std(k)$).
> > >
> > > 2. We then plug the squared gradient norm into the closed-form KL divergence equality $\frac{\lVert \text{offset} \rVert_2^2}{2\sigma^2}$ (e.g. see [a, Proposition 7]) between isotropic Gaussian and its offset (by learning rate $\eta_k$ times gradient difference $\hat{s}_k$) to **compute the one-step KL privacy loss for each iteration $k$**. This gives us mean $\frac{\eta_k\cdot \hat{s}(k)}{2\sigma^2}$ and std $\frac{\eta_k\cdot std(k)}{2\sigma^2}$.
> > >
> > > 3. Finally, we **compose (sum) KL divergence over all iterates** $k=1, \cdots, K$ of noisy GD, via the composition **equality** Equation (89). This gives us mean $\frac{\sum_{k=1}^K\eta_k\cdot \hat{s}(k)}{2\sigma^2}$ and std $\frac{\sum_{k=1}^K\eta_k\cdot std(k)}{2\sigma^2}$ for estimating the KL privacy loss in noisy GD, which we plot in numerical experiments.
> > >
> > > > Q3: KL divergence may not grow with the distribution dimension
> > >
> > > We guess the reviewer had a typo and meant “unreasonable”. To check this claim, it is valuable to first look at one simple example of KL divergence between two $d$-dimensional Gaussians $\mathcal{N}(\mu_1, \sigma^2I_d)$ and $\mathcal{N}(\mu_2, \sigma^2I_d)$, which is $\frac{\lVert \mu_1 - \mu_2\rVert_2^2}{2\sigma^2}$. Therefore, dimension does not factor in KL as long as $\lVert \mu_1 - \mu_2\rVert_2^2$ is small. We can show that the gradient difference (in noisy GD for overparameterized linearized network) achieves this under certain initializations (such as LeCun and Xavier), therefore our KL divergence bound (Corollary 4.2) **does not necessarily grow with the model dimension (increasing width and depth)**, as summarized in our contributions Line 44-52. We elaborate them below.
> > >
> > > (1) On the one hand, **increasing width $m$ always increases the gradient difference norm** ($B$ in Table 1) between the two distributions, and thus increases the KL divergence bound (Corollary 4.2). Intuitively, this is because
> > >
> > > the gradient at each layer is proportional to **the weighted sum of outputs from its preceding layer**, and thus increases with larger hidden-layer width.
> > >
> > > This is **consistent with intuition that the reviewer quoted**.
> > >
> > > (2) On the other hand, if we inverse proportionally set the per-layer initialization variance $\beta_l$ with the width $m_l$ (such that $\frac{\beta_lm_l}{2}<1$), and increase the network depth,  then the gradient difference norm ($B$ in Table 1) and thus the KL privacy bound (Corollary 4.2) decreases.   Intuitively, this is because of the output propagation across the layers in the network.
> > >
> > > For example, under LeCun initialization, the output values after each layer gets zeroed out with 1/2 probability. Consequently, the gradient norm at each layer decays exponentially with the number of preceding layers (depth).
> > >
> > > Such decrease of gradient norm under increasing depth is not a new phenomenon, and has been widely referred to as “gradient vanishing” in the literature.
> > >
> > > To this end, our contribution includes a general bound for the gradient norm (Lemma 4.1), under different initialization, and analyzing when it does not necessarily grow with model dimension. We'd also like to refer to [general response point 1](https://openreview.net/forum?id=IKvxmnHjkL&noteId=jh61g9pgBi) for more discussion about the intuition for this gradient norm bound.
> > >
> > > We hope our response can address the reviewer’s concern and are also happy to address any further comments.
> > >
> > > **Reference**
> > >
> > > [a] Mironov, Ilya. Rényi differential privacy. CSF 2017.

---

> > > > ### Comment · Reviewer_R6b6 · 2023-08-17
> > > >
> > > > I warmly thank the authors for their detailed response, which also clarified better parts of the work.
> > > >
> > > > It looks from your techniques that you are not taking into account the learning rate of the optimization. Different initializations (e.g. NTK vs LeCun) would reasonably require different scalings of the learning rate (see Remark1 of https://arxiv.org/pdf/1806.07572.pdf). Thus, I wonder if your results might have roots in a not fair comparison between two different initializations, as they would need to be compared with the properly rescaled learning rates.
> > > >
> > > > Also, I am a little confused on the NTK initialization you consider in Table1, as it seems different from the one defined in https://arxiv.org/pdf/1806.07572.pdf. Could you clarify this?

---

> > > > > ### Author Response · Authors · 2023-08-18
> > > > >
> > > > > Thank you for the follow-up questions. We answer them one-by-one below.
> > > > >
> > > > > ---
> > > > >
> > > > > > Q1: Learning rate choice
> > > > >
> > > > > We thank the reviewer for the interesting question.
> > > > >
> > > > > (1) The main effect of the learning rate on the theoretical results in Jacot et al. [a] is the equivalence between the neural network and the linear model (that is, the speed at which the neural network converges to the linear model). For detailed results, refer to [b, Section F] and  [c, Section 4] . This **does not materially affect the privacy utility results** for linearized networks in our paper, as we explain in (2) and (3) below.
> > > > >
> > > > > (2) Our main privacy utility analyses focus on the Langevin diffusion algorithm, which is a conceptual algorithm that serves as the limit of noisy GD as learning rate goes to zero. Therefore, our comparison for privacy utility trade-off in Table 1 **does not involve setting any learning rate for the algorithm** (which may be why the reviewer says our techniques do not take into account the learning rate of the optimization).
> > > > >
> > > > > (3) More importantly, our privacy and utility analysis extends seamlessly to a time-scaled Langevin diffusion algorithm, and **adding the time-scaling constant would not affect the privacy utility trade-off results** in Table 1. This is because, under changed time scaling, the lazy training distance constant $\tilde{R}$ does not change as the parameter space is the same, and the gradient norm constant $B$ and the loss scales up by a factor of $a$. This leads to the same excess empirical risk bound under fixed KL privacy budget as our current Corollary 5.4. Below we elaborate the details.
> > > > >
> > > > > - Consider a time-scaled Langevin diffusion algorithm: $dW_t = - a \nabla L(W_t; D)dt + \sqrt{2\sigma^2}dB_t$, where $a$ is a time-scaling constant that plays a similar role as learning rate in discrete noisy GD.
> > > > > - Observe that this time-scaled SDE is equivalent to Langevin diffusion Equation (4) if we replace $L(W; D)$ in Equation (4) with a new loss $\tilde{L}(W;D) = a \cdot L(W; D)$.
> > > > > - Therefore, to prove excess risk bound for time-scaled Langevin diffusion, we only need to use our Corollary 5.4 for the new scaled loss $\tilde{L}(W;D) = a\cdot L(W; D)$. This gives us
> > > > > $$
> > > > > \mathbb{E} [\tilde{L}({W}_{T}^{lin})] \leq \mathcal{O}\left(\frac{{\color{red}a}}{n^2} + \sqrt{\frac{{\color{red}a}\cdot B \tilde{R}}{\varepsilon n }} \log (\varepsilon n)\right),
> > > > > $$
> > > > > where we have highlighted the change due to the loss rescaling by time-scale constant $a$. By rewriting $\tilde{L} = a\cdot L$, we immediately obtain exactly the same excess empirical risk bound as Corollary 5.4. That is, adding the time-scaling constant $a$ would not affect the privacy utility trade-off result in Table 1.
> > > > >
> > > > >
> > > > >
> > > > >
> > > > >
> > > > > Accordingly, our comparison of privacy utility trade-off under different initialization is fair. We will add these clarifications in future revisions of the paper.
> > > > >
> > > > > ---
> > > > >
> > > > > > Q2: NTK initialization formulation
> > > > >
> > > > > Our initialization formulation follows Allen-Zhu et al. [2, Definition 2.3, arxiv v5], and the NTK initialization in [d] uses $N(0,1)$ and the scaling factor $1/\sqrt{m}$ at each layer (except the last layer). Allen-Zhu et al. [2, Theorem 5] proves the equivalence between these two different initializations.
> > > > >
> > > > > ---
> > > > >
> > > > > We hope our response will clarify the reviewer’s concern on learning rate and initialization.
> > > > >
> > > > > **References**
> > > > >
> > > > > [2] Allen-Zhu, Z., Li, Y., & Song, Z. (2018). A Convergence Theory for Deep Learning via Over-Parameterization. arXiv preprint arXiv:1811.03962.
> > > > >
> > > > > [a] Jacot, A., Gabriel, F., & Hongler, C. (2018). Neural tangent kernel: Convergence and generalization in neural networks. Advances in neural information processing systems, 31.
> > > > >
> > > > > [b] Arora, S., Du, S. S., Hu, W., Li, Z., Salakhutdinov, R. R., & Wang, R. (2019). On exact computation with an infinitely wide neural net. Advances in neural information processing systems, 32.
> > > > >
> > > > > [c] Du, Simon S., et al. "Gradient descent provably optimizes over-parameterized neural networks." arXiv preprint arXiv:1810.02054 (2018).

---

> > > > > > ### Comment · Reviewer_R6b6 · 2023-08-20
> > > > > >
> > > > > > I thank the authors for their explanation. I am satisfied, and I apologize again for the pedantic evaluation of the work in this review phase.
> > > > > >
> > > > > > I think further explanation about these points (maybe in the appendix) could be substantially improve the quality of the final version.
> > > > > >
> > > > > > I further increased my score, which I consider final.

---

> > > > > > > ### Author Response · Authors · 2023-08-21
> > > > > > >
> > > > > > > We thank the reviewer R6b6's feedback and support.
> > > > > > > According to your suggestions, we will include these explanations in our main text and appendix for better understanding in our final version.

---

### Official Review · Reviewer_NB2b · 2023-07-06

**Soundness:** 3 good
**Presentation:** 3 good
**Contribution:** 3 good
**Rating:** 7
**Confidence:** 4

**Summary:**

This paper presents the KL privacy bound for training fully connected networks and its linearized variant using the Langevin diffusion algorithm. The study delves into the impact of network width, depth, and initialization on privacy. Notably, the paper reveals that, with specific parameter initialization, the KL privacy bound can be enhanced by increasing the depth of the network. Additionally, the author provides a proof that the privacy-utility trade-off of linearized networks can be improved with depth when employing LeCun initialization.

**Strengths:**

1.The author investigates an intriguing problem that is likely to capture significant interest within the community.

2.The results presented in the paper are not only interesting but also coherent, aligning well with the research topic.

3.The paper is skillfully written, allowing for easy comprehension and smooth flow of ideas.

**Weaknesses:**

The paper lacks numerical experiments to validate the disparity between empirical KL privacy leakage and the privacy bound proposed in this paper. This would shed light on the tightness of the bound. Given that the arguments are based on the upper bound, providing numerical experiments demonstrating the evolution of empirical KL privacy leakage with varying depths and widths, aligned with the theoretical findings, would enhance the persuasiveness of the research.

**Questions:**

Questions:
1. In Lemma 3.2, Equation (7) suggests that for relatively large T, the dominant factor in the KL privacy bound is the fluctuation term of gradient differences, which appears to be independent of the differences between neighboring datasets $D$ and $D'$. Could the author provide further clarification on this matter?
2. Additionally, I am curious about the evolution of the rank of $M_T$ with respect to time T. The author briefly touches upon this aspect, but a more detailed explanation would be appreciated.

Some suggestions and corrections for the paper:
1. In Assumption 2.2, it would be preferable to use a commonly recognized constant symbol such as 'C' instead of '\phi' to denote a constant.
2. There seems to be some confusion in the notations in lines 151 and 152. Is it intended that x_1 = x, y = y_1, and y' = y_2?

**Limitations:**

See above.

---

> ### Author Rebuttal · Authors · 2023-08-09
>
> > The paper lacks numerical experiments to validate the disparity between empirical KL privacy leakage and the privacy bound proposed in this paper. This would shed light on the tightness of the bound. Given that the arguments are based on the upper bound, providing numerical experiments demonstrating the evolution of empirical KL privacy leakage with varying depths and widths, aligned with the theoretical findings, would enhance the persuasiveness of the research.
>
> We have added numerical experiments to validate our privacy bound under different width, depth and initialization. Please see [general response point 2](https://openreview.net/forum?id=IKvxmnHjkL&noteId=jh61g9pgBi) for a detailed discussion, and see the [attached pdf](https://openreview.net/attachment?id=jh61g9pgBi&name=pdf) for complementary plots.
>
> ---
>
> > In Lemma 3.2, Equation (7) suggests that for relatively large T, the dominant factor in the KL privacy bound is the fluctuation term of gradient differences, which appears to be independent of the differences between neighboring datasets $D$ and $D'$. Could the author provide further clarification on this matter?
>
> Yes, the dominating fluctuation term in Lemma 3.2 does not explicitly depend on the gradient difference between datasets $D$ and $D'$. However, it upper bounds the gradient difference at time $T$, due to the relaxed smoothness assumption of loss function, as we explain below.
>
> The key of our analysis in Lemma 3.2, is to reduce the problem of upper bounding the gradient difference at any training time T, to analyzing its two subcomponents. Let $(x, y)$ and $(x', y')$ be the differing data between neighboring datasets $D$ and $D'$, then $\left\lVert \nabla\ell(f_{W_t}(x); y)) - \nabla\ell(f_{W_t}(x'); y')\right\rVert_2^2\leq  \underbrace{2 \left\lVert \nabla\ell(f_{W_0}(x); y)) - \nabla\ell(f_{W_0}(x'); y')\right\rVert_2^2}\_{\text{gradient difference at initialization}} + 2 \beta^2 \underbrace{ \lVert W_t - W_0\rVert_{M_T}^2}\_{\text{parameters' change after training time $T$}}  + 2 c^2.$ This is by the Cauchy-Schwartz inequality, and by the relaxed smoothness assumption that $\lVert \nabla \ell(f_W(x); y)) - \nabla\ell(f_{W'}(x); y)\rVert_2 < \max(c, \beta \lVert W - W'\rVert_{M_T})$ for any parameters $W, W'$ and data $(x, y)$. This assumption is similar to assuming $\beta$-smoothness for the loss function, with a relaxation threshold $c$.
>
> Observe that the second component (parameters' change) in the above equation only depends on the parameter change after training on the dataset $D$ for time $t$. Consequently, its upper bound, as captured by the fluctuation term of gradient difference in Lemma 3.2, does not explicitly depend on the gradient difference between datasets $D$ and $D'$. Instead, the fluctuation term implicitly constrains the gradient difference at training time $T$ by the relaxed smoothness assumption and the computed upper bound on the parameters' change.
>
>
> ---
>
> > Additionally, I am curious about the evolution of the rank of  $M_T$ with respect to time T. The author briefly touches upon this aspect, but a more detailed explanation would be appreciated.
>
> We thank the reviewer for bringing up this important question. For simple models, such as generalized linear models (Song et al. [38]), this rank of gradient space is smaller than the rank of training data covariance matrix, and thus has a constant upper bound throughout training. For more complex deep neural networks, however, upper bounding the rank is significantly more difficult. This is due to the additive noise, which makes the gradient subspace not exactly low-rank, as the model parameters have non-zero probability of landing at regions where the gradients have high rank (or even full rank). Nevertheless, prior empirical results (such as [a, Figure 2] and [b, Figure 2]) show promises that the gradient space could have a constant or even increasingly small rank as training time $T$ increases. Intuitively, this is the effect of optimization, when the gradient at the reached optima in training has low rank. It remains an interesting open problem to establish theoretical bounds for the evolution of the rank of gradient space during training.
>
> On the other hand, there is now a significant amount of literature [1, 26, 27, a, b] that enforces low rank for the gradient space artificially during training. This is especially popular for public-data assisted private training, where a standard approach is to estimate a low-rank gradient space on a public dataset [1, 26] or historical gradients [a, b], and then project the gradients to this space during private training.
>
>
> ---
>
> > Some suggestions and corrections for the paper:
>
> Thanks for the suggestions and pointers to typos, we will correct them in future revisions of the paper.
>
>
> **References**
>
> [a] Yu, D., Zhang, H., Chen, W., Yin, J., & Liu, T. Y. (2021, July). Large scale private learning via low-rank reparametrization. In International Conference on Machine Learning (pp. 12208-12218). PMLR.
>
> [b] Yu, D., Zhang, H., Chen, W., & Liu, T. Y. (2021). Do not let privacy overbill utility: Gradient embedding perturbation for private learning. arXiv preprint arXiv:2102.12677.

---

> > ### Comment · Reviewer_NB2b · 2023-08-21
> > **Thanks for the detailed explanation.**
> >
> > The authors have provided a clear response that addresses my concerns. Consequently, I have adjusted my score upwards.

---

> > > ### Author Response · Authors · 2023-08-21
> > >
> > > We appreciate your positive support on our work.
> > > According to your suggestions, we will clarify our theoretical results as well as numerical validations in the final version.

---

### Official Review · Reviewer_Ej4s · 2023-07-08

**Soundness:** 3 good
**Presentation:** 2 fair
**Contribution:** 2 fair
**Rating:** 5
**Confidence:** 3

**Summary:**

This paper theoretically analyzes how the width, depth, and initialization of deep neural networks affect the privacy utility trade-off. For the special setting of linearized networks, the paper shows that gradient norm and privacy loss hinge on the per-layer variance of the initialization distributions. Thus, the privacy trade-off can be governed by width, depth, and initialization. Interestingly, for LeCun's initialization, the utility can be amplified by increasing depth which is against common intuition that overparameterized models will worsen the privacy utility trade-off.

**Strengths:**

1. The theoretical finding on the connection between depth and privacy-utility trade-off is interesting, that the deeper network can achieve better tradeoff than the shallow ones given LeCun's initialization.
2. It is novel that the network properties, e.g., initialization is analyzed through the Lagevian dynamics on linearized networks.

**Weaknesses:**

1. The presentation coould be improved. Some typos in the formulations make the paper hard to follow.
   * KL privacy bound is used before the definition or explanation in Line 38. Especially, what is the relation between the KL privacy bound and differential privacy bound?
   * The definition of the non-lazy training regime in Table 1 is not defined.
   * For the definition of LeCun initialization, the $\beta$ is layer-independent in Line 262 as $\beta_2=\cdots =\beta_L={1\over m}$. But in Table 1, the $\beta$ is layer-dependent as $\beta_l = \frac{1}{m_{l-1}}$.
   * In Line 24, `is randomze` should be `is randomizing`.
2. The difference between LeCun's initialization $\beta_l=1/m_{l-1}$ and others $\beta_l=2/m_{l-1}$ is mainly in the factor, `2`. The authors could give a more intuitive explanation for why the factor is crucial for privacy.
3. More generally regarding the factor in initialization, analyzing the best factor (probably some value between 0 and 2) may lead to a counterintuitive trade-off between privacy and utility. If I follow the derivations in Line 264-265, the excess empirical risk for LeCun's initialization with $\beta_t=\frac{a}{m_{l-1}}$  with $a\in (0,2)$ could be proportional to $\frac{a}{2}^{L/2}$. This means that the risk bound will be greatly reduced when $a$ approaches zero. However, this is counterintuitive, as a smaller $a$ implies smaller $\beta_t$, and therefore model will be initialized almost as zero. As common knowledge, zero parameters are not favored for the initialization of a deep network. In brief, smaller initial parameters will quickly vanish the activations by layer and therefore the model cannot converge or of low utility. Unless I missed some assumptions, the derivation is problematic due to the above counterintuitive results.

**Questions:**

1. What is the relation between the KL privacy bound and the differential privacy bound? What is the motivation to study the KL privacy bound?
2. Can you explain the counterintuitive derivation given a variable factor in $\beta_l$? See Weakness 3 for details.

----- Update after reading rebuttals ---
The authors have addressed most of my concerns.

**Limitations:**

The authors did not discuss the limitations or potential social impacts of the paper.
1. The paper presents some promising privacy improvement through proper initialization and also demonstrate some increasing risks if networks are not properly initialized. The results could be an important signal for people to treat the initialization seriously. I suggest the authors make the social impact clearly in the paper.
2. The paper has some inherent limitations due to the analysis tools of Lagevian diffusion algorithms, for example, the requirement for linearized networks, large depths, etc. The limitations are hard to address in the scope of the paper but is worth to mention for readers to under the proper use cases.

---

> ### Author Rebuttal · Authors · 2023-08-09
>
>
> We thank the reviewer for the detailed comments and pointers to typos, which we clarify below.
>
> ---
>
> > [Definition for KL privacy]
>
> A randomized algorithm $\mathcal{A}$ satisfies $\varepsilon$-KL privacy if for any neighboring datasets $D$ and $D'$, we have that the KL divergence $KL(\mathcal{A}(D)\rVert \mathcal{A}(D'))\leq \varepsilon$, where $\mathcal{A}(D)$ denotes the algorithm's output distribution on dataset $D$.
>
> KL privacy and relaxed variants of it are commonly used in previous literature [b, c, d].
>
> ---
>
> > [Relation to differential privacy]
>
> KL privacy is a more relaxed, yet closely connected privacy notion to $(\varepsilon, \delta)$ differential privacy.
>
> 1. KL privacy and differential privacy are both worst-case privacy notions over all possible neighboring datasets.
>
> 2. KL privacy and differential privacy are both definitions based on the privacy loss random variable $\log\frac{P(A(D) = o)}{P(A(D’) = o)}, o\sim A(D)$ (following the definition in Abadi et al. Equation 1). KL privacy implies that the privacy loss random variable has a bounded first order moment, while differential privacy requires a high probability argument that the privacy loss random variable is bounded by $\varepsilon$ with probability $1-\delta$. Therefore, KL privacy is generally a more relaxed notion than differential privacy.
>
> 3. Translation to each other: For $\varepsilon = 0$, KL privacy (bounded first-order moment of privacy loss random variable) implies $(0, \delta)$-differential privacy with $\delta = \sqrt{\frac{\varepsilon}{2}}$ by Pinsker inequality. Higher order moments of the privacy loss random variable suffice to prove (\varepsilon, \delta)-differential privacy for $\varepsilon >0$. Reversely, $\varepsilon, \delta)$-DP with $\delta>0$ does not imply KL privacy, as the privacy loss random variable may be large at the tail event with $\delta$ probability.
>
> 4. Due to the connection to the privacy loss random variable (which closely connects to likelihood ratio attack), both differential privacy and KL privacy incur upper bound on the performance curve of inference attacks, such as the membership inference and attribute inference [32, 22], as we discuss in the footnote 2.
>
> ---
>
> > [Definition for non-lazy training in Table 1]
>
> Following the ‘condensed regime’ defined in [31, equation 16], we refer to non-lazy training as settings with unbounded ratio $\frac{\lVert W_t - W_0\rVert_2}{\lVert W_0\rVert_2}$ between the change of model parameters after GD training with time $t$, and the L2 norm of model parameters $W_0$ at initialization. We will add this in the revised version.
>
> ---
>
> > [Definition of per-layer variance of LeCun initialization]
> > For the definition of LeCun initialization, the $\beta$ is layer-independent in Line 262 as $\beta_2 = … = \beta_L = \frac{1}{m}$. But in Table 1, the $\beta$ is layer-dependent as $\beta_l = \frac{1}{m_{l-1}}$
>
> These two lines define the same per-layer variances. This is because $m_1 = … = m_{L-1} = m$ (as in the second line of Table 1 caption), i.e., all the hidden-layers have the same width. We will clarify this in future revisions of the paper.
>
> ---
>
> > The difference between LeCun's initialization and others is mainly in the factor, 2. The authors could give a more intuitive explanation for why the factor is crucial for privacy.
>
> See general response point 1 for a detailed discussion.
>
> ---
>
> > [Counter intuitive example of privacy utility trade-off for point initialization at 0]
>
> We thank the reviewer for bringing up this very insightful example. We will add it to future versions of the paper. **This is not a counterexample to our analysis, because we prove utility in excess empirical risk (not empirical risk).**
>
> Our privacy and utility bounds hold under setting initialization variance $\beta = a/m$ for $a$ that is in a constant order (independent of width m and depth L). If we set $a$ to be zero, the KL privacy bound (Corollary 4.2) and the excess empirical risk bound (Corollary 5.4) will indeed both be zero, as pointed out by the reviewer. This is because the linearized network (Equation 3) expanded at point initialization 0 is a family of constant prediction functions (that maps all input data to 0), no matter what the expansion weights $W$ are (due to the zero NN gradient at 0). Consequently, the gradient throughout noisy GD training is zero. Similarly, the excess risk, which is the training loss difference between the trained model and the optimal model in the family of linearized networks expanded initialization, is also zero.
>
> However, the empirical risk (not the excess empirical risk) is still high despite zero excess empirical risk, because the approximation error of linearized networks expanded at zero is large. That is, excess risk bound for the linearized network with zero initialization setting becomes meaningless.
>
> This is why in this paper we only analyze excess empirical risk bound for commonly used initialization distributions, for which the approximation error is small empirically (Figure 5 [a]). To guide the privacy utility trade-off for arbitrary initialization distributions, it would be necessary to conduct a more comprehensive utility analysis that covers approximation error for linearized networks under different initialization. We leave this as interesting future work.
>
> ---
>
> > [Discussion on limitation]
>
> We thank the reviewer for pointing out the limitation. We have updated our discussion in general response point 3.
>
>
> **References**
>
> [a] Ortiz-Jiménez, Moosavi-Dezfooli, Frossard. What can linearized neural networks actually say about generalization?. NeurIPS 2021.
>
> [b] Barber, Duchi. Privacy and statistical risk: Formalisms and minimax bounds. arXiv preprint arXiv:1412.4451.
>
> [c] Bassily, Nissim, Smith, Steinke, Stemmer, Ullman. Algorithmic stability for adaptive data analysis. STOC 2016.
>
> [d] Wang, Lei, Fienberg. On-average kl-privacy and its equivalence to generalization for max-entropy mechanisms. PSD 2016.

---

> > ### Comment · Reviewer_Ej4s · 2023-08-14
> > **Thank you for the responses**
> >
> > The authors' responses are helpful for my understanding of the paper and address my major concerns. I suggest the authors integrate them into the final version.

---

> > > ### Author Response · Authors · 2023-08-16
> > > **Thank you for the reply**
> > >
> > > We thank the reviewer for the positive reply. According to your suggestions, we will improve the presentation of this work and include the discussion in the revised version.
> > >
> > > We are happy that our clarifications addressed most of your concerns and please feel free to let us know if you have any remaining concerns or further questions.

---

### Official Review · Reviewer_LMCn · 2023-07-09

**Soundness:** 3 good
**Presentation:** 2 fair
**Contribution:** 2 fair
**Rating:** 6
**Confidence:** 2

**Summary:**

The authors propose a KL privacy bound for DP-SGD that depend on the width, depth, and initialisation of the neural network. They show that the privacy bound improves with increasing depth for the LeCun and Xavier initialisation while it degrades with increasing depth.

**Strengths:**

1. The authors came up with an original idea exploring a topic that has so far received little attention in the literature
2. the findings are somewhat surprising and unintuitive

**Weaknesses:**

1. Often there are mistakes in privacy proofs. Having some experiments with privacy auditing to back up the surprising results would make the results more credible.

2. There are many typos in the middle of the paper. Especially related to the use of articles (articles missing) or repetitive words.

3. The authors motivate their contribution by the loose privacy bounds of DP-SGD. But the privacy bounds are then not used to improve the privacy accounting of the neural network.

4. Limitations not discussed in depth

**Questions:**

1. The result about the improving privacy bound is really surprising. Do the authors have an intuition why this is the case?

**Limitations:**

Limitations are not explicitly mentioned. The authors touch upon the restrictive results on linearised networks but this limitation is mostly underplayed. It would be interesting to discuss more when this simplification can lead to worse results.

---

> ### Author Rebuttal · Authors · 2023-08-09
>
> We thank the reviewer for spotting the typos, we will correct them in revisions of the paper. Below we clarify the questions and concerns.
>
> ---
>
> > Often there are mistakes in privacy proofs. Having some experiments with privacy auditing to back up the surprising results would make the results more credible.
>
> We have added numerical experiments to validate our privacy bound under different width, depth and initialization. Please see general response point 2  for a detailed discussion, and see the attached pdf for complementary plots.
>
> ---
>
> > The result about the improving privacy bound is really surprising. Do the authors have an intuition why this is the case?
>
> See general response point 1 for a detailed discussion about intuition for improving privacy bound.
>
> ---
>
> >The authors motivate their contribution by the loose privacy bounds of DP-SGD. But the privacy bounds are then not used to improve the privacy accounting of the neural network.
>
> We are not arguing DP-SGD privacy bound is loose and do not claim a contribution for improving upon the DP-SGD privacy bound in terms of privacy accounting. Indeed, we focus on a different KL privacy notion and a different algorithm (without clipping), and are therefore not comparable with DP-SGD privacy bound. Instead, **we argue that prior analysis for privacy under overparameterization for the DP-SGD algorithm are overly general to network properties such as width, depth and initializations**. And this is due to the significant difficulty for analyzing the effect of gradient clipping in DP-SGD training.
>
> Our approach circumvents this difficulty of analyzing gradient clipping (and thus provides a precise understanding of privacy under overparameterization) by focusing on a new algorithm Langevin diffusion, i.e.,  noisy GD **WITHOUT** gradient clipping. To the best of our knowledge, there is no existing differential privacy bound for this setting (no clipping). Therefore, we focus on KL privacy (which is generally a more relaxed definition than DP), and analyze the interplay between overparameterization and privacy.
>
> We would like to highlight that our main result Theorem 3.1. is a general KL privacy bound that applies to training arbitrary deep neural networks (as exemplified by Lemma 3.2 and our added numerical KL privacy loss in the general response). For the special case of linearized networks, we are further able to analytically analyze how privacy and utility changes with model width, depth and initialization.
>
>
> ---
>
> - [Discussion on limitation]
>
> We thank the reviewer for pointing out the limitation of linearized network. We have added a discussion in our [general response point 3](https://openreview.net/forum?id=IKvxmnHjkL&noteId=jh61g9pgBi), and will clarify this in future revisions of the paper.

---

> > ### Comment · Reviewer_LMCn · 2023-08-18
> >
> > Thank you for the detailed response. After reading the rebuttal and the other reviewers’ comments, I have raised my score.

---

> > > ### Author Response · Authors · 2023-08-21
> > >
> > > Thanks for your support. In our final version, we will include the experimental validation and polish the language based on your suggestions.

---

### Author Rebuttal · Authors · 2023-08-09

# General response

We’d like to thank the reviewers for their insightful comments. Below we address the shared questions and concerns.

---

**1. Intuition for improving privacy bound**

On the one hand, when depth is fixed, increasing width $m$ always increases the gradient norm and thus harms our KL privacy bound Corollary 4.2., which is consistent with the empirical observations in the literature. On the other hand, if we inverse proportionally set the per-layer initialization variance with the width, and increase the network depth, then the gradient difference norm decreases (thus the KL privacy bound improves).

To see why this is happening, observe at the analytical computation of gradient with regard to each layer, which scales with the post-activation output at its previous layer. This pre-activation output is weighted sum of outputs from the preceding layer, and is thus proportional to the initialization variance $\beta$ times the number of neurons $m$ in one layer (i.e., the $\beta_im_i$ factor in Lemma 4.1). After that, the Relu activation flattens the gradient to zero with a 1/2 probability, which divides the factor by two to be $\frac{\beta_im_i}{2}$. By a recursive argument over the number of layers (i.e., depth L), we have that when $\frac{\beta_im_i}{2}<1$ (such as under LeCun and Xavier initialization), the gradient norm (and thus the KL privacy bound) decays exponentially with increasing depth. (See Lemma C.3 for detailed derivations.)

For LeCun initialization, such decrease of gradient norm under increasing depth is not a new phenomenon, and has been widely referred to as  “gradient vanishing” in the literature. A similar phenomenon is also observed in the context of robustness [46].

---

**2. Additional numerical experiments to validate privacy bounds**

**_Numerical estimation procedure._** Theorem 3.1 suggests that KL privacy upper bound scales with the squared gradient norm during training. This could be estimated by empirically average of gradient norm across training runs. We consider training dataset $D$ with 'car' and 'plane' of the CIFAR-10, and all of its neighboring datasets $D'$ that removes a record from $D$, or adds a test record to $D$. We run noisy GD with constant step-size $0.01$ for $50$ epochs on both datasets. .


**_Numerical KL privacy loss validates Lemma 3.2 and Lemma 4.1 for the the growth of KL privacy bound with regard to training time_** Figure 1 of the attached pdf shows numerical KL privacy loss under different initialization distributions, for fully connected networks with width $1024$ and depth $10$. We observe that the KL privacy loss grows linearly at the beginning of training ($<10$ epochs), which validates the first term in the KL privacy bound Lemma 3.2. Moreover, the KL privacy loss under LeCun and Xavier initialization is close to zero at the beginning of training ($<10$ epochs). This shows LeCun and Xavier initialization induce small gradient norm at small training time, which is consistent with our gradient norm bound Lemma 4.1.  However, when the number of epochs is large, the KL privacy loss in Figure 1 grows faster than linear accumulation, thus validating the second term in our KL privacy bound Lemma 3.2 that exponentially grows with training time.


**_Numerical KL privacy loss validates the dependency of KL privacy bounds (Lemma 3.2 and Lemma 4.1) on network width and depth under different initializations_** Figure 2 of the attached pdf show numerical KL privacy loss under different network depth, width and initializations, for a fixed training time. We observe that increasing width and training time always increases KL privacy loss. This is consistent with Lemma 4.1, which shows that increasing width worsens the gradient norm at initialization (given fixed depth), thus harming KL privacy bound Lemma 3.2 at the beginning of training. We also observe that the relationship between KL privacy and network depth depends on the initialization distributions and the training time. Specifically, in Figure 2 (a), when the training time is small (20 epochs), for LeCun and Xavier initializations, the numerical KL privacy loss improves with increasing depth when depth > 8. Meanwhile, when the training time is large (50 epochs) in Figure 2 (b), KL privacy loss worsens with increasing depth throughout all depths. This shows that when the training time is small, the choice of initialization distribution affects the dependency of KL privacy loss on increasing width and depth, thus validating Lemma 3.2 and 4.1.

---

**3. Social impact, Limitations and Future Works**

We thank the reviewers for pointing out several missing discussions. We summarize them below and will add to future revisions.

(1) For DNNs that satisfy a relaxed smoothness condition, our KL privacy bound Lemma 3.2 is dominated by gradient difference at initialization for small training time, while it scales with gradient space rank for large training time. It remains an open problem to compute this bound by estimating the gradient space rank, and to improve it under more relaxed assumptions.

(2) For the simplified setting of linearized neural networks, we prove analytical privacy utility trade-off that improves with increasing depth under certain initializations, given that the width is large enough. Future works include relaxing the assumptions on width and network. Our results also suggest that the choice of activation function crucially affects how privacy changes with increasing width and depth. It is an an interesting open problem as to whether such conclusions generalize to deep neural networks.

(3) Lastly, our utility analysis focus on excess empirical risk, which is only meaningful when the approximation error is small (which is the case for commonly used initialization distribution). To prove meaningful privacy utility trade-off under arbitrary initialization distributions, future works include a more comprehensive utility analysis in approximation error.

---

### Decision · Program_Chairs · 2023-09-21

**Decision:**

Accept (poster)

**Comment:**

The authors investigate a novel question of how initialization and the structure of neural networks affect the privacy guarantee of DP-SGD. This leads to a surprising finding that privacy bound improves with increasing depth for the LeCun and Xavier initialization while it degrades with increasing depth. The reviewers agree that the findings are interesting and the techniques are novel in the privacy literature.